# Optogenetics and electron tomography for structure-function analysis of cochlear ribbon synapses

**Rituparna Chakrabarti**[1,2,3†], **Lina María Jaime Tobón**[3,4,5†], **Loujin Slitin**[1,2,3†],
**Magdalena Redondo Canales**[1,2,3], **Gerhard Hoch**[4,5], **Marina Slashcheva**[6],
**Elisabeth Fritsch**[6], **Kai Bodensiek**[4], **Özge Demet Özçete**[3,4,5], **Mehmet Gültas**[7],
**Susann Michanski**[1,2,3], **Felipe Opazo**[2,8,9], **Jakob Neef**[3,4,5], **Tina Pangrsic**[3,4,5,10,11‡],
**Tobias Moser**[3,4,5,10*], **Carolin Wichmann**[1,2,3,10*]

[1]Molecular Architecture of Synapses Group, Institute for Auditory Neuroscience and InnerEarLab, University Medical Center Göttingen, Göttingen, Germany; [2]Center for Biostructural Imaging of Neurodegeneration, University Medical Center Göttingen, Göttingen, Germany; [3]Collaborative Research Center 889 "Cellular Mechanisms of Sensory Processing", Göttingen, Germany; [4]Institute for Auditory Neuroscience and InnerEarLab, University Medical Center Göttingen, Göttingen, Germany; [5]Auditory Neuroscience & Synaptic Nanophysiology Group, Max Planck Institute for Multidisciplinary Sciences, Göttingen, Germany; [6]Göttingen Graduate School for Neuroscience and Molecular Biosciences, University of Göttingen, Göttingen, Germany; [7]Faculty of Agriculture, South Westphalia University of Applied Sciences, Soest, Germany; [8]NanoTag Biotechnologies GmbH, Göttingen, Germany; [9]Institute of Neuro- and Sensory Physiology, University Medical Center Göttingen, Göttingen, Germany; [10]Multiscale Bioimaging: from Molecular Machines to Networks of Excitable Cells, Göttingen, Germany; [11]Synaptic Physiology of Mammalian Vestibular Hair Cells Group, Institute for Auditory Neuroscience and InnerEarLab, University Medical Center Göttingen, Göttingen, Germany

**\*For correspondence:**
tmoser@gwdg.de (TM);
Carolin.Wichmann@medizin.uni-goettingen.de (CW)

†These authors contributed equally to this work

**Present address:** ‡Experimental Otology Group, Department of Otolaryngology and InnerEarLab, University Medical Center Göttingen, Göttingen, Germany

**Abstract** Ribbon synapses of cochlear inner hair cells (IHCs) are specialized to indefatigably transmit sound information at high rates. To understand the underlying mechanisms, structure-function analysis of the active zone (AZ) of these synapses is essential. Previous electron microscopy studies of synaptic vesicle (SV) dynamics at the IHC AZ used potassium stimulation, which limited the temporal resolution to minutes. Here, we established optogenetic IHC stimulation followed by quick freezing within milliseconds and electron tomography to study the ultrastructure of functional synapse states with good temporal resolution in mice. We characterized optogenetic IHC stimulation by patch-clamp recordings from IHCs and postsynaptic boutons revealing robust IHC depolarization and neurotransmitter release. Ultrastructurally, the number of docked SVs increased upon short (17–25 ms) and long (48–76 ms) light stimulation paradigms. We did not observe enlarged SVs or other morphological correlates of homotypic fusion events. Our results indicate a rapid recruitment of SVs to the docked state upon stimulation and suggest that univesicular release prevails as the quantal mechanism of exocytosis at IHC ribbon synapses.

## Editor's evaluation

This is a technically compelling study that uses optogenetic methods and rapid flash-and-freeze tissue preservation techniques to provide new insights into how the ribbon synapses of cochlear

inner hair cells are able to transmit auditory signals at very high rates. The conclusions of the paper about the mechanisms underlying exocytosis from these synapses are well supported by the data. The findings of this study should be of interest to a broad audience of neurobiologists and sensory physiologists.

## Introduction

Ribbon synapses of cochlear inner hair cells (IHCs) are specialized to maintain high release rates over prolonged periods of time. Their landmark structure, the synaptic ribbon, tethers several dozens of synaptic vesicles (SVs) and keeps them close to the active zone (AZ) membrane (*Moser et al., 2020*; *Rutherford et al., 2021*; *Safieddine et al., 2012*; *Wichmann and Moser, 2015*). Deciphering the mechanisms of SV release and replenishment in IHCs is required to understand their efficient and indefatigable glutamate release. Ultrastructural analysis of SV pools in defined functional states, such as during phasic or sustained transmitter release, is an important approach to investigate presynaptic mechanisms in general.

Numerous studies based on electron tomography (ET) describe the presence of morphologically docked SVs at central synapses (e.g. *Hintze et al., 2021*; *Imig et al., 2014*; *Imig et al., 2020*; *Kusick et al., 2020*; *Maus et al., 2020*; *Siksou et al., 2007*). At such conventional synapses, docked SVs are thought to constitute the readily releasable pool (RRP) (*Schikorski and Stevens, 1997*), while SVs tethered to the AZ might represent morphological correlates for SV recruitment to the release sites (*Cole et al., 2016*; *Fernández-Busnadiego et al., 2010*; *Fernández-Busnadiego et al., 2013*; *Maus et al., 2020*; *Siksou et al., 2007*). Recruitment appears to involve a first step mediated by long tethers of up to 45 nm, followed by the formation of shorter tethers which might correspond to the soluble *N*-ethylmaleimide-sensitive-factor attachment receptor (SNARE) complex (*Cole et al., 2016*; *Fernández-Busnadiego et al., 2010*; *Imig et al., 2014*). Therefore, morphological features like tethering or docking might reflect different functional states of SVs en route to fusion.

Correlating function and structure ideally employs rapid immobilization of the synapses in defined functional states. Recently, SV dynamics were investigated by combining optogenetics with immobilization within milliseconds by high-pressure freezing (Opto-HPF). Optogenetics grants short, precise stimulation of neurons expressing the light-sensitive ion channel channelrhodopsin (ChR) (*Nagel et al., 2002*; *Nagel et al., 2003*). Such precise stimulation allowed to ultrastructurally resolve transient events of exo/endocytosis at several conventional synapses such as *Caenorhabditis elegans* neuromuscular junctions (*Kittelmann et al., 2013*; *Watanabe et al., 2013a*) and murine hippocampal synapses (*Borges-Merjane et al., 2020*; *Imig et al., 2020*; *Watanabe et al., 2013b*).

Until now, structure-function analysis of hair cell ribbon synapses relied on seconds to minutes range depolarization by high $K^+$ (*Chakrabarti et al., 2018*; *Jung et al., 2015a*; *Kroll et al., 2020*; *Lenzi et al., 2002*; *Pangrsic et al., 2010*; *Strenzke et al., 2016*). SVs situated close to the AZ membrane – referred to as the membrane-proximal (MP)-SV pool – are thought to represent the RRP, while SVs around the ribbon – ribbon-associated (RA)-SVs – seem to be recruited for release in a later phase (*Lenzi et al., 2002*). MP-SVs are often connected to the AZ membrane by tethers (*Chakrabarti et al., 2018*; *Frank et al., 2010*; *Vogl et al., 2015*) and seem to be organized in sub-pools based on the number of tethers and to which structure these tethers are connected. These sub-pools might represent different recruitment states of the SVs prior to docking. However, docked SVs are rare in IHCs at rest but become more frequent upon prolonged $K^+$ depolarization (*Chakrabarti et al., 2018*). Yet, $K^+$ depolarization does not enable a time-resolved analysis of exocytosis at IHC ribbon synapses, which undergo synchronous release of many SVs at stimulus onset and sustain release SVs at high rates upon continued stimulation.

A time-resolved analysis is also relevant when addressing the long-standing quest on whether the quantal mechanism of IHC exocytosis entails release of several SVs in a coordinated manner (multivesicular release [MVR]) or the SVs fuse independently from each other (univesicular release [UVR]) in IHCs. From postsynaptic recordings of the spiral ganglion neurons (SGNs), the high variability in the amplitude and shape of spontaneous excitatory postsynaptic currents (EPSCs) was initially interpreted as the release of multiple SVs in a more or less synchronized manner (*Glowatzki and Fuchs, 2002*). The alternative, classical model of UVR, was then proposed based on experiments and modeling (*Chapochnikov et al., 2014*), and further corroborated by direct measurements of single fusion

events (*Grabner and Moser, 2018*). In the UVR framework, amplitude and shape heterogeneity were attributed to glutamate release via a fusion pore with gradual progress toward full collapse fusion (*Chapochnikov et al., 2014*). Such different scenarios might be mirrored in the number of docked SVs, SVs sizes, and the distribution of SV pools. For instance, coordinated MVR by compound and/or cumulative fusion is expected to result in larger vesicles at the AZ.

Here, we implemented Opto-HPF of IHC ribbon synapses to capture the structural correlates of exocytosis. We modified a conventional high-pressure freezing machine (HPM) to control optical stimulation in correlation to freezing on a millisecond time scale. Our study revealed that upon depolarization (i) the number of docked SVs increases, (ii) MP-SVs reside closer to the AZ membrane, (iii) correlates of compound and/or cumulative fusion are lacking, and (iv) the total number of RA-SVs remains unchanged. Our results constitute the first report, to our knowledge, of morphological correlates to exocytosis occurring within milliseconds of stimulation at this highly specialized synapse.

## Results

### Verification of ChR2 expression in IHCs and long-term expression of ChR2

First, we verified the expression of ChR2 in IHCs of both mouse lines, Ai32VC cre[+] and Ai32KI cre[+] (*fl/+ cre[+]* or *fl/fl cre[+]*), using immunofluorescence microscopy (*Figure 1A,B,D*). Alexa-coupled anti-GFP antibodies detected the EYFP-tag of the ChR2 construct and showed a clear expression of the construct at the membrane. For better resolution, we also performed immunogold electron microscopy (*Figure 1C*) with pre-embedding gold-coupled anti-GFP nanobodies. Membrane expression of ChR2 was evenly distributed (*Figure 1C*, arrow), confirming that ChR2 was efficiently expressed at the plasma membrane of IHCs without any apparent intracellular accumulation. Overall, these results confirmed that the *Vglut3* promoter efficiently controlled cre-recombination in ~99% of the analyzed IHCs for both mouse lines similarly to previous studies using *Vglut3 cre* mice (*Jung et al., 2015b*; *Vogl et al., 2016*). Therefore, we decided to pool the results from both genotypes in the following sections, but also analyzed and presented the data for each genotype in the Figure supplements.

Having confirmed proper ChR2 expression in both lines, in a next step, we analyzed potential long-term effects of ChR2 expression on synaptic organization of IHCs in the Ai32KI line. Using confocal microscopy, we compared ribbon synapse numbers of Ai32KI cre[+] IHCs with WT littermate controls at three different age intervals: 4–5 months (G1), 6–7 months (G2), and 9–12 months (G3) (*Figure 1*, *Figure 1—figure supplement 1*, all values can be found in *Table 1*). WT IHCs showed the characteristic decline in the number of ribbon synapses associated with aging (WT G1=9.82 ± 0.87 vs. WT G3=6.39 ± 0.56; p=0.0077) (*Parthasarathy and Kujawa, 2018*; *Sergeyenko et al., 2013*). Comparably, ChR2-expressing IHCs showed a significant decline in the number of ribbon synapses at 6–7 and 9–12 months in comparison to 4–5 months (Ai32KI cre[+] G1=8.97 ± 0.48 vs. Ai32KI cre[+] G2=5.324 ± 0.43; p<0.0001; vs. Ai32KI cre[+] G3=5.69 ± 0.42; p=0.0005). Importantly, there were no differences in the number of ribbons between ChR2-expressing IHCs and WT from the same age groups. We therefore conclude that ChR2 expression does not alter the number of ribbon synapses arguing against adverse effects such as through a potential chronic ChR2-mediated depolarization.

### Depolarization of IHCs using optogenetics

To validate optogenetic stimulation of IHCs, we performed perforated patch-clamp recordings of ChR2-expressing IHCs and applied 473 nm light pulses of different durations and intensities. Evoked photocurrents and photopotentials were measured in voltage-clamp and current-clamp mode, respectively. We employed TEA-Cl and Cs[+] in the bath solution in order to partially block K[+] channels and facilitate photodepolarization. Compared to other compositions (*Jaime Tobón, 2015*), we found 20 mM TEA-Cl and 1 mM Cs[+] to support both sufficient amplitudes and acceptable decay kinetics of photodepolarization but not to obviously alter the resting membrane potential. Under these conditions, strong short light pulses of 5 ms depolarized the cell by more than 50 mV (i.e. going from a holding potential of –84 mV up to –30 mV; *Figure 2A*). Longer light pulses of 10 and 50 ms caused stronger depolarizations even at low irradiances (*Figure 2B, C and D*), even though the peak of photodepolarization was reached with considerable delays (*Figure 2E*). On average, the peak was reached within 10–20 ms after the onset of the light pulse for irradiances above 1 mW/mm²

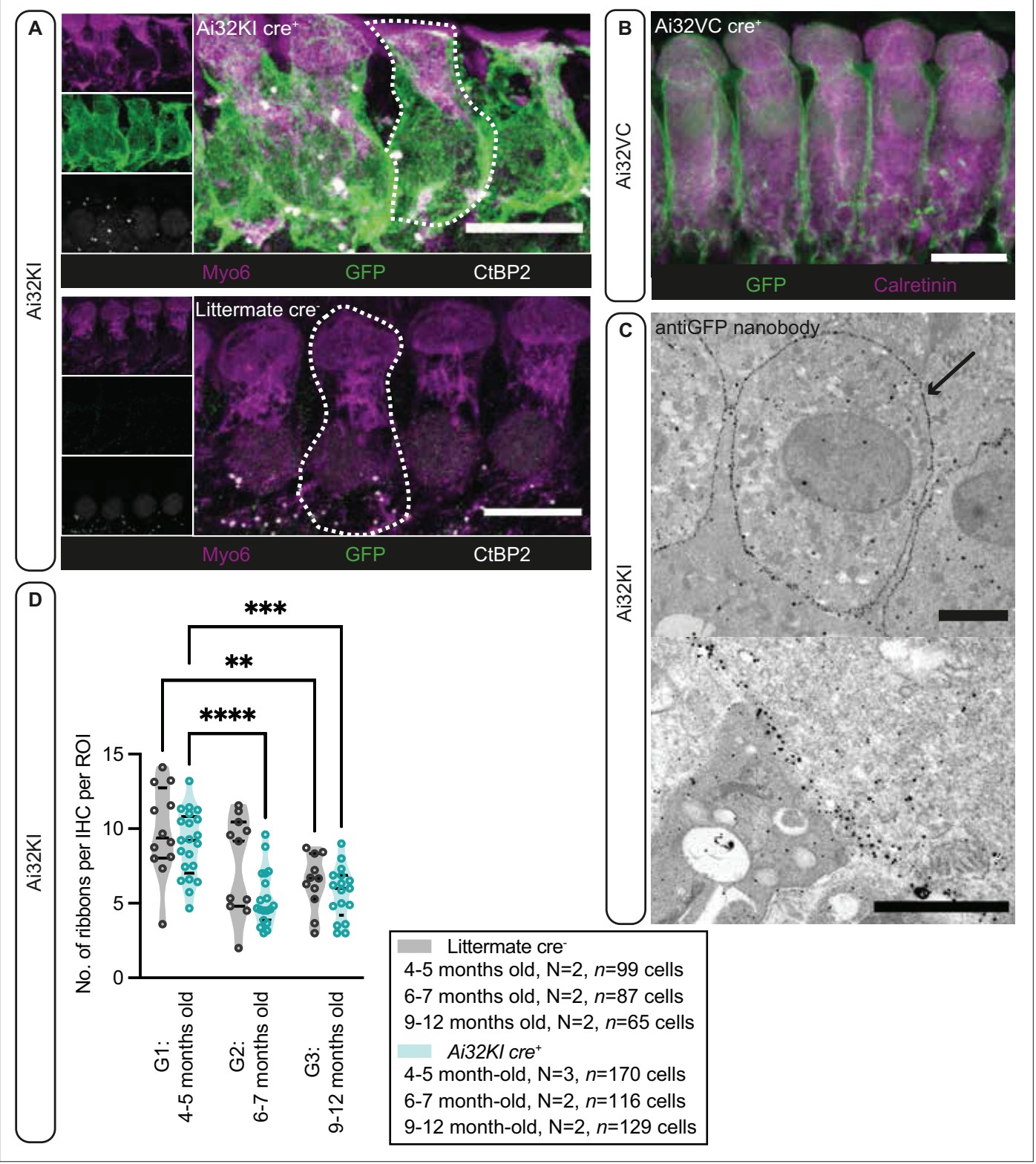

**Figure 1.** Plasma membrane expression of ChR2 in inner hair cells (IHCs). (**A**) IHCs express ChR2 in the plasma membrane. Maximal projection of confocal Z-stacks from the apical turn of the anti-GFP labeled organ of Corti from a 4- to 5-month-old Ai32KI cre+ mice (upper panel) and its littermate control (WT) (lower panel). Myo6 (magenta) was used as counterstaining and CtBP2 labeling shows ribbons at the basolateral pole of IHCs (examples outlined). ChR2 expression (green) is observed along the surface of Ai32KI cre+ IHCs (upper panel) but not in WT IHCs (lower panel). Scale bar, 10 µm. (**B**) Maximal intensity projection of confocal Z-stacks from the apical coil of anti-GFP labeled Ai32VC cre+ IHCs. Calretinin immunostaining was used to

*Figure 1 continued on next page*

*Figure 1 continued*

delineate the IHC cytoplasm. Scale bar, 10 µm. (**C**) Pre-embedding immunogold labeling for electron microscopy was performed with a gold-coupled anti-GFP nanobody recognizing the EYFP of the ChR2 construct. A clear localization at the plasma membrane of IHCs is visible (arrow, upper panel). Scale bar, 2 µm. The magnified image of the membrane labeling is shown in the lower panel. Scale bar, 1 µm. (**D**) The average number of ribbons per IHC is similar between ChR2-expressing IHCs and WT IHCs. Both Ai32KI cre[+] and WT mice showed a comparable decrease in the number of ribbons with age (**p<0.01, ***p<0.001, ****p<0.0001; two-way ANOVA followed by Tukey's test for multiple comparisons). *N*=Number of animals. Violin plots show median and quartiles with data points overlaid.

The online version of this article includes the following figure supplement(s) for figure 1:

**Figure supplement 1.** Long-term expression of ChR2 at inner hair cell (IHC) plasma membrane.

(*Figure 2E*). Higher irradiances decreased the time to peak of the photodepolarization for all stimulation durations. Light pulses of 10 ms at 2–11.5 mW/mm$^2$, in good correspondence to the irradiance recorded for the high-pressure freezing machine with a peak irradiance of 6 mW/mm$^2$ (*Figure 3E and F*), could depolarize the IHC by 27–65 mV (*Figure 2B, D*). Notably, IHCs were photodepolarized by 20 mV within the first 3–10 ms of the light pulse, which presumably suffices to trigger vesicle release on IHCs (assuming a resting potential of –58 mV and based on release thresholds reported *Goutman and Glowatzki, 2007*; *Özçete and Moser, 2021*). The results obtained from both lines were comparable, as shown in *Figure 2—figure supplement 1*.

To test whether our stimulation paradigms would prompt neurotransmitter release, we performed whole-cell patch-clamp recordings from individual afferent boutons contacting ChR2-expressing IHCs. Light pulses of low intensities and short duration were sufficient to trigger release from individual AZs, as proven by EPSCs recorded from three boutons contacting different IHCs (*Figure 4A*). Consistent with previous experiments employing K$^+$ or voltage-clamp stimulation of IHCs (e.g. *Chapochnikov et al., 2014*; *Glowatzki and Fuchs, 2002*; *Goutman and Glowatzki, 2007*), we found variable amplitudes of individual EPSCs. The maximum amplitude of the evoked EPSCs varied between different boutons (from –200 to –700 pA), but remained fairly similar for one individual bouton regardless of the light pulse duration (*Figure 4B*). In contrast, light pulse duration had a major impact on the duration of the evoked EPSCs (i.e. the duration of exocytosis; *Figure 4C*) and consequently, on the total charge transfer (*Figure 4D*). In response to a 50 ms stimulation, the evoked release lasted three times longer than in response to a 10 ms light pulse and could reach up to double the initial charge transfer. In line with the recorded IHC photodepolarization, longer light pulses required lower intensities to

**Table 1.** Ribbon counts in three different age groups of ChR2-expressing inner hair cells (IHCs) and their WT controls. Data are presented as mean ± SEM. p-Values are calculated by two-way ANOVA followed by Tukey's test for multiple comparisons. Results of the comparisons between age groups of the same genotype and between genotypes of the same age group are reported. Significant results are indicated with **p<0.01, ***p<0.001, and ****p<0.0001.

| | N$_{animals}$ | N$_{ROIs}$ | n$_{cells}$ | Ribbon count (mean ± SEM) | Age comparison p-Value | Test | Genotype comparison p-Value | Test |
|---|---|---|---|---|---|---|---|---|
| | | | | | **, WT G1 vs. WT G3 | | ns, WT G1 vs. Ai32KI cre[+] G1 | |
| WT G1 | 2 | 12 | 99 | 9.82±0.87 | **0.0077** | Two-way ANOVA | 0.9124 | Two-way ANOVA |
| | | | | | ns, WT G2 vs. WT G1 | | ns, WT G2 vs. Ai32KI cre[+] G2 | |
| WT G2 | 2 | 11 | 87 | 7.60±0.99 | 0.2045 | Two-way ANOVA | 0.1046 | Two-way ANOVA |
| | | | | | ns, WT G3 vs. WT G2 | | ns, WT G3 vs. Ai32KI cre[+] G3 | |
| WT G3 | 2 | 11 | 65 | 6.39±0.56 | 0.8179 | Two-way ANOVA | 0.9689 | Two-way ANOVA |
| | | | | | ****, Ai32KI cre[+] G1 vs. Ai32KI cre[+] G2 | | | |
| Ai32KI cre[+] G1 | 3 | 21 | 170 | 8.97±0.48 | **<0.0001** | Two-way ANOVA | | |
| | | | | | ns, Ai32KI cre[+] G2 vs. Ai32KI cre[+] G3 | | | |
| Ai32KI cre[+] G2 | 2 | 19 | 116 | 5.32±0.43 | 0.9968 | Two-way ANOVA | | |
| | | | | | ***, Ai32KI cre[+] G3 vs. Ai32KI cre[+] G1 | | | |
| Ai32KI cre[+] G3 | 2 | 17 | 129 | 5.69±0.42 | **0.0005** | Two-way ANOVA | | |

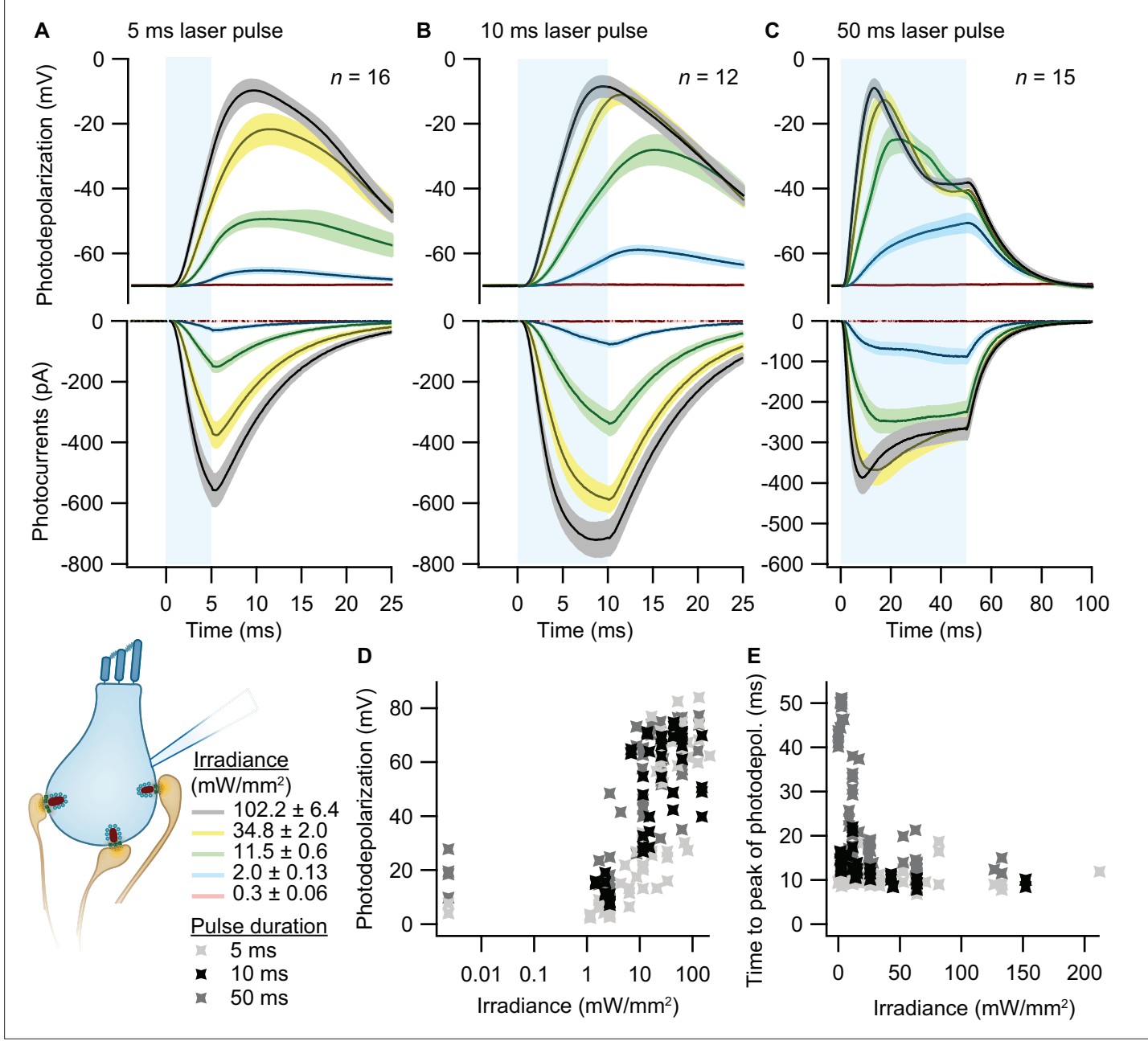

**Figure 2.** Optogenetic depolarization of inner hair cells (IHCs). IHCs expressing ChR2 (Ai32VC cre[+] and Ai32KI cre[+]) were optogenetically stimulated by 473 nm light pulses of increasing irradiance (mW/mm²; presented as mean ± SEM). (**A–C**) Average photocurrents (lower panel) and photodepolarizations (upper panel) of patch-clamped IHCs during 5 ms (A, $n_{cells}$ = 16, $N_{animals}$ = 7), 10 ms (B, $n_{cells}$ = 12, $N_{animals}$ = 4), and 50 ms (C, $n_{cells}$ = 15; $N_{animals}$ = 6) light pulses of increasing irradiances (color coded). Mean is displayed by the continuous line and ± SEM by the shaded area. (**D–E**) Peak of photodepolarization (**D**) and time to peak (**E**) obtained for increasing irradiances of different lengths (light gray 5 ms, black 10 ms, dark gray 50 ms).

The online version of this article includes the following figure supplement(s) for figure 2:

**Figure supplement 1.** Comparison of optogenetic stimulation of inner hair cells (IHCs) from Ai32VC cre[+] and Ai32KI cre[+] mice.

trigger a response; 4–6 mW/mm² were sufficient for 50 ms light pulses. Moreover, as expected from the photodepolarization, EPSCs latency decreased with increasing irradiances (*Figure 4E*). Finally, we quantified the maximal EPSCs charge transfer at 20 ms ($Q_{20ms}$) and 50 ms ($Q_{50ms}$) after the onset of the light pulses. These time points reflect phasic RRP release (20 ms) and sustained release (50 ms) of IHC ribbon synapses (*Johnson et al., 2017*; *Michalski et al., 2017*; *Moser and Beutner, 2000*). For light stimulations of 6–7 mW/mm², $Q_{20ms}$ ranged from 236 pC up to 1300 pC while $Q_{50ms}$ ranged from 850

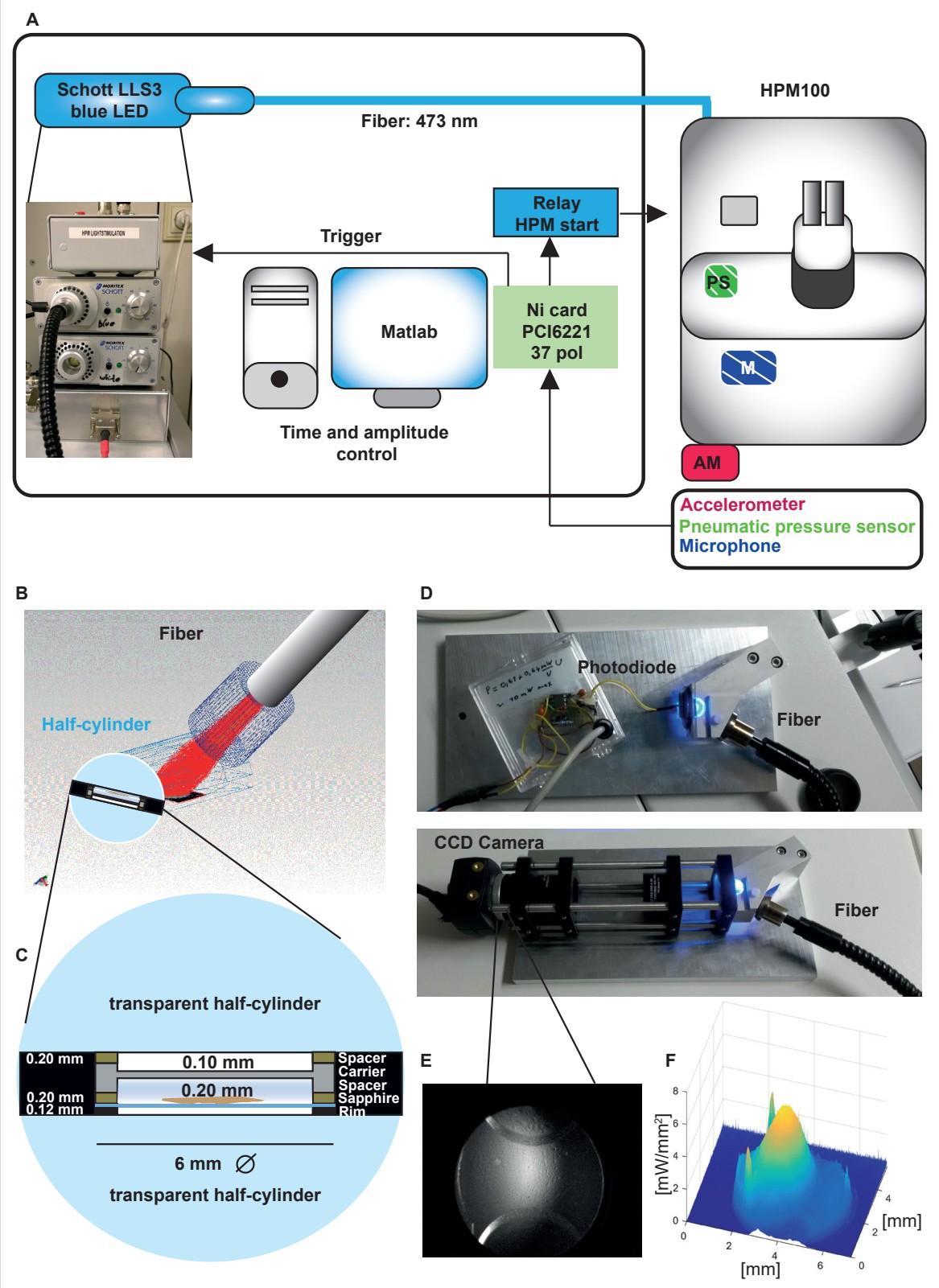

**Figure 3.** Opto-high-pressure freezing machine (HPM) setup and irradiance calculation for the HPM. (**A**) Simplified illustration depicting the components of the external setup installed to control the light stimulation (irradiance and duration), to determine the time point of the freezing onset and command relay (blue, control unit) to the sensors (accelerometer [red, outside, AM], microphone [patterned blue, inside, M], pneumatic pressure sensor, green, inside, PS) to initiate the mechanical sensing process of the HPM100 (*Source code 3*). (**B**) Fiber – cartridge arrangement in the HPM100

*Figure 3 continued on next page*

*Figure 3 continued*

with the fiber at an angle of 60° to the upper half-cylinder: Sample plane: black, fiber: gray, Light rays: red. Mechanical components of HPM100 are not shown. (**C**) Sample loading scheme. (**D**) Re-build chamber replica to enable irradiance calculation externally. (**E**) CCD image of the photodiode in the sample plane. (**F**) The spatial irradiance distribution with a peak irradiance of ~6 mW/mm² at 80% intensity of the light-emitting diode (LED) was calculated using a self-written MATLAB routine *intensityprofilcalculator.m* (***Source code 4***). Depicted are pixel values in irradiance. The complete workflow for Opto-HPF is presented in ***Figure 3—figure supplement 1***.

The online version of this article includes the following figure supplement(s) for figure 3:

**Figure supplement 1.** Workflow of Opto-HPF.

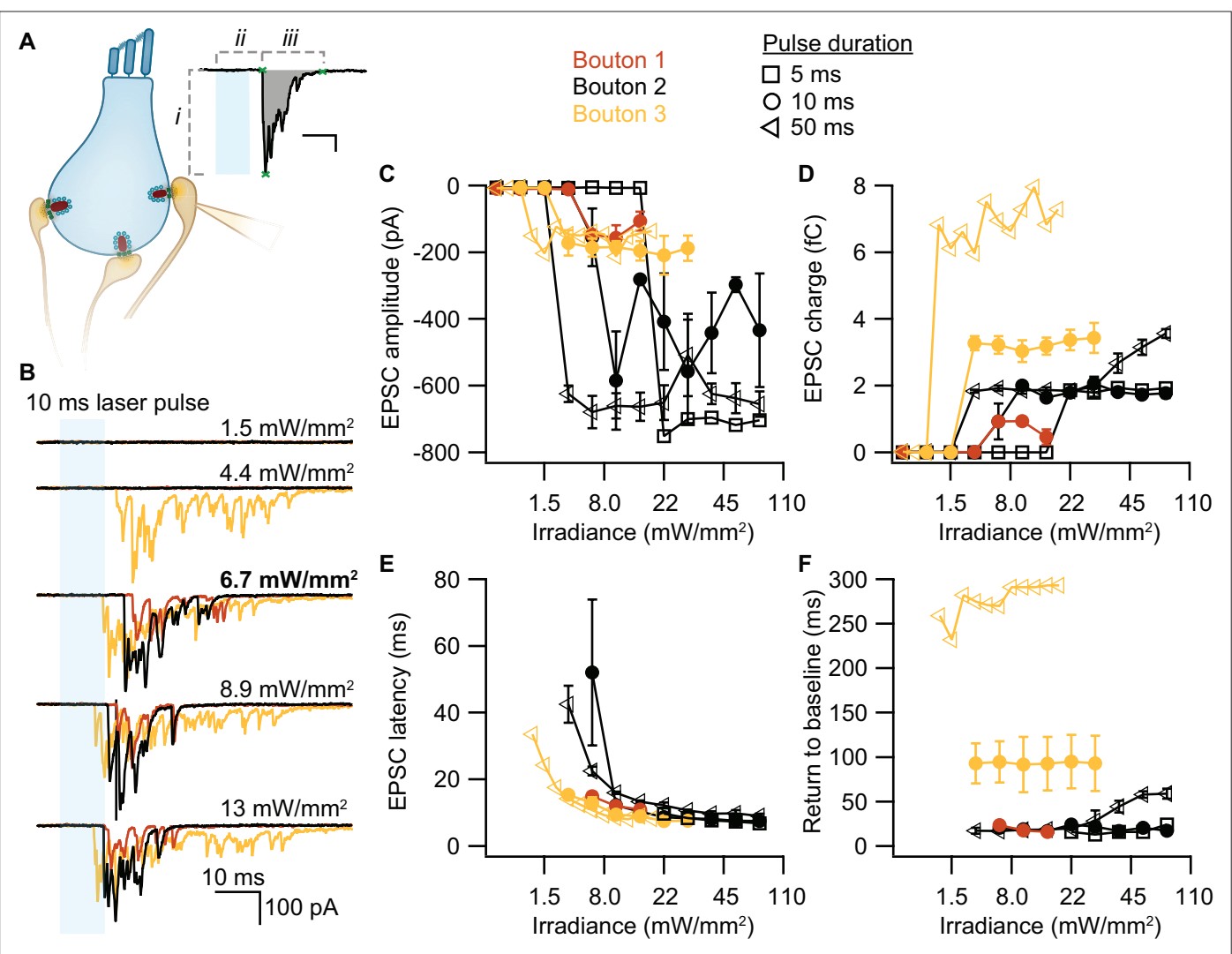

**Figure 4.** Optogenetically-triggered exocytosis at individual ribbon synapses. (**A**) Excitatory postsynaptic currents (EPSCs) upon the optogenetic stimulation of Ai32VC cre⁺ inner hair cells (IHCs) were recorded using whole-cell patch-clamp of the contacting bouton. The response was quantified in terms of EPSC amplitude (i), charge (gray area), latency (ii), and return to baseline (iii) (***Source code 2***). Scale bar as in panel B. (**B**) Recorded EPSCs from three different postsynaptic boutons (different colors for boutons 1–3) in response to increasing light intensities. (**C–F**) Amplitude (**C**), charge (**D**), latency (**E**), and return to baseline (**F**) of the light triggered EPSCs to different light pulse durations (5 ms squares; 10 ms circles; 50 ms triangles). $n_{boutons}$= 3, $N_{animals}$= 2.

to 2450 pC. Importantly, the first recording of all three boutons showed substantial release exceeding 400 pC at these time points. The return to baseline differed among the three boutons and for pulse duration (*Figure 4F*), but neurotransmitter release lasted for at least 15 ms. These electrophysiological findings demonstrate that the chosen stimulation paradigms are sufficient to trigger phasic and sustained SV exocytosis in ChR2-expressing IHCs.

## Developing a method to combine optogenetic stimulation with precisely timed freezing

To correlate structure and function, we performed Opto-HPF, followed by freeze substitution (FS) and subsequent ET (*Figure 3—figure supplement 1*). The commercial HPM100 comes with limitations: the light stimulation duration cannot be set precisely and the precise time point of freezing is not provided. Therefore, groups had previously already modified the HPM100 with custom-made settings adapted to the needs of central synapses (*Watanabe et al., 2013b*). Applying this method to a sensory synapse, which does not operate with all-or-nothing action potential stimulation, we established a more general framework for Opto-HPF using the HPM100.

## Setup for stimulation and freezing relay

First, we determined the irradiance that reaches the sample with a re-built chamber replica equipped with an optical fiber. The radiant flux at the sample was measured to be 37.3mW with a peak irradiance of 6 mW/mm$^2$ where the samples are positioned (*Figure 3*). This irradiance is in accordance with the irradiance values that led to a sufficient depolarization of IHCs to trigger exocytosis in our cell-physiological experiments. With our custom-made setup, we controlled the irradiance, stimulus duration, and the coupling of stimulus onset with the freezing of the specimen (*Source code 3*).

Our three incorporated external sensors, (i) an *accelerometer*, (ii) a *microphone*, and (iii) a *pneumatic pressure sensor* (*Figure 3*, *Figure 5*) allowed us to calculate for each shot the absolute time scale from the *HPM start* till the specimen is reaching 0°C ($T_{\text{HPM delay from START}}$, *Figure 5*).

The pneumatic pressure sensor was located directly at the pneumatically steered needle valve in front of the freezing chamber (*Figure 5A–C*). In contrast to the other sensors, the pneumatic pressure sensor provided a reliable signal of the moment when the needle valve opened to allow influx of pressurized LN$_2$ into the freezing chamber (*Figure 5C*; green curve; the green arrowhead points out the time point when the needle valve opens). We used this point to align the curves obtained from the internal sensors, which show the internal pressure build-up (*Figure 5C*, gray curve) and the gradual temperature decline (*Figure 5C*, purple curve) inside the freezing chamber after the opening of the needle valve. We set the onset (StimStart) of a 100 ms light stimulation between *HPM start* (t=0) and $T_{\text{HPM delay from START}}$ (calculated for each individual freezing; *Figure 5D*).

To obtain short stimulations (ShortStim), StimStart was set at 425 ms. Based on the correlation of the pneumatic pressure sensor curve with the internal pressure and temperature curves, the samples were stimulated during ~17 to ~25 ms before the freezing onset (*Figure 5D*, lower panel). To obtain longer stimulations (LongStim), StimStart was set at 390 ms, which resulted in light stimulation durations from ~48 to ~76 ms before the freezing onset (*Figure 5D*, upper panel).

## First ultrastructural analysis of optogenetic stimulated IHC ribbon synapses, coupling structure to function

We analyzed the ultrastructural changes upon precise optical stimulation of ChR2-expressing IHCs from the apical turn (Ai32VC cre$^+$ and Ai32KI cre$^+$). Our two stimulation paradigms (ShortStim and LongStim, *Figure 6C and D*) aimed to capture two functional states of exocytosis at IHCs. A ShortStim (~17–25 ms) might reflect the changes after/during RRP release, while a LongStim (~48–76 ms) might reflect sustained exocytosis (*Moser and Beutner, 2000*; *Rutherford and Roberts, 2006*). We included two controls (i) B6J under light stimulation (B6J Light; *Figure 6A*) to address potential direct effects of light exposure and (ii) ChR2-expressing IHCs (Ai32VC cre$^+$ and Ai32KI cre$^+$) without any light stimulation (ChR2 NoLight; *Figure 6B*) as controls with the same genetic background of the stimulated samples. For the light control on B6J, we chose the long stimulation protocol assuming that potential direct light effects are strongest with the longer exposure. *Table 2* includes the number of ribbons and animals included from each genotype in the analysis.

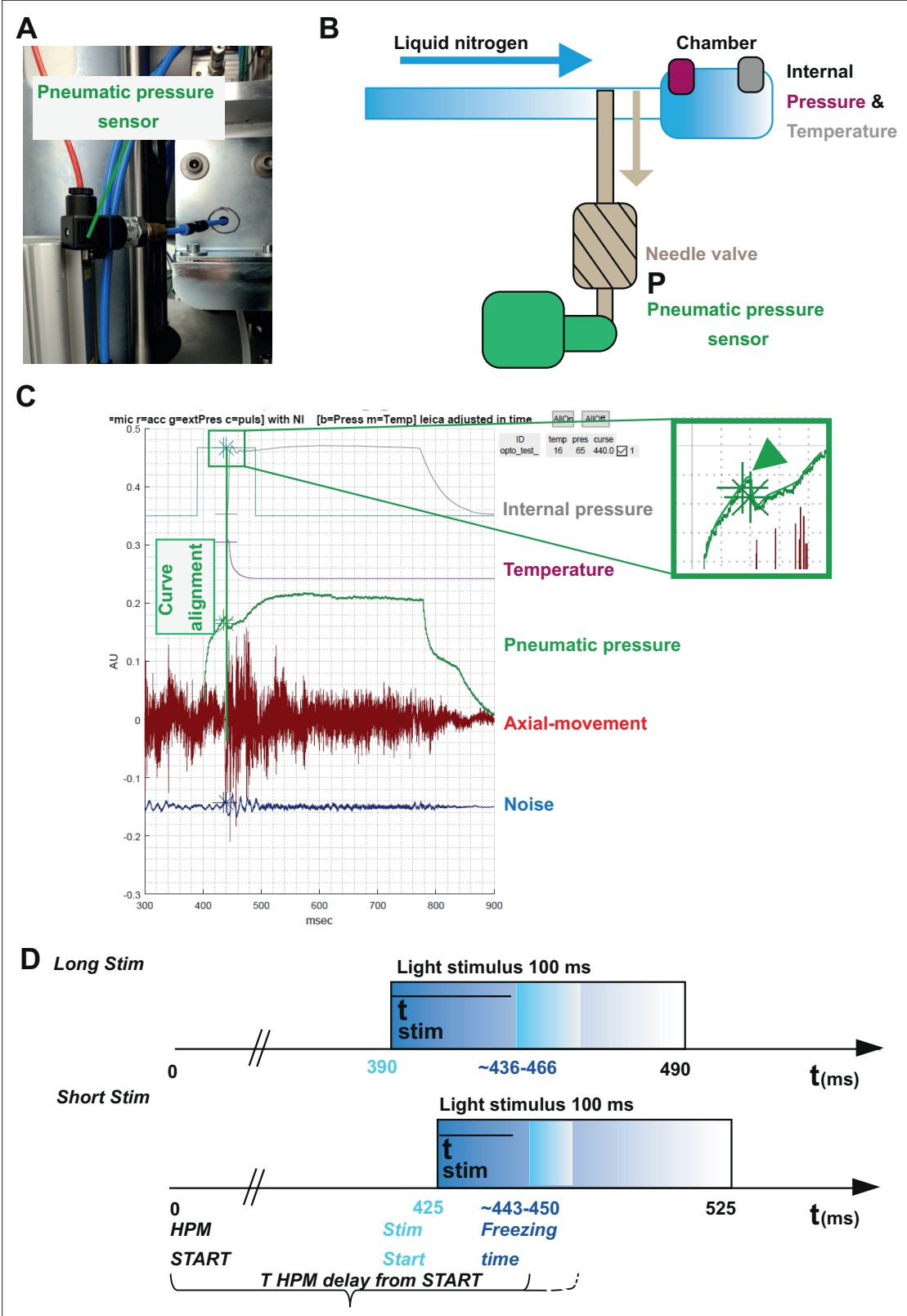

**Figure 5.** Correlating the sensor signals to internal pressure and temperature measured inside the high-pressure freezing machine (HPM). (**A**) Pneumatic pressure sensor inside the HPM100. (**B**) Scheme of the localization of the pneumatic pressure sensor below the pneumatic needle valve, which allows LN₂ influx in the chamber for freezing. (**C**) Depicted are the curves from the different sensors aligned by using the MATLAB GUI (*Source code 5*). The curves of the pressure build-up start and temperature corresponding decline are aligned to the steep drop in the pneumatic pressure curve (green

*Figure 5 continued on next page*

*Figure 5 continued*
arrowhead, inset). (**D**) The outline of the optical stimulation incorporated with HPF: 100 ms light pulse set to a 390 ms delay from HPM start resulted in a stimulation duration of 48–76 ms before freezing ('LongStim') and that set to a 425 ms delay from HPM start in a stimulation of 17–25 ms ('ShortStim').

We started our ultrastructural analysis by determining the total count of MP-SVs and RA-SVs (*Figure 6—figure supplement 1A*). We found no alterations in the size of the MP-SV and RA-SV pools among the various conditions and controls (*Figure 6E*), except for a larger MP-SV pool in ChR2 NoLight compared to B6J Light. This difference could be due to differences in their genetic background and/ or the expression of ChR2 and highlights the importance of including ChR2 NoLight controls in our experimental design. Our estimates of MP-SVs/tomogram in control conditions compare to previous reports (B6J Light 9.73±0.796; ChR2 NoLight 14.88±0.935; Wt Rest 11.7±0.75 [*Chakrabarti et al., 2018*]; Wt Rest 18.5±1.5 [*Kroll et al., 2020*]).

Notably, one limitation of the study is that we cannot tell where the ribbon synapses were located within a given IHC. As synapse properties vary between pillar and modiolar IHC sides (*Kantardzhieva et al., 2013*; *Merchan-Perez and Liberman, 1996*; *Michanski et al., 2019*; *Ohn et al., 2016*), we expect this variance to contribute to the variance of synaptic properties observed for each condition. However, ribbon AZs were randomly sampled for ET; therefore, we expect this variability to be similar across conditions.

## Enhanced SV docking in correlation to stimulus duration

Next, we performed an in-depth analysis of morphological MP-SV sub-pools among the various conditions (*Figure 6F*). Examples for SVs of the morphological sub-pools as well as further representative ribbons from each condition are presented in *Figure 6—figure supplement 2*. As the full inclusion of the ribbon is relatively rare in 250 nm sections (*Figure 6A–D*), representing a limitation of the study, we compared the fractions of non-tethered, tethered, and docked MP-SVs as done previously (*Chakrabarti et al., 2018*). The fraction of non-tethered SVs decreased after a short light pulse (B6J Light = 0.28 ± 0.035, ChR2 NoLight = 0.23 ± 0.026, ChR2 ShortStim = 0.14 ± 0.035), while the fraction of tethered SVs decreased upon a long stimulation (B6J Light = 0.72 ± 0.033, ChR2 NoLight = 0.75 ± 0.025, ChR2 ShortStim = 0.77 ± 0.049, ChR2 LongStim = 0.57 ± 0.034, representative top views depicted in *Figure 6B and D*). Values and information about statistics can be found in *Table 2* and a separate quantification of the used ChR2 mice can be found in *Figure 6—figure supplement 1*.

The fraction of morphologically docked SVs increased upon optogenetic stimulation (*Figure 6F*), being more prominent upon a long stimulation (B6J Light = 0.01 ± 0.006, ChR2 NoLight = 0.01 ± 0.009, ChR2 ShortStim = 0.10 ± 0.034, ChR2 LongStim = 0.20 ± 0.025; representative top views depicted in *Figure 6C and D*). We conclude that optogenetic stimulation changes the sub-pools of MP-SVs, with a notable increase of docked SVs proportional to the stimulation duration and fewer tethered and non-tethered SVs.

## SV distances to the PD and AZ membrane decrease upon long stimulation

It was previously proposed that SVs are recruited to the AZ membrane via tethers, a process that occurs rather close to the presynaptic density (PD) (*Chakrabarti et al., 2018*; *Frank et al., 2010*; *Vogl et al., 2015*). Therefore, we evaluated the distances of all MP-SVs to the AZ membrane and to the PD. We found that the distances of MP-SVs to the AZ membrane decreased upon stimulation (*Figure 7Ai*), indicating recruitment of SVs to the AZ membrane. However, enhanced docking to the AZ membrane mainly accounted for the decrease of average SV distances to the AZ membrane, as it was no longer significant when the non-tethered and tethered sub-pools were analyzed independently (*Figure 7Aii*). Moreover, SVs were found closer to the PD upon long light stimulation (*Figure 7Bi*), likely bringing them close to the voltage-gated $Ca^{2+}$ channels that are situated underneath the PD (*Neef et al., 2018*; *Pangrsic et al., 2018*; *Wong et al., 2014*). When the distance to PD was analyzed for the individual morphological sub-pools, non-tethered SVs of ChR2 LongStim appeared to be in closer proximity to the PD compared to ChR2 NoLight and ChR2 ShortStim (*Figure 7Bii*) although this did not reach statistical significance. In addition, the few docked SVs found in the control conditions were located at a greater distance from the PD compared to the ChR2 stimulated conditions (*Figure 7Bii*, all values can be found in *Table 2* and an overview of the number of docked SVs from this study in

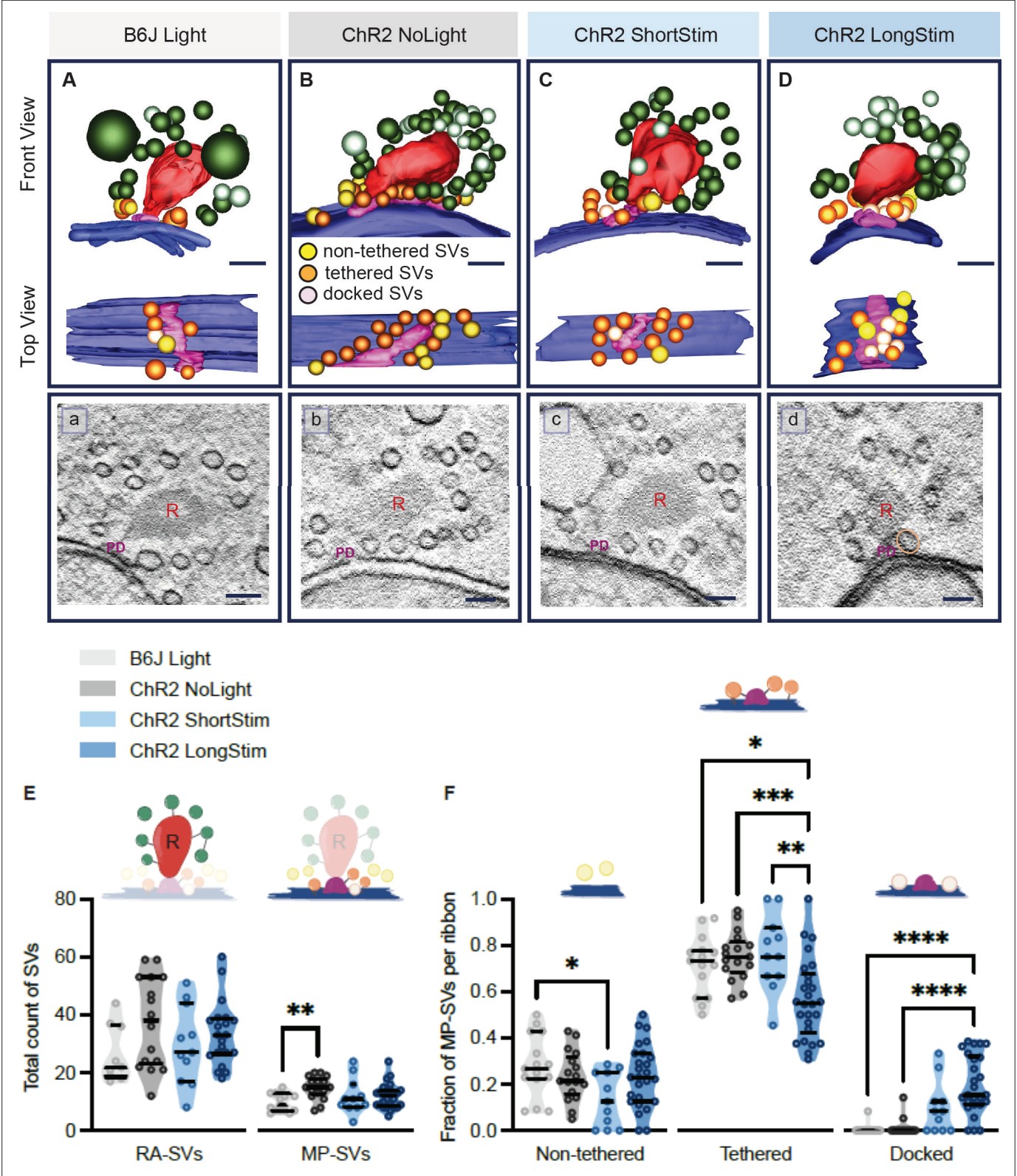

**Figure 6.** Functional active zone (AZ) states differ in their morphologically defined vesicle pools. Representative tomographic 3D reconstructions of AZs at (**A**) B6J Light, (**B**) ChR2 NoLight, (**C**) ChR2 ShortStim, and (**D**) ChR2 LongStim conditions displayed in both front view (upper part of the panel) and top view (ribbon removed from the model, lower part of the panel). (**a–d**) Corresponding virtual sections of A–B. The AZ membrane is shown in blue, presynaptic density in pink (and indicated with PD), ribbons in red (and indicated with R), membrane-proximal (MP)-synaptic vesicles (SVs) (non-tethered

*Figure 6 continued on next page*

*Figure 6 continued*

in yellow, tethered in orange and docked in light pink), ribbon-associated (RA)-SVs (green, light green). Magnification ×12,000; scale bars, 100 nm. (**E**) Total count of SVs per pool (RA- and MP-SV pools), per ribbon. (**F**) The fraction of non-tethered, tethered, and docked MP-SVs per ribbon. Data are presented in mean ± SEM. *p<0.05, **p<0.01, ***p<0.001, and ****p<0.0001. Statistical test: one-way ANOVA followed by Tukey's test (parametric data) and Kruskal-Wallis (KW) test followed by Dunn's test (non-parametric data). **MP-SV pool**: B6J Light: $n_{ribbons}$ = 15, $N_{animals}$ = 2. ChR2 NoLight: $n_{ribbons}$ = 17, $N_{animals}$ = 4. ChR2 ShortStim: $n_{ribbons}$ = 11, $N_{animals}$ = 1. ChR2 LongStim: $n_{ribbons}$ = 26, $N_{animals}$ = 4. **RA-SV pool**: B6J Light: $n_{ribbons}$ = 9, $N_{animals}$ = 1, ChR2 NoLight: $n_{ribbons}$ = 17, $N_{animals}$ = 4. ChR2 ShortStim: $n_{ribbons}$ = 11, $N_{animals}$ = 1. ChR2 LongStim: $n_{ribbons}$ = 21, $N_{animals}$ = 3.

The online version of this article includes the following figure supplement(s) for figure 6:

**Figure supplement 1.** Analysis of morphologically defined vesicle pools for each genotype.

**Figure supplement 2.** Different morphological sub-pools within the membrane-proximal (MP) pool at ribbon synapse active zone (AZ).

comparison to the literature in *Table 3*). We conclude that upon stimulation, SVs are rapidly recruited to dock to the AZ membrane potentially closer to the $Ca^{2+}$ channels. Similar results were obtained for the individual genotypes (*Figure 7—figure supplement 1*).

While the present study focuses on exocytosis, we also wanted to check if clathrin-coated (CC) endocytic structures were intermixing with the MP-pool at the AZ. Therefore, the number of CC structures within 0–500 nm away from the PD was determined (*Supplementary file 1*). In all conditions, CC structures were not very frequent at these locations. Only once a clathrin-coated vesicle (CCV) was observed intermingled with the MP-SV pool and most CC structures were found 200–500 nm away from the PD. The diameter of these structures (excluding the CC) ranged from 63 to 120 nm, which was a slightly broader range than reported previously from fixed IHC ribbon synapses (*Neef et al., 2014*) and the CCVs were 8–78 nm away from the AZ membrane. However, due to their low abundance, we refrained from statistical analysis.

## SV diameters remain largely unchanged in the MP-SV pool

In order to approach possible release mechanisms for IHC ribbon synapses, we determined the SV diameter for all MP-SVs. Previous studies using 15 min $K^+$ stimulation already excluded a large size increase upon prolonged stimulation of SVs close to the AZ membrane (*Chakrabarti et al., 2018*; *Chapochnikov et al., 2014*). However, early exocytosis phases could not be monitored at IHC ribbon synapses up to now. If homotypic or compound fusion takes place, one would expect an increase in diameter of SV close to the AZ membrane (*He et al., 2009*; *Lenzi et al., 2002*). We found no differences in the diameters of the MP-SVs between the stimulated and non-stimulated conditions (*Figure 7Ci*; values and details for statistics in *Table 2*). When separating the sub-pools in non-tethered, tethered, and docked, we found an increase in the diameter of non-tethered SVs upon short stimulation, while all other diameters remained unchanged (*Figure 7Ciii*). Further, we investigated the SV diameter distribution in more detail by sorting all MP-SVs into different bins. We examined small SVs 40 nm in diameter as well as large SVs ≤60 nm, and the frequency distribution between 45 and 60 in 5 nm steps. There were no obvious shifts in the frequency distributions of the SV diameters (*Figure 7Cii* and *Figure 7—figure supplement 1*). In conclusion, homotypic SV fusion events do not seem to take place among MP-SVs of the IHC synapse under our stimulation paradigms.

Stimulation might have effects on the SV diameters or SV numbers at the ribbon due to potential SV mobilization, SV reformation, or homotypic fusion events. Therefore, we determined RA-SV diameters, as well as SV counts at the distal and the proximal ribbon halves, excluding the MP-SVs (*Chakrabarti et al., 2018*; *Chapochnikov et al., 2014*; *Figure 7—figure supplement 2*). We found no changes in the SV counts (*Figure 7—figure supplement 2*) but observed a significant reduction of SV diameters at both ribbon halves upon stimulation (*Figure 7—figure supplement 2*). Furthermore, we classified the RA-SVs in sub-pools, as done in *Chakrabarti et al., 2018*. We analyzed the fractions of RA-SVs without filaments, directly ribbon-attached, only interconnected to other SVs, and ribbon-attached plus interconnected SVs. We found a decreased fraction of ribbon-attached SVs upon long stimulation compared to the B6J Light and an increased fraction of interconnected SVs upon short stimulation compared to ChR2 NoLight and ChR2 LongStim (*Figure 7—figure supplement 2*, values can be found in *Supplementary file 2*).

To summarize, we neither found indications for larger SVs at the ribbon nor at the membrane upon stimulation except for the non-tethered SVs of the MP-SV pool. Therefore, our data do not support

**Table 2.** List of membrane-proximal (MP)-synaptic vesicle (SV) parameters showing the mean ± SEM values, *N*, *n*, p-values and the statistical tests applied.

Data are presented as mean ± SEM. Data was tested for significant differences by one-way ANOVA followed by Tukey's test (parametric data) or Kruskal-Wallis (KW) test followed by Dunn's test (non-parametric data). Significant results are indicated with *p<0.05; **p<0.01; ***p<0.001; and ****p<0.0001.

| | | B6J Light | ChR2 NoLight | ChR2 ShortStim | ChR2 LongStim | Adjusted p-value | Test |
|---|---|---|---|---|---|---|---|
| $N_{animals}$ | | 2 | 4 | 1 | 4 | | |
| $n_{ribbons}$ Total | | 15 | 17 | 11 | 26 | | |
| $n_{ribbons}$ Ai32VC | | | 9 | 11 | 15 | | |
| $n_{ribbons}$ Ai32KI | | | 8 | 0 | 11 | | |
| $n_{SV}$ Total | | 146 | 253 | 132 | 311 | | |
| $n_{SV}$ Non-tethered | | 38 | 60 | 20 | 75 | | |
| $n_{SV}$ Tethered | | 107 | 191 | 99 | 175 | | |
| $n_{SV}$ Docked | | 1 | 2 | 13 | 61 | | |
| MP-SVs count | | 9.73 ±0.796 | 14.88 ±0.935 | 12.00 ±1.844 | 11.96 ±0.833 | **; B6J Light vs. ChR2 NoLight 0.0028 | KW test – Dunn's test |
| Fraction of non-tethered SVs | | 0.28 ±0.035 | 0.23 ±0.026 | 0.14 ±0.035 | 0.24 ±0.026 | *; B6J Light vs. ChR2 ShortStim 0.0271 | ANOVA – Tukey's test |
| Fraction of tethered SVs | | 0.72 ±0.033 | 0.75 ±0.025 | 0.77 ±0.049 | 0.57 ±0.034 | *; B6J Light vs. ChR2 LongStim 0.0151 | ANOVA – Tukey's test |
| | | | | | | ***; ChR2 LongStim vs. ChR2 NoLight 0.0008 | |
| | | | | | | **; ChR2 LongStim vs. ChR2 ShortStim 0.002 | |
| Fraction of docked SVs | | 0.01 ±0.006 | 0.01 ±0.009 | 0.10 ±0.034 | 0.20 ±0.025 | ****; B6J Light vs. ChR2 LongStim <0.0001 | KW test – Dunn's test |
| | | | | | | ****; ChR2 LongStim vs. ChR2 NoLight <0.0001 | |
| Fraction of tethered | Single PM tethered | 0.25 ±0.052 | 0.30 ±0.039 | 0.18 ±0.055 | 0.25 ±0.036 | n.s. | ANOVA – Tukey's test |
| | Single PD tethered | 0.12 ±0.046 | 0.12 ±0.029 | 0.02 ±0.013 | 0.13 ±0.032 | n.s. | KW test – Dunn's test |
| | Interconnected | 0.02 ±0.01 | 0.02 ±0.009 | 0.04 ±0.023 | 0.05 ±0.029 | n.s. | KW test – Dunn's test |
| | Multiple tethered | 0.61 ±0.068 | 0.56 ±0.051 | 0.76 ±0.048 | 0.56 ±0.046 | n.s. | ANOVA – Tukey's test |

*Table 2 continued on next page*

*Table 2 continued*

| | | B6J Light | ChR2 NoLight | ChR2 ShortStim | ChR2 LongStim | Adjusted p-value | Test |
|---|---|---|---|---|---|---|---|
| | | 22.78 | 23.67 | 19.41 | 19.61 | ***; ChR2 LongStim vs. ChR2 NoLight | |
| | | ±1.08 | ±0.776 | ±1.263 | ±0.876 | 0.0007 | |
| | | | | | | *; ChR2 NoLight vs. ChR2 ShortStim | KW test – Dunn's test |
| | All MP-SVs | | | | | 0.0105 | |
| | | 32.42 | 31.81 | 31.32 | 30.06 | | KW test – Dunn's test |
| | Non-tethered | ±2.138 | ±1.655 | ±3.172 | ±1.481 | n.s. | |
| | | 19.56 | 21.36 | 19.51 | 21.94 | | KW test – Dunn's test |
| | Tethered | ±1.077 | ±0.789 | ±1.292 | ±0.982 | n.s. | |
| | | 1.44 | 0.00 | 0.34 | 0.09 | Not tested: low *n* of docked SVs in control and NoLight conditions | |
| Distance to the membrane | Docked | ±0 | ±0 | ±0.18 | ±0.053 | | |
| | | 31.88 | 37.61 | 32.96 | 30.13 | ***; ChR2 LongStim vs. ChR2 NoLight | KW test – Dunn's test |
| | All MP-SVs | ±2.144 | ±1.622 | ±2.192 | ±1.509 | 0.0001 | |
| | | 35.49 | 47.09 | 45.34 | 38.07 | | KW test – Dunn's test |
| | Non-tethered | ±4.386 | ±3.321 | ±3.654 | ±3.124 | 0.0502 | |
| | | 30.12 | 34.52 | 34.12 | 31.49 | | KW test – Dunn's test |
| | Tethered | ±2.426 | ±1.808 | ±2.571 | ±1.975 | n.s. | |
| | | 83.27 | 48.46 | 5.06 | 16.43 | Not tested: low *n* of docked SVs in control and NoLight conditions | |
| Distance to the PD | Docked | ±0 | ±35.88 | ±1.892 | ±2.876 | | |
| | | 49.00 | 48.67 | 49.17 | 48.05 | | KW test – Dunn's test |
| | All MP-SVs | ±0.442 | ±0.25 | ±0.352 | ±0.303 | | |
| | | 50.17 | 49.71 | 53.27 | 48.76 | **; B6J Light vs. ChR2 ShortStim | |
| | | ±1.061 | ±0.63 | ±0.835 | ±0.485 | 0.0093 | |
| | | | | | | ***; ChR2 LongStim vs. ChR2 ShortStim | |
| | | | | | | 0.0004 | |
| | | | | | | **; ChR2 NoLight vs. ChR2 ShortStim | KW test – Dunn's test |
| | Non-tethered | | | | | 0.0069 | |
| | | 48.55 | 48.29 | 48.06 | 47.76 | | KW test – Dunn's test |
| | Tethered | ±0.465 | ±0.259 | ±0.359 | ±0.419 | n.s. | |
| | | 53.70 | 53.70 | 51.29 | 47.97 | Not tested: low *n* of docked SVs in control and NoLight conditions | |
| Diameter of SVs | Docked | ±0 | ±0 | ±0.845 | ±0.773 | | |

MVR via compound fusion or cumulative fusion of several SVs to prevail as quantal mechanism of exocytosis at IHC synapses at least under our experimental conditions.

## Discussion

In the current study, we established Opto-HPF with a millisecond range physiological stimulation, followed by FS and ET for structure-function analysis of IHC ribbon synapses. This enabled

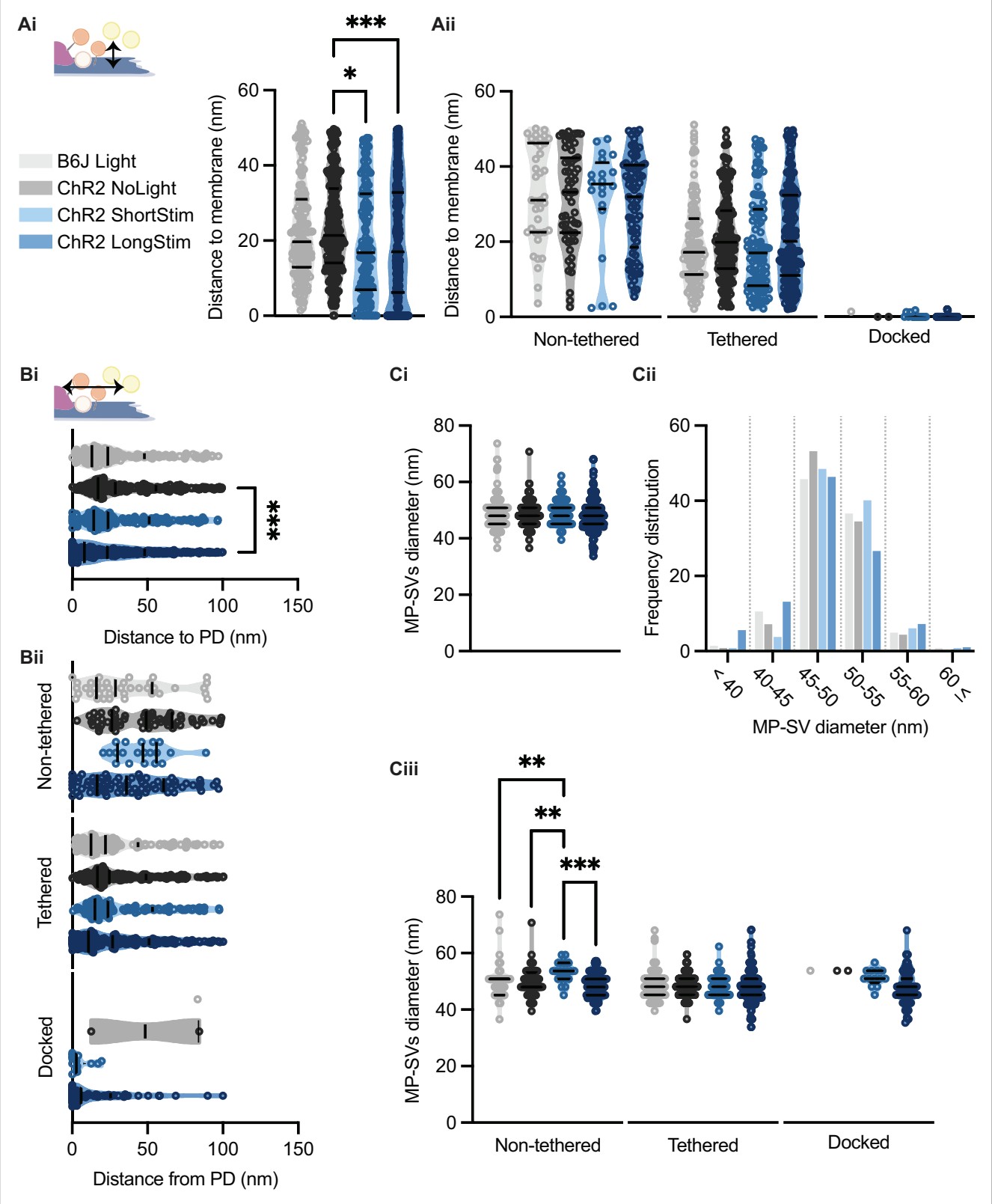

**Figure 7.** Membrane-proximal (MP)-synaptic vesicles (SVs) come closer to the active zone (AZ) membrane and the presynaptic density upon light stimulation. (**Ai**) MP-SVs distance to the AZ membrane. (Aii) Distance of the sub-pools to the AZ membrane. (**Bi**) MP-SVs distance to the PD. (**Bii**) Distance of the sub-pools to the PD. (**Ci**) Diameter of MP-SVs quantified from the outer rim to the outer rim. (**Cii**) Frequency distribution of SV diameter of all MP-SVs. (**Ciii**) SV diameters of the sub-pools. Data are presented in mean ± SEM. *p<0.05, **p<0.01, ***p<0.001, and ****p<0.0001. Statistical test:

*Figure 7 continued on next page*

*Figure 7 continued*

one-way ANOVA followed by Tukey's test (parametric data) and Kruskal-Wallis (KW) test followed by Dunn's test (non-parametric data). B6J Light: $n_{ribbons}$ = 15, $n_{sv}$ = 146, $N_{animals}$ = 2. ChR2 NoLight: $n_{ribbons}$ = 17, $n_{sv}$ = 253, $N_{animals}$ = 4. ChR2 ShortStim: $n_{ribbons}$ = 11, $n_{sv}$ = 132, $N_{animals}$ = 1. ChR2 LongStim: $n_{ribbons}$ = 26, $n_{sv}$ = 311, $N_{animals}$ = 4.

The online version of this article includes the following figure supplement(s) for figure 7:

**Figure supplement 1.** Distances of membrane-proximal (MP)-synaptic vesicles (SVs) to the active zone (AZ) membrane and the presynaptic density as well as their diameters.

**Figure supplement 2.** Analysis of morphologically defined ribbon-associated (RA)-synaptic vesicles (SVs).

near-to-native state preservation of the ultrastructure of exocytic steps occurring within milliseconds and offered a closer correlation to cell-physiological stimulation paradigms widely used in the field. Patch-clamp recordings validated the photoresponses of ChR2-expressing IHCs and demonstrated optogenetically triggered glutamate release. Further, we provide a strategy for precise synchronization of HPF with optogenetic stimulation. In summary (*Figure 8*), our analysis revealed a stimulation-dependent accumulation of docked SVs at IHC AZs. Moreover, we found a slight reduction of the distance of all MP-SVs to the AZ membrane, even more prominent with longer stimulation duration, which is primarily attributed to enhanced docking. Finally, with this near-physiological stimulation, we did not observe large SVs or other morphological correlates of potential homotypic SV fusion events at IHC ribbon synapses.

## Validation of Opto-HPF in IHCs

Using immunofluorescence microscopy, immunogold electron microscopy, and patch-clamp recordings, we observed an efficient and functional expression of ChR2 (H134R) in IHCs of mouse lines that employed Vglut3-dependent Cre expression. Blue light pulses evoked photocurrents and depolarizations similar to previous reports in other cell types (*Boyden et al., 2005*; *Cardin et al., 2010*; *Hernandez et al., 2014*; *Kittelmann et al., 2013*; *Nikolic et al., 2009*). Our patch-clamp recordings of ChR2-expressing IHCs or the postsynaptic bouton revealed that a 10 ms light pulse of 2–11.5 mW/mm$^2$ was sufficient to (i) depolarize the IHC by 50 mV within 10–20 ms and (ii) trigger EPSCs at individual synapses with latencies of 15–20 ms. Longer stimulations of similar irradiance decreased the time to peak of photodepolarization and EPSC latencies and resulted in a sustained release. Part of this sustained release is attributed to the slow repolarization of the IHCs (>40 ms) due to the presence of K$^+$ channel blockers (TEA-Cl and Cs$^+$ in the present study). Based on the charge of the light-evoked EPSCs, the recorded EPSCs most likely reflect the fusion of several SVs at the individual AZ. Under the premise that an individual SV leads to a charge transfer ranging from 50 to 600 fC (*Huang and Moser, 2018*; *Rutherford et al., 2012*), our optogenetic stimulation of IHCs triggers the release of more than 10 SVs on the recorded AZs. This number of released SVs and the presence of an immediate plateau indicates depletion of the RRP even after short and mild light stimulations.

**Table 3.** Comparison of the number of docked synaptic vesicles (SVs) from different studies under different stimulation paradigms.

| Condition/genotype | Publication | Number of tomograms | Average number of docked SVs/AZ | Total number of docked SVs |
|---|---|---|---|---|
| BL6 Light | This study | 15 | 0.07 | 1 |
| ChR2 NoLight | This study | 17 | 0.12 | 2 |
| ChR2 ShortStim | This study | 11 | 1.18 | 13 |
| ChR2 LongStim | This study | 26 | 2.35 | 61 |
| BL6 rest | *Chakrabarti et al., 2018* | 7 | 0.14 | 1 |
| BL6 potassium stim | *Chakrabarti et al., 2018* | 10 | 0.4 | 4 |
| BL6 rest | *Kroll et al., 2020* | 10 | 0.2 | 2 |
| BL6 potassium stim | *Kroll et al., 2020* | 9 | 0.22 | 2 |

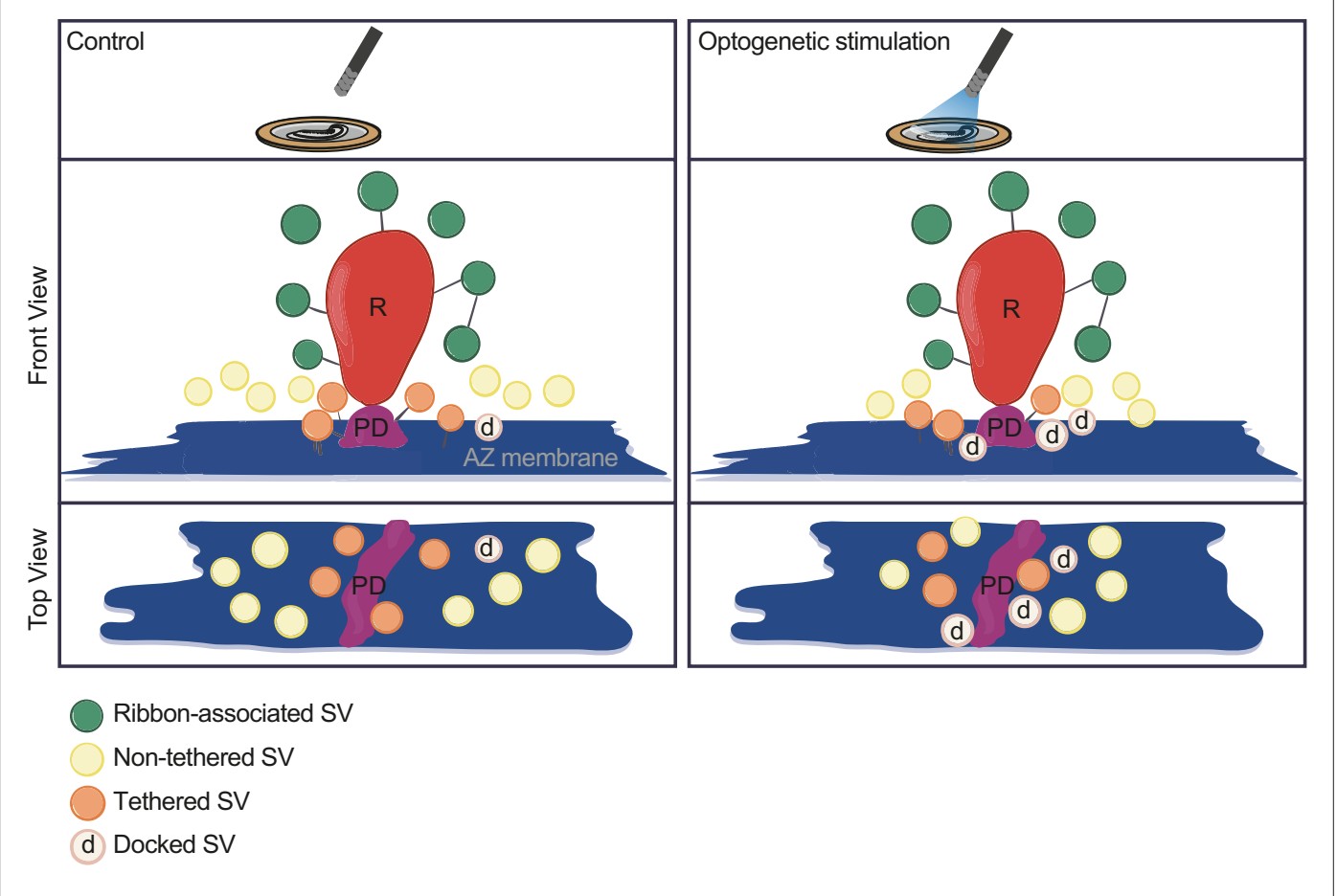

**Figure 8.** Summary. Optogenetic stimulation of inner hair cells (IHCs) recruits synaptic vesicles (SVs) more tightly to the active zone (AZ) membrane and potentially closer to the $Ca^{2+}$ channels. The proportion of docked SVs increased and tethered decreased upon stimulation duration, while the total count of membrane-proximal (MP)-SVs and ribbon-associated (RA)-SVs stayed stable. The distance of MP-SVs to the presynaptic density (PD) slightly decreased upon stimulation.

We note that ChR2, in addition to monovalent cations, also permeates $Ca^{2+}$ ions and poses the question whether optogenetic stimulation of IHCs could trigger release due to direct $Ca^{2+}$ influx via the ChR2. We do not consider such $Ca^{2+}$ influx to trigger exocytosis of SVs in IHCs. Optogenetic stimulation of HEK293 cells overexpressing ChR2 (wildtype version) only raises the intracellular $Ca^{2+}$ concentration up to 90 nM even with an extracellular $Ca^{2+}$ concentration of 90 mM (*Kleinlogel et al., 2011*). IHC exocytosis shows a low $Ca^{2+}$ affinity (~70 μM, *Beutner et al., 2001*) and there is little if any IHC exocytosis for $Ca^{2+}$ concentrations below 10 μM, which is far beyond what could be achieved even by the highly $Ca^{2+}$ permeable ChR2 mutant (CatCh: $Ca^{2+}$ translocating ChR, *Mager et al., 2017*). In addition, we reason that the powerful $Ca^{2+}$ buffering and extrusion by hair cells (e.g., *Frank et al., 2009*; *Issa and Hudspeth, 1996*; *Pangršič et al., 2015*; *Tucker and Fettiplace, 1995*) will efficiently counter $Ca^{2+}$ influx through ChR2 and, thereby limit potential effects on $Ca^{2+}$-dependent replenishment of SVs during ongoing stimulation.

In order to arrive at a reliable Opto-HPF operation when using the HPM100, we added further functionalities to the machine. The pneumatic pressure sensor, which was placed in close proximity to the freezing chamber, accurately enabled us to calculate the time point when pressurized $LN_2$ entered the freezing chamber. This allowed a correlation to the data of internal pressure and temperature sensors, whereas the other sensors provided less reliable signals. To achieve short and long light stimulations in the HPM, we chose a single light pulse with different onset time points. According to the sensor curves, we obtained light stimulations between 17 and 76 ms. Our ShortStim (~20 ms) and LongStim (~50 ms) Opto-HPF paradigms aimed to capture ultrastructural correlates of such phasic

and sustained exocytosis and matched stimulus durations widely used in electrophysiology of hair cell exocytosis (e.g. *Cho et al., 2011*; *Goutman and Glowatzki, 2007*; *Johnson et al., 2017*; *Michalski et al., 2017*; *Moser and Beutner, 2000*; *Parsons et al., 1994*; *Schnee et al., 2005*). IHC patch-clamp indicates that RRP is released within the first 20 ms of step depolarization (*Goutman and Glowatzki, 2007*; *Moser and Beutner, 2000*) while longer depolarizations recruit additional SVs for sustained exocytosis (*Goutman and Glowatzki, 2007*; *Moser and Beutner, 2000*). Furthermore, instead of applying trains of short light pulses as typically used for neuronal cell types (*Berndt et al., 2011*; *Boyden et al., 2005*; *Imig et al., 2020*; *Ishizuka et al., 2006*; *Kleinlogel et al., 2011*; *Lin et al., 2009*), we opted for a continuous light pulse to mimic a step-like receptor potential of IHCs in the high-frequency cochlea of the mouse (*Russell and Sellick, 1978*). The ultrastructural findings upon light stimulation in the HPM undoubtedly reflect snapshots of exocytosis at the IHC synapse.

## Resolving IHCs SV pools with Opto-HPF

Significant efforts have been made to address the mechanisms of SV release at different ribbon synapses by studying morphologically defined SV populations using ET. These studies proposed that the SVs situated close to the AZ membrane represent the 'ultrafast release pool' (*Lenzi and von Gersdorff, 2001*; *Lenzi et al., 1999*), and SVs further away around the ribbon are accessible for slower release (*Lenzi et al., 1999*).

Capacitance measurements (*Beutner and Moser, 2001*; *Johnson et al., 2005*; *Khimich et al., 2005*; *Moser and Beutner, 2000*; *Pangrsic et al., 2010*), fluorescence imaging (*Griesinger et al., 2005*; *Özçete and Moser, 2021*), and recordings from single SGNs (*Buran et al., 2010*; *Frank et al., 2010*; *Goutman and Glowatzki, 2007*; *Jean et al., 2018*; *Jung et al., 2015a*; *Peterson et al., 2014*) propose an RRP with a size between 4 and 45 vesicles per AZ which partially depletes with a time constant of 3–54 ms. The broad range of size and release kinetics estimates results from differences in methods, stimulus paradigms, and experimental conditions as well as in assumptions of model-based data analysis. Moreover, heterogeneity of AZs may also play a role. These physiological estimates of RRP size enclose the number of approximately 10 MP-SVs. Yet, docking of SVs, often considered to be the ultrastructural correlate of fusion competence, is very rare at IHC AZs in non-stimulated conditions (present study, see also *Table 3* and *Chakrabarti et al., 2018*; *Kroll et al., 2020*).

This might reflect spontaneous UVR via a dynamic fusion pore (i.e. 'kiss and run', *Ceccarelli et al., 1979*), which was suggested previously for IHC ribbon synapses (*Chapochnikov et al., 2014*; *Grabner and Moser, 2018*; *Huang and Moser, 2018*; *Takago et al., 2018*) and/or rapid undocking of vesicles (e.g. *Dinkelacker et al., 2000*; *He et al., 2017*; *Nagy et al., 2004*; *Smith et al., 1998*). In the UVR framework, stimulation by ensuing $Ca^{2+}$ influx triggers the statistically independent release of several SVs. Coordinated MVR has been indicated to occur at hair cell synapses (*Glowatzki and Fuchs, 2002*; *Goutman and Glowatzki, 2007*; *Li et al., 2009*) and retinal ribbon synapses (*Hays et al., 2020*; *Mehta et al., 2013*; *Singer et al., 2004*) during both spontaneous and evoked release. We could not observe structures which might hint toward compound or cumulative fusion, neither at the ribbon nor at the AZ membrane under our experimental conditions. Upon short and long stimulation, RA-SVs as well as docked SVs even showed a slightly reduced size compared to controls. However, since some AZs harbored more than one docked SV per AZ in stimulated conditions, we cannot fully exclude the possibility of coordinated release of few SVs upon depolarization.

In contrast to the electrophysiological evidence for a partial RRP depletion, IHC ribbon synapses did not display a significant reduction in MP-SVs upon optogenetic stimulation on the ultrastructural level. Prominent SV docking was reported for retinal ribbon synapses (*Graydon et al., 2011*; *Jackman et al., 2009*), and previously also for auditory ribbon synapses (*Kantardzhieva et al., 2013*; *Pangrsic et al., 2010*). These studies employed chemical fixation, which might alter the number of docked SVs compared to cryo-preserved tissue (*Siksou et al., 2009*). Since other studies on conventional synapses did not report an alteration in the density of docked SVs using either cryo-fixation or aldehyde fixation (*Maus et al., 2020*), some discrepancies might also arise from the different synapse types. In contrast to conventional and retinal ribbon synapses (*Borges-Merjane et al., 2020*; *Imig et al., 2020*; *Watanabe et al., 2013b*), there is a prominent increase in docked SVs when stimulated (present study and *Chakrabarti et al., 2018*; *Kroll et al., 2020*).

Finding an accumulation of docked SVs upon strong depolarization seems puzzling given estimated rates of SV replenishment and subsequent fusion of 180–2000 SV/s at IHC ribbon synapses (***Buran***

*et al., 2010*; *Goutman and Glowatzki, 2007*; *Jean et al., 2018*; *Pangrsic et al., 2010*; *Peterson et al., 2014*; *Schnee et al., 2011*; *Strenzke et al., 2016*). Indeed, such high speed and indefatigable SV release enable firing up to approximately 100 spikes/s in the quiet and steady-state firing of up a few hundred spikes/s upon strong sound stimulation (*Buran et al., 2010*; *Evans, 1972*; *Huet et al., 2016*; *Jean et al., 2018*; *Kiang et al., 1965*; *Liberman and Kiang, 1978*; *Schmiedt, 1989*; *Taberner and Liberman, 2005*). Do these docked SVs represent release ready SVs that are more likely detected upon massive turnover? Do they reflect 'kiss and stay' release events or limited clearance of vesicles following release? Does the rare observation of docked SVs at resting IHC synapses reflect a rapid undocking process?

SV clearance of the AZ (*Neher and Sakaba, 2008*) has been suggested as a potentially rate-limiting mechanism of sustained exocytosis in IHCs of mice with mutations in the genes coding for otoferlin (*Chakrabarti et al., 2018*; *Pangrsic et al., 2010*; *Strenzke et al., 2016*) or endocytic proteins (*Jung et al., 2015a*; *Kroll et al., 2019*; *Kroll et al., 2020*). While our EPSC recordings suggest the ongoing release of neurotransmitter beyond 20 and 50 ms after light onset, limited clearance of the release sites cannot be excluded. The concept implies full collapse fusion followed by clearance of SV proteolipid from the release site for it to engage a new coming SV. Indeed, full collapse fusion followed by clathrin- and dynamin-dependent endocytosis has been indicated for IHCs (*Grabner and Moser, 2018*; *Neef et al., 2014*; *Tertrais et al., 2019*). Yet, unlike for retinal ribbon synapses (*Zampighi et al., 2006*; *Zampighi et al., 2011*), we did not observe omega profiles or hemifusion states of SVs at IHCs AZ opposing the postsynaptic density.

While we have favored the hypothesis that, eventually, fusion pore initiated release typically proceeds to full collapse fusion (*Chapochnikov et al., 2014*), there is support for 'kiss and run' exocytosis (*Alabi and Tsien, 2013*) to occur at IHCs from reports of ultrafast endocytosis (time constant ~300 ms) (*Beutner et al., 2001*; *Neef et al., 2014*; *Tertrais et al., 2019*) and cell-attached capacitance measurements (*Grabner and Moser, 2018*). Could the accumulation of docked SVs during stimulation then represent 'kiss and run' or 'kiss and stay' (*Shin et al., 2018*) release events? Unfortunately, cell-attached membrane capacitance recordings from IHCs did not resolve fusion pores (*Grabner and Moser, 2018*), probably owing to the small SV capacitance (40 aF). Future work including superresolution imaging (*Shin et al., 2018*) and/or Opto-HPF on IHCs with genetic or pharmacological interference might shed light on the existence of a prevalence of 'kiss and run' or 'kiss and stay' at IHC synapses. Freezing times between 5 and 15 ms after the light onset seem necessary to further address this hypothesis.

Finally, a recent study using electrical stimulation and HPF in hippocampal neurons reported full replenishment of docked SVs within 14 ms, but this docking state was only transient, and SVs could potentially undock during the next 100 ms (*Kusick et al., 2020*). Indeed, physiological evidence for reversible priming and docking has been reported for various neurosecretory preparations (e.g. *Dinkelacker et al., 2000*; *He et al., 2017*; *Nagy et al., 2004*; *Smith et al., 1998*). The balance of $Ca^{2+}$ and otoferlin-dependent replenishment of docked and primed SVs with SV fusion and/or undocking/ depriming would then set the 'standing RRP' (*Pangrsic et al., 2010*) and the abundance of docked SVs. Besides increased docking, the decreased distance between SVs and the plasma membrane and PD upon light stimulation supports the previously proposed sequence of events at IHC ribbon synapses (*Chakrabarti et al., 2018*). The sequence for SV release involves tethering and subsequent docking, similarly to what was initially described in conventional synapses using cryo-ET (*Fernández-Busnadiego et al., 2013*). Aside from RIMs, which support SV tethering to the AZ membrane at conventional (*Fernández-Busnadiego et al., 2013*) and IHC ribbon synapses (*Jung et al., 2015b*), otoferlin, (*Pangrsic et al., 2010*; *Vogl et al., 2015*) rather than neuronal SNAREs (*Nouvian et al., 2011*; but see *Safieddine and Wenthold, 1999*) and members of the Munc-13/CAPS families of priming factors (*Vogl et al., 2015*), seems to contribute to preparing SVs for fusion. Despite the distinct molecular composition of the IHC ribbon synapse release machinery, we propose a comparable structural release sequence of SVs. Therefore, reducing the distance to the AZ membrane as well as to the PD might increase the release competence of an SV. $Ca^{2+}$ channels were found to be in close proximity to bassoon, which is the main molecule of the PD (*Wong et al., 2014*). A $Ca^{2+}$ nanodomain-like coupling between $Ca^{2+}$ channels and release sites was reported for mature IHC ribbon synapses (*Brandt et al., 2005*; *Pangršič et al., 2015*; *Wong et al., 2014*). Therefore, a closer distance to the PD would mean a tighter coupling. Such a topography of $Ca^{2+}$ channels and assemblies of the

multidomain AZ proteins RIM2 and bassoon putatively defining SV release sites has recently been reported by MINFLUX nanoscopy for rod photoreceptor ribbon synapses (*Grabner et al., 2022*). Similarly, a reduced distance to the AZ membrane, bringing the SV closer, might deliver the required energy to make an SV finally release competent. However, the mechanism and molecular key players are still not known at IHC ribbon synapses. It will be interesting for future studies to further decipher the underlying molecular machinery at the hair cell synapses and to test whether and to what extent tethering and docking are reversible processes.

## Conclusion

Significant efforts have been made to address the SV exocytosis at ribbon synapses by EM (*Chakrabarti et al., 2018*; *Chapochnikov et al., 2014*; *Lenzi et al., 2002*; *Matthews and Sterling, 2008*; *von Gersdorff et al., 1996*; *Zampighi et al., 2011*). This study offers an experimental approach for structure-function correlation at IHC ribbon synapses. We conclude that activation of ChR2 rapidly depolarizes IHCs and triggers release within few milliseconds in response to brief light flashes. This constitutes a non-invasive approach that overcomes the low temporal resolution of the conventional high K+ depolarization used for electron microscopy of IHC synapses thus far. Combining optogenetic stimulation with high-pressure freezing appears as a promising technique to achieve the temporal and spatial resolution required to study the short-term cellular processes occurring during exocytosis and endocytosis. Notably, we did not observe events that resemble homotypic SV fusion or cumulative fusion close to the AZ membrane of the synaptic ribbon. Further, the RA- as well as the MP-SV pools stayed stable or were rapidly replenished, rather a decrease of the fraction of non-tethered SVs within the MP-SV pool was observed. Finally, the almost complete absence of docked SV in non-stimulated IHCs might reflect the combination of vivid spontaneous release events and fast SV undocking.

# Materials and methods

**Key resources table**

| Reagent type (species) or resource | Designation | Source or reference | Identifiers | Additional information |
|---|---|---|---|---|
| Strain, strain background (*Mus musculus*) | Vglut3-Cre (VC) | *Jung et al., 2015b*; *Vogl et al., 2016* | PMID:26034270 PMID:27458190 | Coding sequence for Cre inserted in start codon of Vglut3/Slc17a8 gene on BAC RP24-88H21 |
| Strain, strain background (*Mus musculus*) | Vglut3-Ires-Cre-KI (KI) | *Lou et al., 2013*; *Vogl et al., 2016* | PMID:23325226; PMID:27458190 | Knock in of ires-Cre cassette 3 |
| Strain, strain background (*Mus musculus*) | B6.Cg-*Gt(ROSA)26Sor*^tm32(CAG-COP4*H134R/EYFP)Hze^/J, common name: Ai32 | *Madisen et al., 2012* | Strain #:024109, RRID: IMSR_JAX:024109 | |
| Strain, strain background (*Mus musculus*) | C57BL/6J | Jackson Laboratory | Strain #: 000664 RRID:IMSR_JAX:000664 | |
| Antibody | Chicken anti-GFP (polyclonal) | Abcam | Cat. #: ab13970; RRID: AB_300798 | IF; Concentrations used: 1:500 |
| Antibody | Rabbit anti-myo6 (polyclonal) | Proteus Biosciences | Cat. #: 25–6791; RRID: AB_10013626 | IF; Concentrations used: 1:200 |
| Antibody | Mouse anti-CtBP2 (monoclonal) | BD Biosciences | Cat. #: 612044; RRID: AB_399431 | IF; Concentrations used: 1:200 |
| Antibody | Mouse anti-neurofilament 200 (monoclonal) | Sigma | Cat. #: N5389; RRID: AB_260781 | IF; Concentrations used: 1:400 |
| Antibody | Rabbit anti-Vglut3 (polyclonal) | SySy | Cat. #: 135 203; RRID: AB_887886 | IF; Concentrations used: 1:300 |
| Antibody | Goat anti-chicken Alexa Fluor 488 (polyclonal) | Invitrogen | Cat. #: A11039; RRID: AB_2534096 | IF; Concentrations used: 1:200 |

*Continued on next page*

*Continued*

| Reagent type (species) or resource | Designation | Source or reference | Identifiers | Additional information |
|---|---|---|---|---|
| Antibody | AbberiorStar 580 goat-conjugated anti-rabbit (polyclonal) | Abberior | Cat. #: 2-0012-005-8; RRID: AB_2810981 | IF; Concentrations used: 1:200 |
| Antibody | AbberiorStar 635p goat-conjugated anti-mouse (polyclonal) | Abberior | Cat. #: 2-0002-007-5; RRID: AB_2893232 | IF; Concentrations used: 1:200 |
| Antibody | Goat anti-mouse Alexa Fluor 647 (polyclonal) | Invitrogen | Cat. #: A-21236; RRID: AB_2535805 | IF; Concentrations used: 1:200 |
| Antibody | Goat anti-rabbit Alexa Fluor 568 (unknown) | Thermo Fisher | RRID: AB_143157 | IF; Concentrations used: 1:200 |
| Sequence-based reagent | Ai32 recombinant_foward | This paper | PCR primers | 5′ – GTGCTGTC TCATCATTTTGGC – 3 |
| Sequence-based reagent | Ai32 recombinant_reverse | This paper | PCR primers | 5′ – TCCATAATCCATGGTGGCAAG – 3 |
| Sequence-based reagent | CGCT recombinant_foward | This paper | PCR primers | 5′ – CTGCTAACCATGTTCATGCC – 3′ |
| Sequence-based reagent | CGCT recombinant_reverse | This paper | PCR primers | 5′ – TTCAGGGTCAGCTTGCCGTA – 3′ |
| Software, algorithm | Patchers Power Tools | Igor Pro XOP (http://www3.mpibpc.mpg.de/groups/neher/index.php?page=software) | RRID: SCR_001950 | |
| Software, algorithm | ImageJ software | ImageJ (http://imagej.nih.gov/ij/) | RRID: SCR_003070 | |
| Software, algorithm | Fiji software | Fiji (http://fiji.sc) | RRID: SCR_002285 | |
| Software, algorithm | Imaris | Oxford Instruments (http://www.bitplane.com/imaris/imaris) | RRID:SCR_007370 | |
| Software, algorithm | Excel | Microsoft (https://microsoft.com/mac/excel) | RRID:SCR_016137 | |
| Software, algorithm | GraphPad Prism software | GraphPad Prism (https://graphpad.com) | RRID:SCR_002798 | |
| Software, algorithm | Igor Pro software package | Wavemetrics (http://www.wavemetrics.com/products/igorpro/igorpro.htm) | RRID: SCR_000325 | |
| Software, algorithm | SerialEM | *Mastronarde, 2005* (https://bio3d.colorado.edu/SerialEM/) | PMID:16182563 | |
| Software, algorithm | 3dmod | *Kremer et al., 1996* (https://bio3d.colorado.edu/imod/doc/3dmodguide.html) | PMID:8742726 | |
| Software, algorithm | IMOD | http://bio3d.colorado.edu/imod | RRID: SCR_003297 | Generating tomograms using the 'etomo' GUI of IMOD |
| Software, algorithm | Gatan Microscopy Suite | http://www.gatan.com/products/tem-analysis/gatan-microscopy-suite-software | RRID: SCR_014492 | Digital micrograph scripting |

*Continued on next page*

*Continued*

| Reagent type (species) or resource | Designation | Source or reference | Identifiers | Additional information |
|---|---|---|---|---|
| Software, algorithm | Patchmaster or Pulse | HEKA Elektronik, ([http://www.heka.com/products/products_main.html#soft_pm](http://www.heka.com/products/products_main.html#soft_pm)) | RRID: SCR_000034 | |
| Software, algorithm | MATLAB | [http://www.mathworks.com/products/matlab/](http://www.mathworks.com/products/matlab/) | RRID: SCR_001622 | |
| Others | IMARIS routine: **Source code 1** | This paper | N/A | To analyze the number of ribbons per IHC. Related to **Figure 1D** |
| Others | IgorPro routine: OptoEPSCs (**Source code 2**) | This paper | N/A | For light-evoked EPSCs. Related to **Figure 4C–F** |
| Others | MATLAB routine: HPMacquire (**Source code 3**) | This paper | N/A | To control light pulse for Opto-HPF in a computer interface. Related to **Figure 3A** |
| Others | MATLAB routine: Intensityprofilecalculator (**Source code 4**) | This paper | N/A | For irradiance analysis. Related to **Figure 3E** |
| Others | MATLAB routine: HPManalyse (**Source code 5**) | This paper | N/A | For sensor data alignment. Related to **Figure 5C** |

## Animals

The mice were bred at the animal facility in the University Medical Center Göttingen (UMG). Animal handling and all experimental procedures were in accordance with the national animal care guidelines issued by the animal welfare committees of the University of Göttingen and the Animal Welfare Office of the State of Lower Saxony (AZ 509.42502/01-27.03).

For expression of ChR2-H134R-EYFP (**Nagel et al., 2003**) in IHCs, we crossbred the B6.Cg-$Gt(ROSA)26Sor^{tm32(CAG-COP4*H134R/EYFP)Hze}$/J mouse line with the common name Ai32 (**Madisen et al., 2012**; RRID:IMSR_JAX:024109) with two different Vglut3-Cre mouse lines, expressing Cre under the control of the *Slc17a8* gene. The ChR2-H134R-EYFP construct is preceded by a STOP codon flanked by loxP sequences such that expression only commences upon Cre recombination ($cre^+/cre^+$ or $cre^+/cre^-$; abbreviated $cre^+$). The first ChR2 *Slc17a8*-driven line, termed Ai32VC, used a previously published transgenic Vglut3-Cre line (**Jung et al., 2015b**; **Vogl et al., 2016**). In the Ai32VC line, animals expressing ChR2 were either *fl/fl cre^+* or *fl/+ cre^+*, which we will abbreviate Ai32VC $cre^+$. For the second ChR2 Vglut3-driven line, termed Ai32KI, we used Vglut3-Ires-Cre-KI mice (**Lou et al., 2013**; **Vogl et al., 2016**). ChR2 expressing animals were either *fl/fl cre^+/cre^-* or *fl/+ cre^+/cre^-*, which we will abbreviate Ai32KI $cre^+$. Littermate controls from both lines (*fl/fl +/+*) are nicknamed WT. C57BL/6J mice are abbreviated 'B6J'.

For immunohistochemistry analysis, three age groups of Ai32KI mice were used. The first group (G1) includes 4- to 5-month-old mice Ai32KI $cre^+$, $N_{animals}$ = 3, $n$=170 cells; and WT, $N_{animals}$ = 2, $n$=99 cells. The second group (G2) corresponds to 6- to 7-month-old mice Ai32KI $cre^+$, $N_{animals}$ = 2, $n$=116 cells; and WT, $N_{animals}$ = 2, $n$=87 cells. The third group (G3) includes 9- to 12-month-old mice Ai32KI $cre^+$, $N_{animals}$ = 2, $n$=129 cells; and WT, $N_{animals}$ = 2, $n$=65cells. For patch-clamp recordings, Ai32VC or Ai32KI mice were used at postnatal days (P) 14–20 (i.e. after the onset of hearing): seven animals were Ai32VC $cre^+$ (*fl/fl*), five animals were Ai32VC $cre^+$ (*fl/+*), and three mice were Ai32KI $cre^+$ (*fl/fl*). For pre-embedding immunogold, we used Ai32KI $cre^+$. For Opto-HPF, Ai32VC $cre^+$, Ai32KI $cre^+$, and B6J mice were used as controls at P14-20 at different stimulation durations (see **Table 4** below).

Some Ai32VC animals showed germline recombination of the EGFP cassette from the CGCT construct (coming from the Vglut3-Cre line) and/or germline recombination of the cassette ChR2-H134R-EYFP due to unspecific Cre recombinase activity. This resulted in the ectopic expression (e.g. in non-IHCs in the cochlea) of EGFP and/or ChR2-H134R-EYFP in the absence of Cre recombinase that could be observed using immunohistochemistry. Additional primers were designed to detect mice with general recombination during genotyping: Ai32 recombinant (forward primer 5' – GTGC

**Table 4.** Genotypes, animal numbers, as well as the ages of the animals used in the experiments.

| Experiment | Genotype | $N_{animals}$ | n (cells/ribbon) | Age |
|---|---|---|---|---|
| Immunohistochemistry | Ai32KI cre$^+$ | 3 | 170 cells | Group 1: 4–5 months |
| | Ai32KI cre$^+$ | 2 | 116 cells | Group 2: 6–7 months |
| | Ai32KI cre$^+$ | 2 | 129 cells | Group 3: 9–12 months |
| Immunohistochemistry | WT | 2 | 99 cells | Group 1: 4–5 months |
| | WT | 2 | 87 cells | Group 2: 6–7 month |
| | WT | 2 | 65 cells | Group 3: 9–12 months |
| Pre-embedding immunogold | Ai32KI cre$^+$ | 1 | | P17 |
| Patch-clamp | Ai32VC cre$^+$(fl/fl) | 7 | 17 cells | P14-17 |
| | Ai32VC cre$^+$(fl/+) | 5 | 5 cells | P14-17 |
| | Ai32KI cre$^+$ | 3 | 21 cells | P16-20 |
| Electron tomography MP-SVs | | | | P14-20 |
| ShortStim | Ai32VC cre$^+$ | 1 | 11 ribbons | |
| LongStim | Ai32VC cre$^+$ | 2 | 15 ribbons | |
| LongStim | B6J | 2 | 15 ribbons | |
| LongStim | Ai32KI cre$^+$ | 2 | 11 ribbons | |
| NoLight | Ai32VC cre$^+$ | 2 | 9 ribbons | |
| NoLight | Ai32KI cre$^+$ | 2 | 8 ribbons | |
| Electron tomography RA-SVs | | | | P14-20 |
| ShortStim | Ai32VC cre$^+$ | 1 | 11 ribbons | |
| LongStim | Ai32VC cre$^+$ | 1 | 10 ribbons | |
| LongStim | B6J | 1 | 9 ribbons | |
| LongStim | Ai32KI cre$^+$ | 2 | 11 ribbons | |
| NoLight | Ai32VC cre$^+$ | 2 | 9 ribbons | |
| NoLight | Ai32KI cre$^+$ | 2 | 8 ribbons | |

TGTC TCATCATTTTGGC – 3', and reverse primer 5' – TCCATAATCCATGGTGGCAAG – 3') and CGCT recombinant (forward primer 5' – CTGCTAACCATGTTCATGCC – 3', and reverse primer 5' – TTCA GGGTCAGCTTGCCGTA – 3'). We excluded animals that showed general recombination for our analysis. The genotype of the animals was determined before the onset of hearing and further corroborated post-mortem.

## Patch-clamp recordings

Perforated patch-clamp recordings from IHCs expressing ChR2-H134R-EYFP were performed as described previously (*Moser and Beutner, 2000*). Briefly, the apical coils of the organ of Corti were dissected from euthanized mice at P14-20 in HEPES Hank's solution containing 5.36 mM KCl, 141.7 mM NaCl, 1 mM MgCl$_2$-6H$_2$O, 0.5 mM MgSO$_4$-7H$_2$O, 10 mM HEPES, 0.5 mg/ml L-glutamine, and 1 mg/ml D-glucose, pH 7.2, osmolarity ~300 mOsm/l. By removing some supporting cells, the basolateral face of the IHCs was exposed and patch-clamp was established using Sylgard-coated 1.5 mm borosilicate pipettes. The intracellular pipette solution contained: 135 mM KCl, 10 mM HEPES, 1 mM MgCl$_2$, and 300 µg/ml amphotericin B (osmolarity ~290 mOsm/l). The organ of Corti was bathed in an extracellular solution containing 126 mM NaCl, 20 mM TEA-Cl, 2.8 mM KCl, 2 mM CaCl$_2$, 1 mM MgCl$_2$, 1 mM CsCl, 10 mM HEPES, and 11.1 mM D-glucose, pH 7.2, osmolarity ~300 mOsm/l. All patch-clamp recordings were performed at room temperature (RT) (20–25°C). An EPC-9 amplifier (HEKA electronics) controlled by Pulse or Patchmaster software (HEKA electronics) was used for the measurements in

current-clamp or voltage-clamp mode. Currents were leak corrected using a p/10 protocol. IHCs with leak currents exceeding –50 pA at –84 mV holding potential or with a series resistance higher than 30 MΩ were excluded from the analysis. A red filter was positioned between the light source and the recording chamber to avoid partial depolarization of the IHCs by the transillumination light.

In order to assess optogenetically evoked IHC exocytosis, postsynaptic recordings from afferent boutons were performed as described previously (*Glowatzki and Fuchs, 2002*; *Huang and Moser, 2018*). Whole-cell patch-clamp recordings from the postsynaptic bouton was established using heat-polished, Sylgard-coated 1 mm thin-glass borosilicate pipettes. The intracellular solution contained 137 mM KCl, 5 mM EGTA, 5 mM HEPES, 1 mM Na$_2$-GTP, 2.5 mM Na$_2$-ATP, 3.5 mM MgCl$_2$·6H$_2$O and 0.1 mM CaCl$_2$, pH 7.2, and osmolarity of ~290 mOsm/l. Boutons with leak currents exceeding –100 pA at –94 mV holding potential were excluded from the analysis. The series resistance of the bouton recordings was calculated offline as reported in *Huang and Moser, 2018*. Recordings with a bouton series resistance >80 MΩ were discarded.

## Photostimulation for cell physiology

Photostimulation of IHCs was achieved using a blue 473 nm laser (MBL 473, CNI Optoelectronics). Irradiance and duration of the light pulses were controlled using the EPC-9 amplifier via a custom controller unit, allowing the transformation of a particular voltage to a particular laser power (i.e. photostimulation during 5, 10, or 50 ms from 2 to 5 V with different increasing steps). A FITC filter set was used to direct the stimulating blue light to the sample. Radiant flux (mW) was measured before each experiment with a laser power meter (LaserCheck; Coherent Inc or Solo2 Gentec-eo) placed under the 40× objective lens. The diameter of the illumination spot was estimated using a green fluorescent slide and a stage micrometer, and it was used to calculate the irradiance in mW/mm$^2$. Photocurrents were measured in voltage-clamp mode and the photodepolarization in current-clamp mode. Only the first series of evoked photocurrents and photodepolarizations were analyzed in order to rule out potential changes in the photoresponses due to inactivation of ChR2-H134R-EYFP (*Lin et al., 2009*).

## Immunohistochemistry

Freshly dissected apical turns of the organ of Corti (as described above) were fixed on ice for 60 min with 4% formaldehyde in phosphate-buffered saline (PBS: 137 mM NaCl, 2.7 mM KCl, 8 mM N$_2$HPO$_4$, 0.2 mM KH$_2$PO$_4$), followed by 3×10 min wash with PBS. A blocking step was performed for 1 hr at RT with goat serum dilution buffer (GSDB: 16% goat serum, 450 mM NaCl, 0.3% Triton X-100, and 20 mM phosphate buffer, pH 7.4). Afterward, the samples were incubated overnight in a wet chamber at 4°C with the following GSDB-diluted primary antibodies: chicken anti-GFP (1:500, Abcam, ab13970; RRID:AB_300798), rabbit anti-myo6 (1:200, Proteus Biosciences, 25-6791; RRID:AB_10013626), mouse anti-CtBP2 (1:200, BD Biosciences, 612044; RRID:AB_399431), mouse anti-neurofilament 200 (1:400, Sigma, N5389; RRID:AB_260781), and rabbit anti-Vglut3 (1:300, SySy, 135 203; RRID:AB_887886). After 3×10 min wash with wash buffer (450 mM NaCl, 0.3% Triton X-100, and 20 mM phosphate buffer, pH 7.4), GSDB-diluted secondary antibodies were applied for 1 hr at RT: goat anti-chicken Alexa Fluor 488 (1:200, Invitrogen, A11039; RRID:AB_2534096), AbberiorStar 580 goat-conjugated anti-rabbit (1:200, Abberior, 2-0012-005-8; RRID:AB_2810981), AbberiorStar 635p goat-conjugated anti-mouse (1:200, Abberior, 2-0002-007-5; RRID:AB_2893232), goat anti-mouse Alexa Fluor 647 (1:200, Invitrogen, A-21236; RRID:AB_2535805), and goat anti-rabbit Alexa Fluor 568 (1:200, Thermo Fisher, RRID:AB_143157). A final washing step was done for 3×10 min with wash buffer and, exclusively in Ai32VC samples, for 1×10 min in 5 mM phosphate buffer. The samples were mounted onto glass slides with a drop of mounting medium (Mowiol 4-88, Roth) and covered with glass coverslips.

Confocal images were acquired using an Abberior Instruments Expert Line STED microscope with a 1.4 NA 100× oil immersion objective and with excitation lasers at 488, 561, 594, and 633 nm. Images were processed using the FIJI software (*Schindelin et al., 2012*) and assembled with Adobe Illustrator Software.

## Immunogold pre-embedding

In order to verify the membrane localization of ChR2 within the IHC, we performed pre-embedding immunogold labeling using nanogold (1.4 nm gold)-coupled nanobodies (information see below) for

the line Ai32KI cre[+] (N=1) on a freshly dissected organ of Corti. The labeling was essentially done as described in **Strenzke et al., 2016**, with a few modifications. Samples were fixed in 2% paraformaldehyde with 0.06% glutaraldehyde in 1× piperazine-N,N′-bis(2-ethanesulfonic acid)-EGTA-MgSO$_4$ (PEM) solution (0.1 M PIPES; 2 mM EGTA; 1 mM MgSO$_4$ × 7H$_2$O) for 90 min on ice and subsequently washed twice for 15 min each in 1× PEM at RT. Next, the samples were blocked for 1 hr in 2% bovin serum albumin (BSA)/3% normal horse serum in 0.2% PBS with Triton X-100 detergent (PBST) at RT. The incubation with the anti-GFP 1.4 nanogold-coupled nanobody was performed overnight at 4°C: anti-GFP in PBS with 0.1% PBST 1:100. On the next day, samples were washed three times for 1 hr in PBS at RT and post-fixed for 30 min in 2% glutaraldehyde in PBS at RT. After four washes for 10 min in distilled H$_2$O at RT, silver enhancement was performed for 4 min in the dark using the Nanoprobes Silver enhancement Kit (Nanoprobes, Yaphank, NY, USA). After incubation, the solution was quickly removed and washed twice with water for a few seconds. After removal of the enhancement solution, 4×10 min washing steps in distilled H$_2$O were performed. Subsequently, samples were fixed for 30 min in 2% OsO$_4$ in 0.1 M cacodylate buffer (pH 7.2) and washed 1 hr in distilled H$_2$O. Samples were further washed over night in distilled H$_2$O at 4°C. On the next day, dehydration was performed: 5 min 30%; 5 min 50%; 10 min 70%; 2×10 min 95%; 3×12 min 100%; 1×30 min, 1×1.5 hr 50% pure EtOH, and 50% epoxy resin (Agar100, Plano, Germany) at RT on a shaker. Samples were then incubated overnight in pure epoxy resin at RT on a shaker. On day 4, another incubation step took place in pure epoxy resin for 6 hr on a shaker and finally samples were transferred to embedding molds with fresh epoxy resin for polymerization for 2 days at 70°C.

## Stoichiometric conjugation of an anti-GFP nanobody with 1.4 nm gold particle

Anti-GFP nanobody carrying a single ectopic cysteine at its C-terminus (NanoTag Biotechnologies GmbH, Cat# N0301-1mg) was used for conjugation with 1.4 nm mono-maleimide gold particles (Nanoprobes Inc, Cat# 2020-30 NMOL). The ~30 nmol of anti-GFP nanobody was first reduced using 10 mM tris (2-carboxyethyl)phosphine (TCEP) for 30 min on ice. Next, the excess of TCEP was removed using a NAP-5 gravity column and the nanobody immediately mixed with lyophilized 120 nmol of mono-maleimide 1.4 nm gold particles. The mixture was incubated for 4 hr on ice with sporadic movement. Finally, the excess of unconjugated gold was removed using an Äkta pure 25 FPLC, equipped with a Superdex 75 Increase 10/300 column.

## Sample mounting for Opto-HPF

Dissection and mounting for HPF was performed in the same extracellular solution as used for patch-clamp, containing 126 mM NaCl, 20 mM TEA-Cl, 2.8 mM KCl, 2 mM CaCl$_2$, 1 mM MgCl$_2$, 1 mM CsCl, 10 mM HEPES, and 11.1 mM D-glucose, pH 7.2, osmolarity ~300 mOsm/l. After dissection, the samples (Ai32VC cre[+] or Ai32KI cre[+] [both: *fl/+ cre[+] or fl/fl cre[+]*]) were mounted upside down (**Figure 3B C**) due to the specific insertion mechanism of the HPM100. The sapphire disc of 6 mm Ø and 0.12 mm thickness (Leica Microsystems, Wetzlar, Germany) was placed into a sample holder middle plate with a rim of 0.2 mm (Leica Microsystems, Wetzlar, Germany). Thereafter, the first 6 mm Ø and 0.2 mm thick spacer ring (Leica Microsystems, Wetzlar, Germany) was placed, forming a cavity. The freshly dissected organ of Corti was then placed into this cavity filled with the above-mentioned extracellular solution. The 0.2 mm side of the 6 mm Ø type A aluminum carrier (Leica Microsystems, Wetzlar, Germany) was placed onto the sample firmly. Next, the second spacer ring of the same dimensions as the first one was placed over the carrier, making a 1.02 mm sample enclosure. Finally, the samples were sandwiched between two transparent cartridges (Leica Microsystems, Wetzlar, Germany). The sample sandwich was then flipped 180° during the insertion process allowing the sample to face toward the light source inside the HPM100 (**Figure 3C**).

## Setup for stimulation and freezing relay

The HPM100 (Leica Microsystems, Wetzlar, Germany) is equipped with a trigger box and an optical fiber that reaches the freezing chamber inside the machine (**Figure 3A**). The HPM100 allows immediate initiation of the freezing process after loading the sample on the cartridge mount and pressing the *process* button of the HPM100. However, the company configuration does not provide a precise temporal control of the freezing process onset in correlation to the light stimulation onset and duration.

Therefore, we installed an external control for the blue light-emitting diode (LED) stimulation and subsequent freezing process initiation. This setup had three key functionalities: (i) external control of the *stimulus* onset and duration; (ii) precise control of the time point at which the freezing process initiates by interfacing with the optical trigger box (account for *HPM start* and *HPM delays* from the start); (iii) command relay to the *accelerometer*, the *pneumatic pressure sensor* and the *microphone* to detect the mechanical and acoustical processes within the HPM till the end of the freezing process (*Figure 3A*, *Figure 5*).

To have an external control of the irradiance and duration of the light stimulation used for Opto-HPF (*Figure 3B C*), an *LLS-3* LED blue light (A20955; 473 nm) source (Schott and Moritex) was used for stimulation. *LLS-3* LED setup allowed the distinction between manual (by intensity control knob, which is maintained at 0) and automated intensity control (through *RSS-232* input). The latter was used with 80 mV selected as the command voltage (see calibration of irradiance at sample below). A *PCI 6221* interface card (37 pins, National Instrument, NI) and an *RS-S232* interface were used to communicate the external LED control box to the computer. This allowed controlling of the amplitude and duration of the light pulse via the computer interface (*Source code 3*). A flexible optical fiber (Leica Microsystems, Wetzlar, Germany) transmitted the blue light from the source to the sample inside the freezing chamber of the HPM.

The *START remote port* of the optical trigger box was connected to the HPM100 via the *J3* cable. This allowed us to *START* and *PAUSE* the freezing process externally (either manually or automatically). Light pulse duration could be defined manually or automatically via the computer interface to have different light stimulation durations of the specimen.

## Irradiance measurements for the HPM100

In order to be able to determine the light irradiance that reaches the sample, the inside of the HPM freezing chamber was replicated in an in-house workshop. In this chamber replica, include the sample carriers with the upper half-cylinder, as well as the LED light source. The exact angle and distances of the original machine are fully replicated based on the technical drawing, kindly provided by Leica Microsystems.

The radiant flux ($\Phi_e$) measurement involves two main custom-made components, a mechano-optical arrangement and an optical detector. As an optical detector, we used a combination of a bare photodiode (First Sensor, PS100-6 THD) and an operational amplifier circuit (operational amplifier: Burr Brown, OPA 637). It incorporates negative voltage bias over the photodiode and a low noise setting due to relative strong current feedback. The photodiode was covered with a neutral density (ND) filter (5% transmission) and brought as close as possible to the sample plane (*Figure 3D*, upper panel). The ND filter was necessary to not to drive the circuit into saturation. The whole detector arrangement was calibrated with a laser light source emitting at 488 nm, which corresponds to the center wavelength of the LED source. In more detail, the calibration was performed with an expanded beam that fits well on the active area of the photodiode and an ND filter with calibrated transmission (active area: 10×10 mm$^2$). The radiant flux can be calculated from the detector output voltage ($U$) with the linear equation:

$$\phi_e(U) = 0.67 + 0.64[mW/V] * U[V] * 20$$

The light distribution measurements were performed by imaging scattered excitation light in the sample plane (*Figure 3C*). For imaging, we used two achromates (f=50 mm) in a configuration with a magnification factor of 1:1 and a CCD camera (IDS, UI-3250ML-M-GL) as detector (*Figure 3D*, lower panel). The spatial irradiance distribution (*Figure 3F*) is derived by transferring the gray values of the image (*Figure 3E*; *Source code 4*) into a radiometric magnitude. Here, we determined the intensity of each pixel ($E_e/p$). First, the intensity per gray value ($gv$) increment was calculated by normalizing the sum of the gray values (- background) in the imaged area to the measured radiant flux ($\Phi_e$). By multiplying the $gv$ of each pixel, we determined the intensity per pixel

$$\frac{E_e}{p} = \frac{\Phi_e}{\sum\limits_{x,y=1}^{n,m} gv_{xy}} * gv$$

The radiant flux at the sample was measured to be 37.3 mW with a peak irradiance of 6 mW/mm$^2$ at the center of the chamber (*Figure 3F*).

### Installing additional sensors at the HPM100

In the HPM100, the freezing process initiates directly after loading the sample (for further details, refer to https://www.leica-microsystems.com) but it does not precisely trigger and monitor the freezing process on an absolute millisecond time scale with respect to the externally initiated *HPM start*. The internal pressure and temperature sensors of the HPM100 offer a freezing curve for each sample with precise readout for temperature and pressure development inside the freezing chamber. These values are stored on a USB stick and are available as an excel file. This internal recording only starts when the internal pressure measured in the freezing chamber reaches 65 bar, and, therefore, it does not provide the absolute time elapsed between *HPM start* ($t=0$) and the time point when the sample is frozen. We incorporated three external sensors: (i) an *accelerometer,* (ii) a *microphone,* and (iii) a *pneumatic pressure sensor* (*Figure 3*, *Figure 5*, for description see below). These additional sensors allowed us to calculate the absolute time scale from the *HPM start* till the sample reaches 0°C (assumed as 'frozen') for each individual freezing (*Figure 5*).

### Accelerometer

The accelerometer from Disynet GmbH (Germany) was externally installed under the HPM100 in order to detect vibration caused during the whole process, from sample insertion till pressure release.

### Microphone

A microphone (MKE2, Sennheiser electronic GmbH & Co, Germany) was installed inside the machine close to the freezing chamber in order to detect acoustic signal (noise) changes during the process from sample insertion till pressure release.

### Pneumatic pressure sensor

The pneumatic pressure sensor (pressure sensor type A-10, WIKA, Germany; different from the HPM100 internal pressure sensor sitting inside the freezing chamber) was installed below the pneumatic needle valve, which opens at 7 bar (pneumatic pressure valve) to regulate the liquid nitrogen ($LN_2$) entry in the freezing chamber. The pneumatic pressure sensor detects the pneumatic pressure build-up and reveals the exact time point when the valve opens and the pneumatic pressure drops. This sensor is controlled by the same external control unit that also triggers the start of the HPM and is proven to be the most reliable readout to correlate the freezing of the samples to the stimulation.

The absolute time scale (in ms) was correlated to the typical pressure curve inside the freezing chamber. This curve shows a pressure build-up (critical pressure of 1700 bar), followed by a plateau during freezing, and finalized by a rapid pressure drop. The temperature curve, detected by the internal thermal sensor, on the other hand, shows a steady drop in the temperature (*Figure 5*). These curves obtained from the internal sensors (pressure and temperature inside the freezing chamber) were correlated to the signals obtained from the three external sensors, whereby the pneumatic pressure sensor delivered a clear characteristic signal at the beginning of the pressure build-up (*Figure 5C*; *Source code 5*). The recordings from the external sensors start when *HPM start* ($t=0$) (*Figure 5*).

### Opto-HPF and freezing procedure

When the HPM was ready for freezing (showing 'ready for freezing' on the HPM display), the freezing process was halted externally by pressing *PAUSE* on the trigger box (Leica Microsystems, Wetzlar, Germany). Subsequently, the sample sandwich was mounted as described above (*Figure 3B, C*) and inserted into the HPM. By pressing the *START* (*HPM start*: $t=0$) command (undoing the *PAUSE* command), the process was resumed: the sample was stimulated for the duration chosen with the installed computer interface and the freezing proceeded to completion.

Several factors must be considered to precisely calculate the time point when the sample is frozen after the *HPM start*. This delay, referred as $T_{HPM\ delay\ from\ START}$, is a sum of the delays caused by $LN_2$ compression and entry, mechanical processes inside the machine (e.g. placing the cartridge in the freezing chamber, valve openings), and specimen freezing. It is the total time it requires from initiating the freezing process by pressing START till the time point, when the sample is frozen. A good quality of freezing requires a steep pressure increase and rapid temperature drop inside the freezing chamber. Since these parameters vary for each individual freezing, it is critical to measure them to accurately calculate $T_{HPM\ delay\ from\ START}$. Furthermore, the time required to build up pressure on $LN_2$ and subsequent

valve opening to the freezing chamber strongly relies on mechanical processes inside the machine and it varied for each individual freezing round. According to the HPM100 manual, the time to pressurize the $LN_2$ ($T_{N2\ pressurized}$) is around ~400 ms after *HPM start*. Once $LN_2$ reaches the required pressure, the pneumatic needle valve opens to let $LN_2$ inside the freezing chamber. The external pneumatic pressure sensor detects these changes in pneumatic pressure outside the freezing chamber (**Figure 5A and B**). The recorded pneumatic pressure curve shows a small dip (**Figure 5C**, inset, asterisk) before the final steady increase and sudden drop. This small dip in the pressure reflects the opening of the valve. This pressure dip recorded by the external sensor can be correlated to the pressure built-up recorded by the internal sensor in the freezing chamber. This correlation sets the absolute time axis and determines the duration of the mechanical delays prior to freezing ($T_{mechanics}$). We also account for the time required for the specimen to reach 0°C ($T_{specimen\ at\ 0}$). The exact temperature of the specimen cannot be monitored (**Watanabe et al., 2013b**), as the internal thermal sensor only provides information about the temperature in the freezing chamber. In our HPM100 instrument, the freezing chamber reached 0°C ($T_{chamber\ at\ 0}$) at 5.41±0.26 ms (mean ± SD). This parameter was calculated from the summation of *rise time* and *shift* (*p/T*) from 10 test shots, similar to a previous report (**Watanabe et al., 2013b**). *Rise time* corresponds to the time required for the pressure to reach 2100 bar, while *shift* p/T describes the time required for the temperature to drop below 0°C in relation to the pressure rise. Further delays include the time required for the sapphire disc to cool down ($T_{sapphire\ at\ 0}$) (0.01 ms, as estimated in **Watanabe et al., 2013b**) and for the sample center to reach 0°C ($T_{sample\ center\ at\ 0}$) (1.1 ms, as estimated in **Watanabe et al., 2013b**). All together, the specimen reaches 0°C in approximately 6.52 ms ($T_{specimen\ at\ 0} = T_{chamber\ at\ 0} + T_{sapphire\ at\ 0} + T_{sample\ center\ at\ 0}$). This time might be an overestimation since we assume that the sample does not cool during the first ms after the chamber is filled with $LN_2$, as stated previously (**Watanabe et al., 2013b**).

Overall, we estimated the delay from *HPM start* as follows:

$T_{HPM\ delay\ from\ START}$
$= T_{N2\ pressurized} + T_{mechanics} + T_{specimen\ at\ 0}$
$= T_{N2\ pressurized} + T_{mechanics} + (T_{chamber\ at\ 0} + T_{sapphire\ at\ 0} + T_{sample\ center\ at\ 0}) = 400 +$ **individually determined per freezing round** $+ \sim 5.41 + 0.01 + 1.1$ (±0.26 ms due to the variability of $T_{chamber\ at\ 0}$).
where $T_{N2\ pressurized} = 400$ ms, and $T_{specimen\ at\ 0} = 5.41$ (±0.26) + 0.01 + 1.1 ms.
$T_{mechanics}$ ranged between 25.4 and 41.6 ms and were individually determined for each freezing round.
The $T_{HPM\ delay\ from\ START}$ for our experiments ranged from 431.92 to 448.12 ms. The onset (StimStart) of a 100 ms light stimulus was set after 390 and 425 ms from *HPM start*. We subtracted Stim-Start from $T_{HPM\ delay\ from\ START}$ to obtain the actual stimulation duration until the sample was frozen for each freezing round (**Figure 5D**).
$\text{Stim} = T_{HPM\ delay\ from\ START} - \text{StimStart}$
$\text{ShortStim} = HPM_{delay\ from\ START} - 425$ ms
$\text{LongStim} = HPM_{delay\ from\ START} - 390$ ms

The light stimulation duration ranged between 17 and 25 ms for ShortStim and 48–76 ms LongStim.

## Sample processing via FS, ultrathin sectioning, and post-staining

FS was performed in an EM AFS2 (Leica Microsystems, Wetzlar, Germany) according to published work (**Chapochnikov et al., 2014**; **Jung et al., 2015b**; **Siksou et al., 2007**; **Vogl et al., 2015**). Briefly, the samples were incubated for 4 days in 0.1% tannic acid in acetone at –90°C. Three washing steps with acetone (1 hr each) were performed at –90°C. Then, 2% osmium tetroxide in acetone was applied to the sample and incubated at –90°C for 7 hr. The temperature was raised to –20°C (5 °C/hr increment) for 14 hr in the same solution. The samples were then incubated at –20°C for 17 hr in the same solution. The temperature was further increased automatically from –20°C to 4°C for 2.4 hr (10 °C/hr increment). When the temperature reached 4°C, the samples were washed in acetone three times (1 hr each) and brought to RT by placing them under the fume hood. Finally, the samples were infiltrated in epoxy resin (Agar 100, Plano, Germany). The next day, the samples were embedded in fresh 100% epoxy resin and polymerized at 70°C for 48 hr in flat embedding molds.

After trimming, 70 nm ultrathin sections were obtained using a UC7 ultramicrotome (Leica Microsystems, Wetzlar, Germany) with a 35° diamond knife (DiAtome, Switzerland). These 70 nm sections were

used to check the freezing quality and find the region of interest (ROI). Furthermore, 70 nm sections were also used to analyze the samples from pre-embedding immunogold labeling.

For ET, semithin 250 nm sections were obtained. The embedded organ of Corti was approached tangentially from the pillar side to ensure that a row of several IHCs was obtained per section. For each sample, at least 20–30 grids with approximately four to six sections of 250 nm were prepared. Post-staining was performed with 4% uranyl acetate in water or uranyl acetate replacement solution (Science Services, EMS) for 40 min and briefly (<1 min) with Reynold's lead citrate in a closed staining compartment in the presence of NaOH to exclude atmospheric $CO_2$ and avoid lead precipitates. After that, grids were washed two times on water droplets with previously boiled and cooled distilled water.

## Transmission electron microscopy and ET

2D electron micrographs were taken from the 70 nm ultrathin sections at 80 kV using a JEM1011 TEM (JEOL, Freising, Germany) equipped with a Gatan Orius 1200A camera (Gatan, Munich, Germany).

We randomly sampled ribbon AZs in the 250 nm sections used for tomograms. This resulted in differences in the cutting plane of the ribbon synapses represented by either a cross-section, a longitudinal section, or a section in between both (*Supplementary file 3*). The grids were systematically screened for ribbons that could be used for ET (0–5 ribbons per grid, often none, mostly 1 or 2) to ensure that ribbon synapses from different IHCs and locations within IHCs were studied. It is difficult to determine the exact number of analyzed IHCs since we did not consistently acquire serial sections. Therefore, it is not possible for us to determine where the analyzed ribbon synapses were located within the IHC.

ET was performed as described previously (*Chakrabarti et al., 2018*; *Wong et al., 2014*). Briefly, 10 nm gold beads (British Bio Cell/Plano, Germany) were applied to both sides of the stained grids. For 3D, single-tilt series were acquired at 200 kV using a JEM2100 TEM (JEOL, Freising, Germany) mostly from −60° to +60° with a 1° increment at 12,000× using the Serial-EM software package (*Mastronarde, 2005*) with a Gatan Orius 1200A camera (Gatan, Munich, Germany). We typically acquired 140–170 virtual sections for each tomogram, but the exact number can vary (*Supplementary file 4*). Tomograms were generated using the IMOD GUI etomo (*Kremer et al., 1996*).

After tomogram generation, we screened for tomograms suitable for further analysis, whereby samples with poor tissue integrity and heavy freezing artifacts at the AZ were excluded. These freezing artifacts were identified by the formation of long filamentous artifacts in the cytoplasm, the nucleus, and specifically, in the vicinity of the IHC ribbon synapses. Notably, the organ of Corti is a sensitive tissue, difficulties for the freezing quality arise from the liquid-filled tunnel of Corti, which is in close proximity to the IHCs. Therefore, for our analysis, we only selected tomograms with a continuous presynaptic AZ, a synaptic cleft and round-shaped SVs.

## Model rendering and image analysis

Tomograms were segmented semi-automatically using 3dmod (*Kremer et al., 1996*) with a pixel size of 1.18 nm at 12,000x magnification. The presynaptic AZ membrane was defined by the area occupied by a clear postsynaptic density as well as a regular synaptic cleft. The AZ membrane of a ribbon synapse was then assigned as a *closed* object and manually segmented every 15 virtual sections for five consecutive virtual sections and then interpolated across the Z-stack. The synaptic ribbons and the PD were also assigned as *closed* objects and were manually segmented for the first 10, middle 20, and last 10 virtual sections and then interpolated across the Z-stack using the interpolator tool of 3dmod. Interpolation was corrected manually in each virtual section thereafter.

MP-SVs were defined as the vesicles localized in the first row from the AZ-membrane, with a maximum 50 nm distance measured vertically from the vesicle's outer layer to the AZ membrane (*Figure 6—figure supplement 1*, upper panel) and with a maximum lateral distance of 100 nm to the PD (*Figure 6—figure supplement 1*, lower panel) (*Chakrabarti et al., 2018*; *Jung et al., 2015b*). Docked SVs were assigned as SVs in 0–2 nm distance to the AZ membrane, as previously done for HPF/FS tomograms of synapses (*Chakrabarti et al., 2018*; *Imig et al., 2014*). RA-SVs were defined as the first row of SVs with a maximal distance of 80 nm from the ribbon surface to the vesicle membrane in each tomogram (*Figure 6—figure supplement 1*, upper panel), excluding the SVs already in the MP-pool (*Chakrabarti et al., 2018*). The distances of the SVs to individual structures like the AZ membrane, PD, and ribbon were measured using the *Measure*

drawing tool in 3dmod at the maximum projection of the vesicle in the tomogram as done previously (*Chakrabarti et al., 2018*). Further, as previously shown in *Chapochnikov et al., 2014*, and *Chakrabarti et al., 2018*, the ribbons were divided into two halves, the distal and the proximal half (excluding the MP-SVs as well as the PD), for analysis of SV counts and diameters (*Figure 7*). CC structures (CCV or CC invagination) were identified 0–500 nm distant to the PD. CCVs were defined as round vesicles with a clear CC, while invaginations were infoldings also clearly decorated with a CC.

All regular vesicles (also including CCVs) were annotated using a spherical scattered object at its maximum projection of the vesicle in the tomogram, marking the outer leaflet (for CCV: outer leaflet, excluding the CC) of the vesicles. Due to excellent ultrastructural preservation obtained from HPF/FS, most of the regular vesicles (also including CCVs) were spherical and were annotated using a spherical *Scattered* object at its maximum projection of the vesicle in the tomogram (*Chakrabarti et al., 2018*), marking the outer leaflet (for CCV: outer leaflet, excluding the CC) of the vesicles. The diameter of the sphere was manually adjusted for each vesicle to rule out any possible errors with the cutting plane or compression. The vesicle radius ($r$) was determined automatically (*Helmprobst et al., 2015*) using the program imodinfo option –p of IMOD software package (*Kremer et al., 1996*). Then the diameter ($D$) was computed with $D=2r$. CC structures that did not appear spherical were segmented manually as closed objects by taking the diameter of the longest and shortest axis in the virtual section showing the maximum projection. The average out of these measurements was taken as the diameter. All outputs were obtained in nm.

## Data analysis

Electrophysiological data were analyzed using the IgorPro 6 software package (Wavemetrics; RRID:SCR_000325), Patchers Power Tools (RRID:SCR_001950), and a custom-written script (*Source code 2*). Evoked photocurrents and photodepolarizations were estimated from the peak of current and depolarization, respectively, following the light pulse. Time to the peak was calculated from the onset of the light stimulus to the peak of the photodepolarization. EPSCs amplitude and charge were computed from the onset of the light pulse until the end of the release. EPSCs latency was calculated from light-pulse$_{onset}$ until EPSC$_{onset}$ (corresponding to avg. baseline ±4 SD) and the time of return to baseline was estimated by EPSC$_{offset}$-EPSC$_{onset}$.

Confocal sections were visualized using the FIJI software (*Schindelin et al., 2012*; RRID:SCR_002285). The analysis of ribbon numbers using the *z*-projections of the stacks was performed in the IMARIS software using custom plug-ins (*Source code 1*) of IMARIS (RRID:SCR_007370), whereby the number of ribbons within an ROI was obtained using the *Spots function*. The average ribbons per IHC was calculated by dividing the number of spots detected by the number of IHCs for each ROI.

For each data set, sample sizes ($n$) were decided according to typical samples sizes in the respective fields (e.g., electrophysiology, ET, and those used for confocal imaging of cochlear IHCs ribbon synapses). Respective sections, figure legends, and tables report the $n$, number of replicates, animal numbers ($N_{animals}$), and the statistical test used.

Model rendering and image analysis criteria for ET are described in the previous section. For statistics, the average SV count and the fraction of SVs in the RA- and MP-pool were calculated per ribbon ($n_{ribbons}$; *Figure 6* and *Figure 6—figure supplement 1*). The average RA- and MP-SV diameters and the average distance of MP-SVs to the AZ membrane and PD were calculated per SV ($n_{SV}$; *Figure 7*, *Figure 7—figure supplement 1*, and *Figure 7—figure supplement 2*).

All data sets were tested for normal distribution (Saphiro-Wilk test) and equality of variances (Brown-Forsythe test). For data sets following a normal distribution and with equality of variances, we used parametric statistical tests (one-way ANOVA and two-way ANOVA), followed by a post hoc test for multiple comparisons (Tukey's test). For non-parametric data sets, we performed Kruskal-Wallis tests, followed by Dunn's test.

For the SV diameter quantification (*Figure 7Cii* and *Figure 7—figure supplement 1C*), we also categorized SV diameters into bins similar to previous studies (*Chakrabarti et al., 2018*; *Hintze et al., 2021*). All statistical analyses and graphs were done using IGOR Pro software 6, GraphPad Prism (RRID:SCR_002798) version 9, and/or R software (version 4.0.3).

## Acknowledgements

We thank AJ Goldak, S Langer, S Gerke, and C Senger-Freitag for expert technical assistance. We would like to thank P Wenig for the technical support in establishing the HPM setup. We would like to thank T Mager for helpful discussion on the irradiance and the company Leica for support with the sensors. Funding: This work was funded by grants of the German Research Foundation through the collaborative research center 889 (projects A02 to TM, A07 to CW, B09 to TP), the collaborative research center 1286 (A04 to CW, B05 to TM, and Z04 to FO), the Leibniz program (to TM), Niedersächsisches Vorab (TM) and EXC 2067/1- 390729940 (MBExC to TM) and Erwin Neher Fellowship to LMJT. LMJT received an Erasmus Mundus scholarship – Neurasmus during part of this work. In addition, this research was supported by Fondation Pour l'Audition (FPA RD-2020-10) to TM.

## Additional information

### Competing interests

Felipe Opazo: is a shareholder of Nanotag Biotechnologies GmbH. The other authors declare that no competing interests exist.

### Funding

| Funder | Grant reference number | Author |
| --- | --- | --- |
| Deutsche Forschungsgemeinschaft | CRC 889 Project A02 | Tobias Moser |
| Deutsche Forschungsgemeinschaft | CRC 889 Project A07 | Carolin Wichmann |
| Deutsche Forschungsgemeinschaft | CRC 1286 Project A04 | Carolin Wichmann |
| Deutsche Forschungsgemeinschaft | CRC 1286 Project B05 | Tobias Moser |
| Deutsche Forschungsgemeinschaft | CRC 1286 Project Z04 | Felipe Opazo |
| Leibniz Program | Leibniz Prize | Tobias Moser |
| Niedersächsisches Ministerium für Wissenschaft und Kultur | Niedersächsisches Vorab | Tobias Moser |
| European Commission | Erasmus Mundus Neurasmus Scholarship | Lina María Jaime Tobón |
| Fondation Pour l'Audition | FPA RD-2020-10 | Tobias Moser |
| Multiscale Bioimaging (MBExC) is a Cluster of Excellence of the University of Göttingen, Germany | EXC 2067/1- 390729940 | Tobias Moser |
| Max Planck Institute for Multidisciplinary Sciences | Erwin Neher Fellowship | Lina María Jaime Tobón |
| Deutsche Forschungsgemeinschaft | CRC 889 Project B09 | Tina Pangrsic |

The funders had no role in study design, data collection and interpretation, or the decision to submit the work for publication.

### Author contributions

Rituparna Chakrabarti, Conceptualization, Formal analysis, Investigation, Methodology, Supervision, Writing – original draft, Writing – review and editing, established and performed Opto-HPF, electron tomography, analysis of the ultrastructural data, performed pre-embedding immunogold, supervised Loujin Slitin, Marina Slashcheva and Elisabeth Fritsch; Lina María Jaime Tobón, Conceptualization,

Formal analysis, Funding acquisition, Investigation, Visualization, Writing – original draft, Writing – review and editing, performed all cell physiology and according analysis and contributed to the immunohistochemistry, prepared figures; Loujin Slitin, Formal analysis, Investigation, Visualization, Writing – review and editing, performed Opto-HPF, electron tomography, analysis of the ultrastructural data and prepared figures; Magdalena Redondo Canales, Formal analysis, Investigation, Visualization, Writing – review and editing, performed immunofluorescence, analysis of ribbon numbers, helped in pre-embedding immunogold analysis and Opto-setup and prepared figures; Gerhard Hoch, Methodology, Resources, Software, Writing – review and editing, programmed the MATLAB GUI, MATLAB interface, installed the sensors to the Opto-HPF; Marina Slashcheva, Formal analysis, Investigation, Writing – review and editing, performed part of the ultrastructural analysis and electron tomography; Elisabeth Fritsch, Formal analysis, Investigation, Writing – review and editing, performed part of the ultrastructural analysis and electron tomography; Kai Bodensiek, Methodology, Resources, Software, Writing – review and editing, established and performed the irradiance measurement; Özge Demet Özçete, Methodology, Resources, Writing – review and editing, designed primers for genotyping; Mehmet Gültas, Formal analysis, Writing – review and editing, performed the statistical analysis for the SV diameters; Susann Michanski, Formal analysis, Investigation, Writing – review and editing, supported HPF experiments and statistical analysis of the SV diameters; Felipe Opazo, Funding acquisition, Methodology, Resources, Writing – review and editing, developed nanogold coupled nanobodies and helped to design the immunogold labeling protocol; Jakob Neef, Investigation, Supervision, Writing – review and editing, supervised Lina María Jaime Tobón for cell physiology and contributed to immunostainings; Tina Pangrsic, Funding acquisition, Methodology, Resources, Writing – review and editing, contributed to the KI line; Tobias Moser, Conceptualization, Funding acquisition, Supervision, Project administration, Writing – original draft, Writing – review and editing, designed the study and supervised Lina María Jaime Tobón for cell physiology and Özge Demet Özçete for the the molecular biology work; Carolin Wichmann, Conceptualization, Funding acquisition, Supervision, Project administration, Visualization, Writing – original draft, Writing – review and editing, designed the study, supervised Rituparna Chakrabarti, Loujin Slitin, Magdalena Redondo Canales, Susann Michanski, Marina Slashcheva and Elisabeth Fritsch for Opto-HPF and immunostainings and prepared figures

### Author ORCIDs
Rituparna Chakrabarti ⓘ http://orcid.org/0000-0001-9481-7553
Lina María Jaime Tobón ⓘ http://orcid.org/0000-0002-6752-7750
Loujin Slitin ⓘ http://orcid.org/0000-0003-4256-5782
Susann Michanski ⓘ http://orcid.org/0000-0001-5893-1981
Felipe Opazo ⓘ http://orcid.org/0000-0002-4968-9713
Jakob Neef ⓘ http://orcid.org/0000-0002-4757-9385
Tobias Moser ⓘ http://orcid.org/0000-0001-7145-0533
Carolin Wichmann ⓘ http://orcid.org/0000-0001-8868-8716

### Ethics
Animal handling and all experimental procedures were in accordance with the national animal care guidelines issued by the animal welfare committees of the University of Göttingen and the Animal Welfare Office of the State of Lower Saxony (AZ 509.42502/01-27.03).

### Decision letter and Author response
Decision letter https://doi.org/10.7554/eLife.79494.sa1
Author response https://doi.org/10.7554/eLife.79494.sa2

---

## Additional files

### Supplementary files
• Supplementary file 1. List of clathrin-coated (CC) structures at the active zone (AZ). Statistics could not be performed due to the rare abundance of CC structures at the AZ in our tomograms. The gray columns indicate CC structures intermingled with the membrane-proximal (MP)-synaptic vesicle (SV) pool.
• Supplementary file 2. List of ribbon-associated (RA)-synaptic vesicle (SV) parameters showing

the mean ± SEM values, N, n, p-values and the statistical tests applied. Data are presented as mean ± SEM. Data was tested for significant differences by one-way ANOVA followed by Tukey's test (parametric data) or Kruskal-Wallis (KW) test followed by Dunn's test (non-parametric data). Significant results are indicated with $*p<0.05$; $**p<0.05$; and $****p<0.0001$.

• Supplementary file 3. Cutting planes of the analyzed ribbon synapses for each condition. For each ribbon, we determined the cutting plane and classified it as either a cross-section, a longitudinal section, or a section in between both.

• Supplementary file 4. The range of virtual sections per condition. To judge the size of the tomograms for each condition, we determined the number (and percentage) of tomograms with virtual sections in the most frequent range of 140–170. This range is usually around 50% for all conditions.

• MDAR checklist

• Source code 1. IMARIS custom plug-ins for the analysis of *Figure 1D*.

• Source code 2. Igor Pro custom-written analysis (OptoEPSCs) of light-evoked excitatory postsynaptic currents (EPSCs) related to *Figure 4C–F*.

• Source code 3. MATLAB scripts (HPMacquire) for the computer interface to control the light pulse for Opto-HPF. Related to *Figure 3A*.

• Source code 4. MATLAB script (Intensityprofilecalculator) for the analysis of the irradiance in *Figure 3E*.

• Source code 5. MATLAB scripts (HPManalyse) for the alignment of the data obtained from the Opto-HPF sensors. Related to *Figure 5C*.

### Data availability

All research materials and biological reagents used in this paper are reported in the Materials and Method section. The custom routines and scripts used in the manuscript are provided as Source Code. The raw data files, including the numerical data associated with the figures, are available on the Open Science Framework DOI https://doi.org/10.17605/OSF.IO/WFJVE.

The following dataset was generated:

| Author(s) | Year | Dataset title | Dataset URL | Database and Identifier |
|---|---|---|---|---|
| Chakrabarti R, Jaime Tobón LM, Slitin L, Redondo-Canales M, Hoch G, Slashcheva M, Fritsch E, Bodensiek K, Özçete ÖD, Gültas M, Michanski S, Opazo F, Neef J, Pangrsic T, Moser T, Wichmann C | 2022 | Optogenetics and electron tomography for structure-function analysis of cochlear ribbon synapses | https://osf.io/wfjve/ | Open Science Framework, 10.17605/OSF.IO/WFJVE |

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
