## [Editor Report]

This is a technically compelling study that uses optogenetic methods and rapid flash-and-freeze tissue preservation techniques to provide new insights into how the ribbon synapses of cochlear inner hair cells are able to transmit auditory signals at very high rates. The conclusions of the paper about the mechanisms underlying exocytosis from these synapses are well supported by the data. The findings of this study should be of interest to a broad audience of neurobiologists and sensory physiologists.

---

## [Decision Letter]

**Decision letter after peer review:**

Thank you for submitting your article "Optogenetics and electron tomography for structure-function analysis of cochlear ribbon synapses" for consideration by *eLife*. Your article has been reviewed by 4 peer reviewers, one of who is a member of our Board of Reviewing Editors, and the evaluation has been overseen by Andrew King as the Senior Editor. The reviewers have opted to remain anonymous.

Essential revisions:

(1) The use of 20 mM TEA and 1 mM Cs is a concern since this will probably depolarize quickly the IHC and deplete the docked vesicle pool. I think an important missing control is to record from an afferent fiber in a normal external solution and then quickly perfuse 20 mM TEA and 1 mM Cs on the synapse. Does this lead to EPSC events? Could this explain the lack of docked vesicles? Alternatively, or in addition, the authors can also do a perforated patch on the IHC in current clamp mode and test if 20 mM TEA and 1 mM Cs depolarize the IHC and thus lead to exocytosis.

(2) The previous paper by Chakrabarti et al. (EMBO reports, 2018) shows beautiful ET images of docked vesicles and RA vesicles at mouse IHC ribbons in both resting and high K-stimulated conditions. It did not use 20 mM TEA and 1 mM Cs. I am thus wondering if the lack of docked vesicles in the current work is due to the use of ChR2, which is Ca permeable, and/or due to the putative depolarization caused by 20 mM TEA and 1 mM Cs? Can the authors please control for this possibility?

(3) The authors give quite long light pulses, which I understand they tried to relate to sustained IHC stimulation. How giving a 50 ms ChR2 light pulse might affect synaptic properties and release should be better discussed. Catching exocytosis omega shapes, including with synchrony of multiple vesicle fusion with fusion pores, is not easy. Even with shorter light-pulses of 5 or 10 ms, omega shapes were rarely observed by multiple people who have used the method due to much faster kinetics of exocytosis, as opposed to ChR stim kinetics plus in combination with potential freezing delay during high-pressure freezing – even a couple of ms makes a difference at this time scale and, depending on the machine, it could go up to 7 ms. I also support the controls proposed above, to better address the lack of docked vesicles they observed, and better discussion/revision of their conclusions.

(4) My request is for a more complete demonstration of the resulting structural data and the methods for analysis because the preparation is novel. The images that were chosen for figure 6 show very large vesicles (panel A – larger than show up in Figure 7Cii), large vacuoles and a narrow synaptic cleft (panel C), a ribbon contacting the membrane (panel D), and a seemingly larger fraction of non-tethered vesicles than reported in Figure 6F. More revealing would be a sequence of images through a single active zone for each control and condition, a demonstration of tethers to the presynaptic density (PD), cell membrane, other vesicles, etc…, and an indication of how distances between the vesicle and cell membrane or PD are measured, and addition of measurements of tether length. These images could be added as supplemental figure(s). The authors have included more images in previous reports, so providing similar information for this preparation should be straightforward.

These demonstrations of data analysis methods should be sufficient to address the resulting variance in results, especially given the differences in the MP population between the two control groups (Figure 6E). The authors should provide an indicator of completeness for each AZ (did all AZs extend through the same imaging depth – the reconstructions suggest this is not the case, were some sampled along the axis of the PD and thereby more complete, etc…). An interesting question is whether the increase in docked vesicle fraction (up to 0.4) is at the expense of tethered or non-tethered vesicles, so linking these data for individual synaptic sites in a supplemental figure, at least for selected AZs, would be revealing. I would like to know also if the distances of tethered vesicles to the membrane (Figure 7A, B) changed with long stimulation when docked vesicles were removed from the calculation (perhaps they were, but Figure 7 suggests otherwise). Measurement of tether lengths would add insight to this question. The authors should also comment on the significance of a 3-4nm shift in vesicle position, and to what extent such a small number (less than the thickness of a lipid bilayer) means functionally.

And, I'll add a couple of requests for clarification of statistical analyses. Data in tables are presented as mean {plus minus} standard error. Which number (N or n) was used in this calculation? It is understandable that the number of animals is small, given the difficulty of the experimental preparation. The reader should know if synaptic sites were sampled from the same hair cell in each animal, or if multiple hair cells were sampled, so a report of these numbers would be valuable.

*Reviewer #1 (Recommendations for the authors):*

(1) The use of 20 mM TEA and 1 mM Cs is a concern since I suppose this will depolarize quickly the IHC and deplete the docked vesicle pool. I think an important missing control is to record from an afferent fiber in normal external solution and then quickly perfuse 20 mM TEA and 1 mM Cs on the synapse. Does this lead to EPSC events? Could this explain the lack of docked vesicles? Alternatively, the authors can also do a perforated patch on the IHC in current clamp mode and test if 20 mM TEA and 1 mM Cs depolarize the IHC and thus lead to exocytosis.

(2) The previous paper by Chakrabarti et al. (EMBO reports, 2018) shows beautiful ET images of docked vesicles and RA vesicles at mouse IHC ribbons in both resting and high K-stimulated conditions. It did not use 20 mM TEA and 1 mM Cs. I am thus wondering if the lack of docked vesicles in the current work is due to the use of ChR2, which is Ca permeable, and/or due to the putative depolarization caused by 20 mM TEA and 1 mM Cs? Can the authors please control for this possibility?

(3) EM and ET at auditory hair cells and retinal ribbon synapses have been performed by other groups and this should be briefly discussed (see e.g. Graydon et al., JNeurosci. 2011; Jackman et al., Nature Neurosci. 2009; Kantardzhieva et al., JCN, 2013). All these papers show evidence for docked vesicles in ribbon synapses. For example, Graydon et al., JNeurosci. 2011, show that docked vesicles in the bottom row of the ribbons can account for the rapid RRP size; Jackman et al., Nature Neurosci. 2009 show a gradient of vesicles along the ribbon in stimulated conditions (see their beautiful Figure 4), and Kantardzhieva et al., JCN, 2013, study docked vesicles near the ribbon of cat IHCs and report from both low and high spontaneous rate fibers and their opposed ribbons.

(4) Evidence for UVR in IHC exocytosis is quite good (Chapochnikov et al., Neuron, 2014), but a recent analysis of evoked EPSC events at IHCs with deconvolution methods could not distinguish between UVR and MVR (see Niwa et al. JNeurophysiol. 25:2021). In addition, evidence for Ca-dependent MVR in turtle and frog auditory hair cells is quite convincing (see Li et al., JNeurosci., 2009; Schnee et al., JNerophysiol., 2013). Likewise, for rod photoreceptor synapses MVR bursts involved vesicles in the readily releasable pool of vesicles and were triggered by the opening of nearby ribbon-associated Ca channels (see Hays et al., JGP, 2020). Rod bipolar cell ribbon synapses also exhibit coordinated MVR (see Mehta et al., Neuron, 2013). These papers should be cited and briefly discussed. Thus, the issue of MVR and UVR as revealed by spontaneous mEPSC events is still under debate at ribbon synapses and in the IHC synapse and requires more investigation. Hence, the great interest in the current paper that uses flash-and-freeze for the first time in the IHC.

*Reviewer #2 (Recommendations for the authors):*

I have a few comments and questions to pose to the authors:

1. While the authors were quite thorough in providing details on the method and taking measurements during experiments – both electrophysiology and electron microscopy, could there be any concern in the different types of optical stimulation used for each? E.g: LED and wide-field illumination used for electron microscopy versus a laser with a 40x objective used for electrophysiology?

2. On line 471-472, the authors say "These sections were used to control for freezing quality and find the region of interest, or to perform pre-embedding immunogold labeling".

Can the authors clarify this? Because under immunogold pre-embedding methods (line 240), they describe pre-embedding methods using fixed-sample preparations with paraformaldehyde. But the use of samples that have been high-pressure frozen, followed by freeze-substitution and embedding for subsequent immunogold labeling undergoes a different procedure of "POST-embedding" immunogold labeling that requires exposure of the epitopes.

3. In line 633 the authors mention using TEA-Cl and Cs+ in the bath for electrophysiology experiments. Were they also used for electron microscopy Opto-HPF experiments? If yes, this should be more clearly stated.

4. Regarding the applied pulse durations of ~20 ms (ShortStim) and ~50 ms (LongStim). The authors describe their use to try to capture phasic and sustained exocytosis and match the step-wise stimulus durations used in IHC exocytosis with electrophysiology. While this makes sense, can the authors describe what the potential consequences could be for IHCs and analysis of exocytosis with ChR2 (and potential calcium entry)?

5. Line 74, small typo on the last word: "OF" not "PF".

*Reviewer #3 (Recommendations for the authors):*

My comments here are methodological. High-resolution views using ET reveal structures not visible using standard TEM. Hence, the sampling of ribbons may be important. Were all ribbon synapses located on a particular region of the inner hair cell membrane? Since location may relate to the spontaneous rate of vesicle release, location could contribute to variance in the data.

Synaptic vesicles were segmented as spheres, but can be distorted and appear oblong due to tissue compression during sectioning (as is visible in Figure 6). How, then, were the diameters of these vesicles estimated, and did they form a large percentage of the total?

The authors indicate that synapses were generally incomplete within the 250 nm tissue thickness, and this feature can lead to erroneous statistical differences, such as between non-stimulated controls. How can the data be normalized, then, or at least a sub-population of fortuitous complete AZs analyzed, to justify differences between control and stimulated animals?

Probably my confusion, but why do the numbers in Table 1 differ from those in Figure 6?

It would be interesting to know if the increased number of docked vesicles was at the expense of tethered or non-tethered vesicles by linking the measures from individual presynaptic densities.

Double-tilt ET provides greater spatial resolution, although at the cost of electron dose and time to collect data. Have the authors experimented with double tilt protocols and, if so, why not employ them?

Ectopic expression of channelrhodopsin was described, but the way this information factored into the experimental design was not clear.

Were distances from vesicle to presynaptic membrane measured only within sections, or in 3D space? Even a moderately tangential section plane relative to the presynaptic membrane can lead to error in this measurement.

Provide a discussion of why two control groups were chosen. In most cases their differences with stimulated groups were similar, except in Figure 6F.

*Reviewer #4 (Recommendations for the authors):*

(1) It would be of interest to look at the size of the docked vesicles independently (rather than just membrane proximal). These would presumably be the population one would want to look at.

(2) I assume the results in figure 3 were also carried out with the same concentrations of Cs+ and TEA. Is that correct? I couldn't find that in the description of those experiments.

(3) line 794: "We found no alterations in the size of the MPSV and RA-SV pools among the various conditions and controls (Figure 6E), except for a larger MP-SV pool in ChR2 NoLight compared to B6J Light. This might reflect different proportions of ribbon-type AZs contained in the tomograms obtained from 250 nm sections that do not always include the full AZs." Why would such an artifact be limited to these controls? Shouldn't we be worried about similar artifacts in other aspects of the data?

---

## [Author Response]

Essential revisions:(1) The use of 20 mM TEA and 1 mM Cs is a concern since this will probably depolarize quickly the IHC and deplete the docked vesicle pool. I think an important missing control is to record from an afferent fiber in a normal external solution and then quickly perfuse 20 mM TEA and 1 mM Cs on the synapse. Does this lead to EPSC events? Could this explain the lack of docked vesicles? Alternatively, or in addition, the authors can also do a perforated patch on the IHC in current clamp mode and test if 20 mM TEA and 1 mM Cs depolarize the IHC and thus lead to exocytosis.

We thank the reviewer for the appreciation of our previous work.

We performed control experiments to address this concern using two different solutions (Author response image 1). The first solution, without TEA and Cs, had the same composition as the resting solution used in Chakrabarti et al., 2018 (5 mM KCl, 136.5 mM NaCl, 1 mM MgCl2, 1.3 mM CaCl_2_, 10 mM HEPES, and 11.1 mM D-glucose, pH 7.2, ~300 mOsmol/l). The second solution, with TEA and Cs, was the same one use in the current study (2.8 mM KCl, 126 mM NaCl, 1 mM MgCl2, 2 mM CaCl_2_, 10 mM HEPES, 20 mM TEA-Cl, 1 mM CsCl, and 11.1 mM D-glucose, pH 7.2, ~300 mOsm/l). We performed perforated patch-clamp recordings of IHCs in current-clamp mode (panel A) to record the IHC potential before and after perfusion of the solutions (panel Ai). Perfusion of the solution with TEA and Cs slightly hyperpolarized the IHC potential by 5.56 ± 1.615 mV (4 cells; p-value = 0.1250, Wilcoxon matched-pairs signed rank test). Additionally, we performed ruptured patch-clamp recordings from two afferent boutons and recorded spontaneous EPSCs before and after perfusion (panel B). EPSCs were detected using Neuromatic (Rothman and Silver, 2018). For the first bouton, the number of EPSCs before perfusion was 68 EPSCs in one minute. After perfusion, there was a slight reduction to 49 EPSCs in one minute. For the second bouton, the number of EPSCs remained constant to 1 EPSC in one minute before and after perfusion of solution with TEA and Cs. Therefore, the rare occurrence of docked SVs in unstimulated conditions is not a reflection of a depolarized IHC. This observation probably reflects the natural state of the IHC ribbon synapse.

**Author response image 1. sa2fig1:** The IHC potential and spontaneous release does not change significantly in the presence of 20 mM TEA and 1mM Cs. (**A**) The IHC potential was measured using perforated patch-clamp recordings in current clamp mode. Two different solutions, without and with 20 mM TEA and 1 mM CsCl, were quickly perfused (>100 mL/h) during the recordings. (**Ai**) The IHC potential was monitored during continuous recordings of 20 s. (**Aii**) There was no significant change in the IHC potential with the two solutions. The presence of TEA and Cs slightly hyperpolarized the IHC potential by 5.56 ± 1.615 mV (4 cells; p-value = 0.1250, Wilcoxon matched-pairs signed rank test). (**B**) Spontaneous EPSCs were recorded from individual afferent boutons using continuous ruptured patch-clamp recordings of 20 s. (**Bi**) We quantified the number of EPSCs occurring during one minute before the start of perfusion and during one minute after the bath solution has been exchanged.

(2) The previous paper by Chakrabarti et al. (EMBO reports, 2018) shows beautiful ET images of docked vesicles and RA vesicles at mouse IHC ribbons in both resting and high K-stimulated conditions. It did not use 20 mM TEA and 1 mM Cs. I am thus wondering if the lack of docked vesicles in the current work is due to the use of ChR2, which is Ca permeable, and/or due to the putative depolarization caused by 20 mM TEA and 1 mM Cs? Can the authors please control for this possibility?

Regarding the ultrastructure and presence/absence of docked SVs of ribbon synapses in IHCs bathed in saline with different ionic composition, we would like to note that we indeed reported very few docked SVs in wild type IHCs at resting conditions without K^+^ channel blockers in Chakrabarti et al. EMBO Rep 2018 and in Kroll et al., 2020, JCS. In both studies, a solution without TEA and Cs was used for the experiments (resting solution Chakrabarti: 5 mM KCl, 136.5 mM NaCl, 1 mM MgCl2, 1.3 mM CaCl_2_, 10 mM HEPES, pH 7.2, 290 mOsmol; control solution Kroll: 5.36 mM KCl, 139.7 mM NaCl, 2 mM CaCl_2_, 1 mM MgCl2, 0.5 mM MgSO_4_, 10 mM HEPES, 3.4 mM L-glutamine, and 6.9 mM D-glucose, pH 7.4). Similarly, our current study shows very few docked SVs in the resting condition even in the presence of TEA and Cs. Based on the results presented in ‘Author response image 1’, we assume that the scarcity of docked SVs under control conditions is not due to depolarization induced by a solution containing 20 mM TEA and 1 mM Cs but is rather representative for the physiological resting state of IHC ribbon synapses. Upon 15 min high potassium depolarization, the number of docked SVs only slightly increased as shown in Chakrabarti et al., 2018 and Kroll et al. 2020, but it was not statistically significant. In the current study, we report a similar phenomenon, but here depolarization resulted in a more robust increase in the number of docked SVs.

To compare the data from the previous studies with the current study, we included an additional table 3 (line 676) now in the discussion with all total counts (and average per AZ) of docked SVs.

The lack of docked vesicles in the current work could be due to the use of ChR2, which is Ca permeable.

In response to the reviewers’ concern, we now discuss the ca^2+^ permeability of ChR2 in addition to the above comparison to our previous studies that demonstrated very few docked SVs in the absence of K^+^ channel blockers and ChR2 expression in IHCs. We are not entirely certain, if the reviewer refers to potential dark currents of ChR2 (e.g. as an explanation for a depletion of docked vesicles under non-stimulated conditions) or to photocurrents, the influx of ca^2+^ through ChR2 itself, and their contribution to ca^2+^ concentration at the active zone.

However, regardless this, we consider it unlikely that a potential contribution of ca^2+^ influx via ChR2 evokes SV fusion at the hair cell active zone.

First of all, we note that the ca^2+^ affinity of IHC exocytosis is very low. As first shown in Beutner et al., 2001 and confirmed thereafter (e.g. Pangrsic et al., 2010), there is little if any IHC exocytosis for ca^2+^ concentrations at the release sites below 10 µM. Two studies using CatCh (a ChR2 mutant with higher ca^2+^ permeability than wildtype ChR2 Kleinlogel et al., 2011; Mager et al., 2017) estimated a max intracellular ca^2+^ increase below 10 µM, even at very negative potentials that promote ca^2+^ influx along the electrochemical gradient or at high extracellular ca^2+^ concentrations of 90 mM. In our experiments, IHCs were depolarized, instead, to values for which extrapolation of the data of Mager et al., 2017 indicate a submicromolar ca^2+^ concentration. In addition, we and others have demonstrated powerful ca^2+^ buffering and extrusion in hair cells (e.g. Tucker and Fettiplace, 1995; Issa and Hudspeth., 1996; Frank et al., 2009 Pangrsic et al., 2015). As a result, the hair cells efficiently clear even massive synaptic ca^2+^ influx and establish a low bulk cytosolic ca^2+^ concentration (Beutner and Moser, 2001; Frank et al., 2009). We reason that these clearance mechanisms efficiently counter any ca^2+^ influx through ChR2. This will likely limit potential effects of ChR2 mediated ca^2+^ influx on ca^2+^ dependent replenishment of synaptic vesicles during ongoing stimulation.

We have now added the following in the discussion (starting in line 620):

“We note that ChR2, in addition to monovalent cations, also permeates ca^2+^ ions and poses the question whether optogenetic stimulation of IHCs could trigger release due to direct ca^2+^ influx via the ChR2. We do not consider such ca^2+^ influx to trigger exocytosis of synaptic vesicles in IHCs. Optogenetic stimulation of HEK293 cells overexpressing ChR2 (wildtype version) only raises the intracellular ca^2+^ concentration up to 90 nM even with an extracellular ca^2+^ concentration of 90 mM (Kleinlogel et al., 2011). IHC exocytosis shows a low ca^2+^ affinity (~70 µM, Beutner et al., 2001) and there is little if any IHC exocytosis for ca^2+^ concentrations below 10 µM, which is far beyond what could be achieved even by the highly ca^2+^ permeable ChR2 mutant (CatCh: ca^2+^ translocating Channelrhodopsin, Mager et al., 2017). In addition, we reason that the powerful ca^2+^ buffering and extrusion by hair cells (e.g., Frank et al., 2009; Issa and Hudspeth, 1996; Pangršič et al., 2015; Tucker and Fettiplace, 1995) will efficiently counter ca^2+^ influx through ChR2 and, thereby limit potential effects on ca^2+^ dependent replenishment of synaptic vesicles during ongoing stimulation.”

(3) The authors give quite long light pulses, which I understand they tried to relate to sustained IHC stimulation. How giving a 50 ms ChR2 light pulse might affect synaptic properties and release should be better discussed.

Please see our reply above.

Catching exocytosis omega shapes, including with synchrony of multiple vesicle fusion with fusion pores, is not easy. Even with shorter light-pulses of 5 or 10 ms, omega shapes were rarely observed by multiple people who have used the method due to much faster kinetics of exocytosis, as opposed to ChR stim kinetics plus in combination with potential freezing delay during high-pressure freezing – even a couple of ms makes a difference at this time scale and, depending on the machine, it could go up to 7 ms. I also support the controls proposed above, to better address the lack of docked vesicles they observed, and better discussion/revision of their conclusions.

We fully agree with the reviewer that omega shapes are in general hard to observe at synapses. As also mentioned in the manuscript, we did not observe any of them. We can also report that we could not see them in data sets from previous publications (Vogl et al., 2015, JCS; Jung et al., 2015, PNAS).

Regarding the lack of docked SVs, please see our reply above. As already pointed out, we had the same observation in previous independent data sets such as from Chakrabarti et al. 2018 EMBO rep and Kroll et al. 2020 JCS, where neither ChR2 nor TEA/Cs were used. Therefore, we conclude that this phenomenon is rather a feature of IHC ribbon synapse biology. However, as mentioned above, we included a discussion on the contribution of ChR2 to ca^2+^ influx/exocytosis.

(4) My request is for a more complete demonstration of the resulting structural data and the methods for analysis because the preparation is novel.

We thank the reviewer for appreciating our data and included more information as pointed out below.

The images that were chosen for figure 6 show very large vesicles (panel A – larger than show up in Figure 7Cii), large vacuoles and a narrow synaptic cleft (panel C), a ribbon contacting the membrane (panel D), and a seemingly larger fraction of non-tethered vesicles than reported in Figure 6F. More revealing would be a sequence of images through a single active zone for each control and condition, a demonstration of tethers to the presynaptic density (PD), cell membrane, other vesicles, etc…, and an indication of how distances between the vesicle and cell membrane or PD are measured, and addition of measurements of tether length. These images could be added as supplemental figure(s). The authors have included more images in previous reports, so providing similar information for this preparation should be straightforward.

Thank you very much for pointing this out. We now included more details in supplemental figures and in the text.

Precisely, we added:

– More details about the morphological sub-pools (analysis and images):

– We now show a sequence of images with different tethering states of membrane proximal SVs together with examples for docked and non-tethered SVs as we did in Chakrabarti et al., 2018 for each condition (Figure 6—figure supplement 2, line 438). Moreover, we included for each condition additional information, we selected further tomograms, one per condition, and depict two additional virtual sections: Figure 6—figure supplement 2.

– Moreover, we present a more detailed quantification for the different morphological sub-pools:

For the MP-SV pool, we analyzed the SV diameters and the distances to the AZ membrane and PD of different SV sub-pools separately, we now included this information in Figure 7

For the RA-SVs, we analyzed in addition the morphological sub-pools and the SV diameters in the distal and the proximal ribbon part as done in Chakrabarti et al. 2018. We now added a new supplement figure (Figure 7—figure supplement 2, line 558 and a supplementary file 2).

– We replaced the virtual section in panel 6D: In the old version, it appeared that the ribbon was contacting the membrane and we realized that this virtual section was not representative: actually, the ribbon was not directly contacting the AZ membrane, a presynaptic density was still visible adjacent to the docked SVs. To avoid potential confusion, we selected a different virtual section of the same tomogram and now indicated the presynaptic density also as graphical aid in Figure 6.

– The focus of the current study was not on the molecular characterization of the tethers like in previous papers, where we presented the tether length (including Vogl et al., 2015, JCS, Chakrabarti et al., 2018, Jung et al., 2015, PNAS). In these studies, we observed an alteration in tether length as a result of the mutation. Also Fernandez-Busnadiego et al. 2013 JCB showed in their study an impact on the tether length due to the lack of RIM1α.

In our previous study on inner hair cell ribbon synapses (Chakrabarti et al. 2018), the average tether lengths in wt at rest were: single tether length 22.7 ± 1.7 nm; multiple tether length 22.6 ± 7.1, not significantly different upon stimulation: single tether length 22.8 ± 2.1 nm, multiple tethers were absent upon high potassium stimulation (from Chakrabarti et al., 2018). This shows a rather uniform tether length, even after stimulation.

Additionally, we used the published dataset from (Chakrabarti et al. 2018, wild type under resting conditions) to clarify the correlation between tether length and SV distance to membrane. We found a significant but rather weak correlation between distance to the AZ membrane and the tether length (Spearman r = 0.4955; p-value = 0.0018; and Author response image 2). For the sake of the present study, we believe that the distance of a synaptic vesicle to the AZ and the PD, and specifically the changes in fractions of docked/tethered and non-tethered SVs, provide a suitable and sufficient account of the steps preceding exocytosis. Therefore, we refrained from measuring the tether length for the current study.

**Author response image 2. sa2fig2:** Correlation between tether length and the distance of a tethered SV to the plasma membrane. Dataset from Chakrabarti et al. 2018, wild type under resting conditions.

Finally, we included a better description in the method section of how the distances to the PD and the AZ membrane were measured (line 1237):

“The distances of the SVs to individual structures like the AZ membrane, PD and ribbon were measured using the Measure drawing tool in 3dmod at the maximum projection of the vesicle in the tomogram as done previously (Chakrabarti et al., 2018).”

These demonstrations of data analysis methods should be sufficient to address the resulting variance in results, especially given the differences in the MP population between the two control groups (Figure 6E). The authors should provide an indicator of completeness for each AZ (did all AZs extend through the same imaging depth – the reconstructions suggest this is not the case, were some sampled along the axis of the PD and thereby more complete, etc…).

We now included more details of our structural data and thank the reviewer for appreciating this structural information. Moreover, we added a new table to the material and method section (Supplementary file 4) to the Materials and methods section, showing the range of virtual sections for each condition. Furthermore, we added a table to the material and method section (Supplementary file 3) that now states, how many ribbons per condition were hit longitudinal, in a cross-section or between these orientations.

Finally, to control for the variations resulting from different ribbon proportions/angles, we focused on assessing the fractions of tethered SVs or docked SVs, rather than their total numbers. Normalization to the AZ area, as it is often done for the quantification of SVs at conventional synapses, is not perfectly suitable for ribbon synapses. The AZ area does not necessarily correlate with the ribbon size and often exceeds the tomogram, as the reviewer also mentioned above. As said, for these reasons we chose to present the fractions as also done in previous studies (Chakrabarti et al., 2018 or Kroll et al., 2020, JCS).

An interesting question is whether the increase in docked vesicle fraction (up to 0.4) is at the expense of tethered or non-tethered vesicles, so linking these data for individual synaptic sites in a supplemental figure, at least for selected AZs, would be revealing.

We thank the reviewer for pointing this out, the larger fraction of docked SVs upon stimulation is on the expense of tethered SVs as shown in Figure 6F. For example, when comparing Figure 6B and D (top views), a clear difference in the proportion of docked/tethered SVs is visible, which we now also mention in the manuscript.

I would like to know also if the distances of tethered vesicles to the membrane (Figure 7A, B) changed with long stimulation when docked vesicles were removed from the calculation (perhaps they were, but Figure 7 suggests otherwise). Measurement of tether lengths would add insight to this question.

This is an important question, the analysis was done with all MP-SVs, including the docked vesicles. However, we now show the distances of all SVs to the AZ membrane and in addition docked, tethered and non-tethered SV distances to the AZ membrane separately. We added the new graph to Figure 7. We now write in the manuscript from line 456ff:

“However, enhanced docking to the AZ membrane mainly accounted for the decrease of average SV distances to the AZ-membrane, as it was no longer significant when the non-tethered and tethered sub-pools were analyzed independently (Figure 7Ai, Aii).”

– Regarding the tether length, we would like to point out that the focus of the current study was not on the molecular characterization of the tethers like in previous papers, where we presented the tether length (including Vogl et al., 2015, JCS, Chakrabarti et al., 2018, Jung et al., 2015, PNAS). In these studies, we observed an alteration in tether length as a result of the mutation. Also Fernandez-Busnadiego et al. 2013 JCB showed in their study an impact on the tether length due to the lack of RIM1α.

In our previous study on inner hair cell ribbon synapses (Chakrabarti et al. 2018), the average tether lengths in wt at rest were: single tether length 22.7 ± 1.7 nm; multiple tether length 22.6 ± 7.1, not significantly different upon stimulation: single tether length 22.8 ± 2.1 nm, multiple tethers were absent upon high potassium stimulation (from Chakrabarti et al., 2018). This shows a rather uniform tether length, even after stimulation.

Additionally, we used the published dataset from (Chakrabarti et al. 2018, wild type under resting conditions) to clarify the correlation between tether length and SV distance to membrane. We found a significant but rather weak correlation between distance to the AZ membrane and the tether length (Spearman r = 0.4955; p-value = 0.0018; and Author response image 2). For the sake of the present study, we believe that the distance of a synaptic vesicle to the AZ and the PD, and specifically the changes in fractions of docked/tethered and non-tethered SVs, provide a suitable and sufficient account of the steps preceding exocytosis. Therefore, we refrained from measuring the tether length for the current study.

The authors should also comment on the significance of a 3-4nm shift in vesicle position, and to what extent such a small number (less than the thickness of a lipid bilayer) means functionally.

We thank the reviewer for pointing this out, we now discuss this topic from line 763 onwards.

“Despite of the distinct molecular composition of the IHC ribbon synapse release machinery, we propose a comparable structural release-sequence of SVs. Therefore, reducing the distance to the AZ membrane as well as to the PD might increase the release competence of a SV. Ca^2+^ channels were found to be in close proximity to bassoon, which is the main molecule of the PD (Wong et al., 2014). A ca^2+^ nanodomain-like coupling between ca^2+^ channels and release sites was reported for mature IHC ribbon synapses (Brandt et al., 2005; Pangršič et al., 2015; Wong et al., 2014). Therefore, a closer distance to the PD would mean a tighter coupling. Such a topography of ca^2+^ channels and assemblies of the multidomain AZ proteins RIM2 and bassoon putatively defining SV release sites has recently been reported by MINFLUX nanoscopy for rod photoreceptor ribbon synapses (Grabner et al., 2022). Similarly, a reduced distance to the AZ membrane, bringing the SV closer might deliver the required energy to make a SV finally release competent. However, the mechanism and molecular key players are still not known at IHC ribbon synapses.”

And, I'll add a couple of requests for clarification of statistical analyses. Data in tables are presented as mean {plus minus} standard error. Which number (N or n) was used in this calculation? It is understandable that the number of animals is small, given the difficulty of the experimental preparation. The reader should know if synaptic sites were sampled from the same hair cell in each animal, or if multiple hair cells were sampled, so a report of these numbers would be valuable.

In order to be transparent on that we show in our plots always the individual data points. For figure 6, figure 6 supplement 1 and figure 7 supplement 7 the analysis is done per ribbon, thus the values included in the legends are n_ribbons_. For figure 7 and figure 7 supplement 1, the analysis was performed for SVs, thus the values are in the legends are n_sv_. We also included these values in table 2 and supplementary file 2.

Moreover, we now provide more details on the sampling procedure in the method part.

We wrote from line 1166 onwards:

“The embedded organ of Corti was approached tangentially from the pillar side to ensure that a row of several IHCs was obtained per section. For each sample, at least 20-30 grids with approximately 4-6 sections of 250 nm were prepared.”

And from line 1180 onwards:

“We randomly sampled ribbon AZs in the 250 nm sections used for tomograms. This resulted in differences in the cutting plane of the ribbon synapses represented by either a cross-section, a longitudinal section, or a section in between both (Supplementary file 3). The grids were systematically screened for ribbons that could be used for ET (0 – 4/5 ribbons per grid, often 0, frequently 1 or 2) to ensure that ribbon synapses from different IHCs and locations within IHCs were studied. It is difficult to determine the exact number of analyzed IHCs since we did not consistently acquire serial sections. Therefore, it is not possible for us to determine where the analyzed ribbon synapses were located within the IHC.”

Reviewer #1 (Recommendations for the authors):(1) The use of 20 mM TEA and 1 mM Cs is a concern since I suppose this will depolarize quickly the IHC and deplete the docked vesicle pool. I think an important missing control is to record from an afferent fiber in normal external solution and then quickly perfuse 20 mM TEA and 1 mM Cs on the synapse. Does this lead to EPSC events? Could this explain the lack of docked vesicles? Alternatively, the authors can also do a perforated patch on the IHC in current clamp mode and test if 20 mM TEA and 1 mM Cs depolarize the IHC and thus lead to exocytosis.

We performed control experiments to address this concern using two different solutions (Author response image 1). The first solution, without TEA and Cs, had the same composition as the resting solution used in Chakrabarti et al., 2018 (5 mM KCl, 136.5 mM NaCl, 1 mM MgCl2, 1.3 mM CaCl_2_, 10 mM HEPES, and 11.1 mM D-glucose, pH 7.2, ~300 mOsmol/l). The second solution, with TEA and Cs, was the same one use in the current study (2.8 mM KCl, 126 mM NaCl, 1 mM MgCl2, 2 mM CaCl_2_, 10 mM HEPES, 20 mM TEA-Cl, 1 mM CsCl, and 11.1 mM D-glucose, pH 7.2, ~300 mOsm/l). We performed perforated patch-clamp recordings of IHCs in current-clamp mode (panel A) to record the IHC potential before and after perfusion of the solutions (panel Ai). Perfusion of the solution with TEA and Cs slightly hyperpolarized the IHC potential by 5.56 ± 1.615 mV (4 cells; p-value = 0.1250, Wilcoxon matched-pairs signed rank test). Additionally, we performed ruptured patch-clamp recordings from two afferent boutons and recorded spontaneous EPSCs before and after perfusion (panel B). EPSCs were detected using Neuromatic (Rothman and Silver, 2018). For the first bouton, the number of EPSCs before perfusion was 68 EPSCs in one minute. After perfusion, there was a slight reduction to 49 EPSCs in one minute. For the second bouton, the number of EPSCs remained constant to 1 EPSC in one minute before and after perfusion of solution with TEA and Cs. Therefore, the rare occurrence of docked SVs in unstimulated conditions is not a reflection of a depolarized IHC. This observation probably reflects the natural state of the IHC ribbon synapse.

(2) The previous paper by Chakrabarti et al. (EMBO reports, 2018) shows beautiful ET images of docked vesicles and RA vesicles at mouse IHC ribbons in both resting and high K-stimulated conditions. It did not use 20 mM TEA and 1 mM Cs. I am thus wondering if the lack of docked vesicles in the current work is due to the use of ChR2, which is Ca permeable, and/or due to the putative depolarization caused by 20 mM TEA and 1 mM Cs? Can the authors please control for this possibility?

Regarding the ultrastructure and presence/absence of docked SVs of ribbon synapses in IHCs bathed in saline with different ionic composition:

We would like to note that we indeed reported very few docked SVs in wild type IHCs at resting conditions without K^+^ channel blockers in Chakrabarti et al. EMBO Rep 2018 and in Kroll et al., 2020, JCS. In both studies, a solution without TEA and Cs was used for the experiments (resting solution Chakrabarti: 5 mM KCl, 136.5 mM NaCl, 1 mM MgCl2, 1.3 mM CaCl_2_, 10 mM HEPES, pH 7.2, 290 mOsmol; control solution Kroll: 5.36 mM KCl, 139.7 mM NaCl, 2 mM CaCl_2_, 1 mM MgCl2, 0.5 mM MgSO_4_, 10 mM HEPES, 3.4 mM L-glutamine, and 6.9 mM D-glucose, pH 7.4). Similarly, our current study shows very few docked SVs in the resting condition even in the presence of TEA and Cs. Based on the results presented in ‘Author response image 1’, we assume that the scarcity of docked SVs under control conditions is not due to depolarization induced by a solution containing 20 mM TEA and 1 mM Cs but is rather representative for the physiological resting state of IHC ribbon synapses. Upon 15 min high potassium depolarization, the number of docked SVs only slightly increased as shown in Chakrabarti et al., 2018 and Kroll et al. 2020, but it was not statistically significant. In the current study, we report a similar phenomenon, but here depolarization resulted in a more robust increase in the number of docked SVs.

To compare the data from the previous studies with the current study, we included an additional table 3 (line 676) now in the discussion with all total counts (and average per AZ) of docked SVs.

(3) EM and ET at auditory hair cells and retinal ribbon synapses have been performed by other groups and this should be briefly discussed (see e.g. Graydon et al., JNeurosci. 2011; Jackman et al., Nature Neurosci. 2009; Kantardzhieva et al., JCN, 2013). All these papers show evidence for docked vesicles in ribbon synapses. For example, Graydon et al., JNeurosci. 2011, show that docked vesicles in the bottom row of the ribbons can account for the rapid RRP size; Jackman et al., Nature Neurosci. 2009 show a gradient of vesicles along the ribbon in stimulated conditions (see their beautiful Figure 4), and Kantardzhieva et al., JCN, 2013, study docked vesicles near the ribbon of cat IHCs and report from both low and high spontaneous rate fibers and their opposed ribbons.

As pointed out above, the observation of only a few docked SVs at IHC active zones under resting conditions was a consistent finding also in previous publications (Chakrabarti et al., 2018, EMBO rep, Kroll et al., 2020, JCS) using HPF/FS in combination with ET. However, we fully agree with the reviewer that our discussion on SV docking and observations made at other ribbon synapse types could be expanded. Therefore, we now included additional literature on morphological vesicles pools of ribbon synapses in the discussion.

It needs to be pointed out that we used HPF/FS instead of chemical fixation. It was shown in previous studies that the immobilization method can influence the number of docked SVs/the SVs distance to the membrane. E.g. Siksou et al. (2009, EJN) showed that in HPF-immobilized samples of Munc13 mutants docked SVs cannot be found, while SV docking was apparent under chemical fixation. However, other studies did not find an altered number of docked SVs in general upon aldehyde fixation compared to cryo-preservation (Maus et al., 2020, Cell Rep.), but differences in SV distribution (Korogod et al., 2015, *eLife*; Maus et al., 2020). So, using HPF/FS might reveal SV distributions at ribbon synapses that differ from those found in studies using chemical fixation. Furthermore, different ribbon synapse types might have different biological needs as well as functional/molecular properties as reviewed in Moser et al., Physiol Rev 2019. We now included these points into our discussion and wrote from line 697 on:

“Prominent SV docking was reported for retinal ribbon synapses (Graydon et al., 2011; Jackman et al., 2009), and previously also for auditory ribbon synapses (Kantardzhieva et al., 2013; Pangrsic et al., 2010). These studies employed chemical fixation, which might alter the number of docked SVs compared to cryo-preserved tissue (Siksou et al., 2009). Since other studies on conventional synapses did not report an alteration in the density of docked SVs using either cryo-fixation or aldehyde fixation (Maus et al., 2020), some discrepancies might also arise from the different synapse types.”

(4) Evidence for UVR in IHC exocytosis is quite good (Chapochnikov et al., Neuron, 2014), but a recent analysis of evoked EPSC events at IHCs with deconvolution methods could not distinguish between UVR and MVR (see Niwa et al. JNeurophysiol. 25:2021). In addition, evidence for Ca-dependent MVR in turtle and frog auditory hair cells is quite convincing (see Li et al., JNeurosci., 2009; Schnee et al., JNerophysiol., 2013). Likewise, for rod photoreceptor synapses MVR bursts involved vesicles in the readily releasable pool of vesicles and were triggered by the opening of nearby ribbon-associated Ca channels (see Hays et al., JGP, 2020). Rod bipolar cell ribbon synapses also exhibit coordinated MVR (see Mehta et al., Neuron, 2013). These papers should be cited and briefly discussed.

In response to this comment, we have expanded this part for a broader discussion of the relevant literature in our discussion. It is important to point out that we do not claim that the evoked release is mediated by one single SV. As discussed in the paper (line 672), we consider that our optogenetic stimulation of IHCs triggers the release of more than 10 SVs per AZ. This falls in line with the previous reports of several SVs fusing upon stimulation. As pointed out by the reviewer, this type of evoked MVR is probably mediated by the opening of ca^2+^ channels in close proximity to each SV ca^2+^ sensor. We indeed sometimes observed more than one docked SV per AZ upon long optogenetic stimulation. This could reflect that possibility. However, given the absence of large structures directly at the ribbon or the AZ membrane that could suggest the compound fusion of several SVs prior or during fusion, we argue against compound MVR release at IHCs.

As mentioned above, we added to the discussion (from line 679 onwards).

We wrote:

“This might reflect spontaneous univesicular release (UVR) via a dynamic fusion pore (i.e. ‘kiss and run’, Ceccarelli et al., 1979), which was suggested previously for IHC ribbon synapses (Chapochnikov et al., 2014; Grabner and Moser, 2018; Huang and Moser, 2018; Takago et al., 2019) and/or and rapid undocking of vesicles (e.g. Dinkelacker et al., 2000; He et al., 2017; Nagy et al., 2004; Smith et al., 1998). In the UVR framework, stimulation by ensuing ca^2+^ influx triggers the statistically independent release of several SVs. Coordinated multivesicular release (MVR) has been indicated to occur at hair cell synapses (Glowatzki and Fuchs, 2002; Goutman and Glowatzki, 2007; Li et al., 2009) and retinal ribbon synapses (Hays et al., 2020; Mehta et al., 2013; Singer et al., 2004) during both spontaneous and evoked release. We could not observe structures which might hint towards compound or cumulative fusion, neither at the ribbon nor at the AZ membrane under our experimental conditions. Upon short and long stimulation, RA-SVs as well as docked SVs even showed a slightly reduced size compared to controls. However, since some AZs harbored more than one docked SV per AZ in stimulated conditions, we cannot fully exclude the possibility of coordinated release of few SVs upon depolarization.”

Thus, the issue of MVR and UVR as revealed by spontaneous mEPSC events is still under debate at ribbon synapses and in the IHC synapse and requires more investigation. Hence, the great interest in the current paper that uses flash-and-freeze for the first time in the IHC.

We thank the reviewer for the appreciation of our work.

Reviewer #2 (Recommendations for the authors):I have a few comments and questions to pose to the authors:1. While the authors were quite thorough in providing details on the method and taking measurements during experiments – both electrophysiology and electron microscopy, could there be any concern in the different types of optical stimulation used for each? E.g: LED and wide-field illumination used for electron microscopy versus a laser with a 40x objective used for electrophysiology?

That is indeed a very important point that we invested in heavily. For making optogenetic stimulation most comparable between the two different set-ups we estimated the irradiance (i.e. mW/mm^2^). So in essence, for properly relating structure and function we aimed for identical irradiances in HPF as we had used for functional characterization. As described in the method part, we built a chamber that mimics the freezing chamber geometry in order to determine the irradiance as it would reach the sample in the freezing machine. This way, we have a comparable readout for both experimental setups.

2. On line 471-472, the authors say "These sections were used to control for freezing quality and find the region of interest, or to perform pre-embedding immunogold labeling".Can the authors clarify this? Because under immunogold pre-embedding methods (line 240), they describe pre-embedding methods using fixed-sample preparations with paraformaldehyde. But the use of samples that have been high-pressure frozen, followed by freeze-substitution and embedding for subsequent immunogold labeling undergoes a different procedure of "POST-embedding" immunogold labeling that requires exposure of the epitopes.

Thank you for pointing this out. This sentence was misleading, we now clarified the statement. It was meant that we used 70 nm ultrathin sections to screen and analyze blocks from both methods, HPF/FS as well as pre-embedding.

We now wrote from line 1163 on:

“These 70 nm sections were used to check the freezing quality and find the region of interest. Furthermore, 70 nm sections were also used to analyze the samples from pre-embedding immunogold labeling.”

3. In line 633 the authors mention using TEA-Cl and Cs+ in the bath for electrophysiology experiments. Were they also used for electron microscopy Opto-HPF experiments? If yes, this should be more clearly stated.

We apologize for not being clear. We now included the recipe also for our opto-HPF method description.

4. Regarding the applied pulse durations of ~20 ms (ShortStim) and ~50 ms (LongStim). The authors describe their use to try to capture phasic and sustained exocytosis and match the step-wise stimulus durations used in IHC exocytosis with electrophysiology. While this makes sense, can the authors describe what the potential consequences could be for IHCs and analysis of exocytosis with ChR2 (and potential calcium entry)?

In response to the reviewers’ concern, we now discuss the ca^2+^ permeability of ChR2 in addition to the above comparison to our previous studies that demonstrated very few docked SVs in the absence of K^+^ channel blockers and ChR2 expression in IHCs. We are not entirely certain, if the reviewer refers to potential dark currents of ChR2 (e.g. as an explanation for a depletion of docked vesicles under non-stimulated conditions) or to photocurrents, the influx of ca^2+^ through ChR2 itself, and their contribution to ca^2+^ concentration at the active zone.

However, regardless this, we consider it unlikely that a potential contribution of ca^2+^ influx via ChR2 evokes SV fusion at the hair cell active zone.

First of all, we note that the ca^2+^ affinity of IHC exocytosis is very low. As first shown in Beutner et al., 2001 and confirmed thereafter (e.g. Pangrsic et al., 2010), there is little if any IHC exocytosis for ca^2+^ concentrations at the release sites below 10 µM. Two studies using CatCh (a ChR2 mutant with higher ca^2+^ permeability than wildtype ChR2 Kleinlogel et al., 2011; Mager et al., 2017) estimated a max intracellular ca^2+^ increase below 10 µM, even at very negative potentials that promote ca^2+^ influx along the electrochemical gradient or at high extracellular ca^2+^ concentrations of 90 mM. In our experiments, IHCs were depolarized, instead, to values for which extrapolation of the data of Mager et al., 2017 indicate a submicromolar ca^2+^ concentration. In addition, we and others have demonstrated powerful ca^2+^ buffering and extrusion in hair cells (e.g. Tucker and Fettiplace, 1995; Issa and Hudspeth., 1996; Frank et al., 2009 Pangrsic et al., 2015). As a result, the hair cells efficiently clear even massive synaptic ca^2+^ influx and establish a low bulk cytosolic ca^2+^ concentration (Beutner and Moser, 2001; Frank et al., 2009). We reason that these clearance mechanisms efficiently counter any ca^2+^ influx through ChR2. This will likely limit potential effects of ChR2 mediated ca^2+^ influx on ca^2+^ dependent replenishment of synaptic vesicles during ongoing stimulation.

We have now added the following in the discussion (starting in line 620):

“We note that ChR2, in addition to monovalent cations, also permeates ca^2+^ ions and poses the question whether optogenetic stimulation of IHCs could trigger release due to direct ca^2+^ influx via the ChR2. We do not consider such ca^2+^ influx to trigger exocytosis of synaptic vesicles in IHCs. Optogenetic stimulation of HEK293 cells overexpressing ChR2 (wildtype version) only raises the intracellular ca^2+^ concentration up to 90 nM even with an extracellular ca^2+^ concentration of 90 mM (Kleinlogel et al., 2011). IHC exocytosis shows a low ca^2+^ affinity (~70 µM, Beutner et al., 2001) and there is little if any IHC exocytosis for ca^2+^ concentrations below 10 µM, which is far beyond what could be achieved even by the highly ca^2+^ permeable ChR2 mutant (CatCh: ca^2+^ translocating channelrhodopsin, Mager et al., 2017). In addition, we reason that the powerful ca^2+^ buffering and extrusion by hair cells (e.g., Frank et al., 2009; Issa and Hudspeth, 1996; Pangršič et al., 2015; Tucker and Fettiplace, 1995) will efficiently counter ca^2+^ influx through ChR2 and, thereby limit potential effects on ca^2+^ dependent replenishment of synaptic vesicles during ongoing stimulation.”

5. Line 74, small typo on the last word: "OF" not "PF".

We corrected this

Reviewer #3 (Recommendations for the authors):My comments here are methodological. High-resolution views using ET reveal structures not visible using standard TEM. Hence, the sampling of ribbons may be important. Were all ribbon synapses located on a particular region of the inner hair cell membrane? Since location may relate to the spontaneous rate of vesicle release, location could contribute to variance in the data.

Unfortunately, we cannot tell about the specific location of the recorded ribbons. Tomograms were taken from different samples and different cells and likely cover modiolar and pillar sides of the IHCs. Indeed, this might contribute to data variability. Therefore, we now included this limitation in the manuscript. We wrote in line 370ff (results):

“Notably, one limitation of the study is that we cannot tell where the ribbon synapses were located within a given IHC. As synapse properties vary between pillar and modiolar IHC sides (Kantardzhieva et al., 2013; Merchan-Perez and Liberman, 1996; Michanski et al., 2019; Ohn et al., 2016), we expect this variance to contribute to the variance of synaptic properties observed for each condition. However, ribbon AZ were randomly sampled for ET; therefore, we expect this variability to be similar across conditions.”

Moreover, we now provide more details on the sampling procedure in the method part.

We wrote from line 1166 onwards:

“The embedded organ of Corti was approached tangentially from the pillar side to ensure that a row of several IHCs was obtained per section. For each sample, at least 20-30 grids with approximately 4-6 sections of 250 nm were prepared.”

And from line 1180 onwards:

“We randomly sampled ribbon AZs in the 250 nm sections used for tomograms. This resulted in differences in the cutting plane of the ribbon synapses represented by either a cross-section, a longitudinal section, or a section in between both (Supplementary file 3). The grids were systematically screened for ribbons that could be used for ET (0 – 4/5 ribbons per grid, often 0, frequently 1 or 2) to ensure that ribbon synapses from different IHCs and locations within IHCs were studied. It is difficult to determine the exact number of analyzed IHCs since we did not consistently acquire serial sections. Therefore, it is not possible for us to determine where the analyzed ribbon synapses were located within the IHC.”

Synaptic vesicles were segmented as spheres, but can be distorted and appear oblong due to tissue compression during sectioning (as is visible in Figure 6). How, then, were the diameters of these vesicles estimated, and did they form a large percentage of the total?

We agree that the virtual section was not ideally chosen and some cutting artifacts are visible. We rescreened our tomograms and the occurrence of oblong vesicles is rare.

All SVs were measured automatically as pointed out in the method section line 1247ff:

“All regular vesicles (also including CCVs) were annotated using a spherical scattered object at its maximum projection of the vesicle in the tomogram, marking the outer leaflet (for CCV: outer leaflet, excluding the CC) of the vesicles. Due to excellent ultrastructural preservation obtained from HPF/FS, most of the regular vesicles (also including CCVs) were spherical and were annotated using a spherical Scattered object at its maximum projection of the vesicle in the tomogram (Chakrabarti et al., 2018), marking the outer leaflet (for CCV: outer leaflet, excluding the CC) of the vesicles. The diameter of the sphere was manually adjusted for each vesicle to rule out any possible errors with the cutting plane or compression. The vesicle radius (r) was determined automatically (Helmprobst et al., 2015) using the program imodinfo option –p of IMOD software package (Kremer et al., 1996).”

The authors indicate that synapses were generally incomplete within the 250 nm tissue thickness, and this feature can lead to erroneous statistical differences, such as between non-stimulated controls. How can the data be normalized, then, or at least a sub-population of fortuitous complete AZs analyzed, to justify differences between control and stimulated animals?

Thank you for this valid concern. In response to this concern, we included more information about the range of tomograms and their cutting planes and addressed the possible limitations of the techniques.

We added a new table to the material and method section (Supplementary file 4) to the Materials and methods section, showing the range of virtual sections for each condition. Furthermore, we added a table to the material and method section (Supplementary file 3) that now states, how many ribbons per condition were hit longitudinal, in a cross-section or between these orientations.

To control for the variations resulting from different ribbon proportions/angles, we focused on assessing the fractions of tethered SVs or docked SVs, rather than their total numbers. Normalization to the AZ area, as it is often done for the quantification of SVs at conventional synapses, is not perfectly suitable for ribbon synapses. The AZ area does not necessarily correlate with the ribbon size and often exceeds the tomogram, as the reviewer also mentioned above. As said, for these reasons we chose to present the fractions as also done in previous studies (Chakrabarti et al., 2018 or Kroll et al., 2020, JCS).

We further added that this is one limitation of the study in line 398:

“As the full inclusion of the ribbon is relatively rare in 250-nm sections (Figure 6A-D), representing a limitation of the study, we…”

Probably my confusion, but why do the numbers in Table 1 differ from those in Figure 6?

We rechecked the numbers, the difference likely resulted from pooling the data sets, as described in the manuscript, and from the fact that we did not analyze the RA-SVs in all tomograms. We tried to be clear now in the relevant legends and in the tables as well as in the method part.

It would be interesting to know if the increased number of docked vesicles was at the expense of tethered or non-tethered vesicles by linking the measures from individual presynaptic densities.

This is indeed the case. The quantification of the morphological sub-pools (non-tethered, tethered and docked) was performed in fractions of MP-SVs, (we as shown in Figure 6F). Thus, that the number of docked SVs increased upon long stimulation indeed comes at the expense of a significantly smaller and the fraction of tethered SVs decreased significantly. For example, when comparing Figure 6B and D (top views), a clear difference in the proportion of docked/tethered SVs is visible, which we now also mention in the manuscript.

Double-tilt ET provides greater spatial resolution, although at the cost of electron dose and time to collect data. Have the authors experimented with double tilt protocols and, if so, why not employ them?

Thanks for the recommendation, which we will take into consideration when planning future studies. We have very little experience with double tilts, but decided to go for single tilt tomograms for the reasons mentioned by the reviewer.

Ectopic expression of channelrhodopsin was described, but the way this information factored into the experimental design was not clear.

We believe, the reviewer refers to the following sentence:

“This resulted in the ectopic expression (e.g. in non-IHCs in the cochlea) of EGFP and/or ChR2-H134R-EYFP in the absence of Cre-recombinase”.

We apologize for not being fully clear, we designed a second pair of primers to control for this recombination and used only animals not-recombined. We stated this now in the manuscript in line 855/856:

“We excluded animals that showed general recombination for our analysis.”

Were distances from vesicle to presynaptic membrane measured only within sections, or in 3D space? Even a moderately tangential section plane relative to the presynaptic membrane can lead to error in this measurement.

We apologize for not being precise in describing the quantification. We measured the shortest distance from the outer SV membrane to the AZ membrane in the virtual section, which showed the largest SV diameter, we now stated in the method part in line 1237ff:

“The distances of the SVs to individual structures like the AZ membrane, PD and ribbon were measured using the Measure drawing tool in 3dmod at the maximum projection of the vesicle in the tomogram as done previously (Chakrabarti et al., 2018).”

Provide a discussion of why two control groups were chosen. In most cases their differences with stimulated groups were similar, except in Figure 6F.

It is true that the two control groups showed only minor differences, however, we chose these two groups, because we wanted to control for the effect of light on IHC ribbon synapses and for the genetic background. Therefore, we prefer to keep them as separate data sets.

We added the following statements to the manuscript (line 353ff):

“We included two controls (i) B6J under light stimulation (B6J Light; Figure 6A) to address potential direct effects of light exposure and (ii) ChR2-expressing IHCs (Ai32VC cre^+^ and Ai32KI cre^+^) without any light stimulation (ChR2 NoLight; Figure 6B) as controls with the same genetic background of the stimulated samples.”

And in line 364ff:

“This difference could be due to differences in their genetic background and/or the expression of ChR2 and highlights the importance of including ChR2 NoLight controls in our experimental design. Our estimates of MP-SVs/tomogram in control conditions compare to previous reports (B6J Light 9.73 ± 0.796; ChR2 NoLight 14.88 ± 0.935; Wt Rest 11.7 ± 0.75 (Chakrabarti et al., 2018); Wt Rest 18.5 ± 1.5 (Kroll et al., 2020)).”

Reviewer #4 (Recommendations for the authors):(1) It would be of interest to look at the size of the docked vesicles independently (rather than just membrane proximal). These would presumably be the population one would want to look at.

We agree and added this information to Figure 7 and the Table 3. The 1 and 2 docked SVs found in B6J Light and ChR2 NoLight conditions, respectively, have a larger diameter (53.7 nm) than the average from the ChR2 stimulated conditions (51.29 nm for ChR2 ShortStim and 47.97 nm for ChR2 Long Stim). However, we did not perform any statistical test given the low number of docked SVs in the control conditions.

(2) I assume the results in figure 3 were also carried out with the same concentrations of Cs+ and TEA. Is that correct? I couldn't find that in the description of those experiments.

Thank you very much for pointing this out, the experiments are described in the method section from line 878 onwards. These experiments were done in a standard extracellular solution, which did not contain TEA or Cs.

However, we performed control experiments to address the concern using two different solutions (Author response image 1). For these control experiments, the first solution, without TEA and Cs, had the same composition as the resting solution used in Chakrabarti et al., 2018 (5 mM KCl, 136.5 mM NaCl, 1 mM MgCl2, 1.3 mM CaCl_2_, 10 mM HEPES, and 11.1 mM D-glucose, pH 7.2, ~300 mOsmol/l). The second solution, with TEA and Cs, was the same one use in the current study (2.8 mM KCl, 126 mM NaCl, 1 mM MgCl2, 2 mM CaCl_2_, 10 mM HEPES, 20 mM TEA-Cl, 1 mM CsCl, and 11.1 mM D-glucose, pH 7.2, ~300 mOsm/l).

We performed perforated patch-clamp recordings of IHCs in current-clamp mode (panel A) to record the IHC potential before and after perfusion of the solutions (panel Ai). Perfusion of the solution with TEA and Cs slightly hyperpolarized the IHC potential by 5.56 ± 1.615 mV (4 cells; p-value = 0.1250, Wilcoxon matched-pairs signed rank test).

Additionally, we performed ruptured patch-clamp recordings from two afferent boutons and recorded spontaneous EPSCs before and after perfusion (panel B). EPSCs were detected using Neuromatic (Rothman and Silver, 2018). For the first bouton, the number of EPSCs before perfusion was 68 EPSCs in one minute. After perfusion, there was a slight reduction to 49 EPSCs in one minute. For the second bouton, the number of EPSCs remained constant to 1 EPSC in one minute before and after perfusion of solution with TEA and Cs. Therefore, the rare occurrence of docked SVs in unstimulated conditions is not a reflection of a depolarized IHC. This observation probably reflects the natural state of the IHC ribbon synapse.

(due to the specific insertion mechanism of th3) line 794: "We found no alterations in the size of the MPSV and RA-SV pools among the various conditions and controls (Figure 6E), except for a larger MP-SV pool in ChR2 NoLight compared to B6J Light. This might reflect different proportions of ribbon-type AZs contained in the tomograms obtained from 250 nm sections that do not always include the full AZs." Why would such an artifact be limited to these controls? Shouldn't we be worried about similar artifacts in other aspects of the data?

We agree and rephrased the statement. We now clarified and put the different cutting planes as well as the different proportions of the ribbon synapses included in the tomograms as a general limitation and a possible contribution to variance. Furthermore, specifically for the difference between these two conditions, we stated the following from line 364 onwards:

We now wrote:

“This difference could be due to differences in their genetic background and/or the expression of ChR2 and highlights the importance of including ChR2 NoLight controls in our experimental design. Our estimates of MP-SVs/tomogram in control conditions compare to previous reports (B6J Light 9.73 ± 0.796; ChR2 NoLight 14.88 ± 0.935; Wt Rest 11.7 ± 0.75 (Chakrabarti et al., 2018); Wt Rest 18.5 ± 1.5 (Kroll et al., 2020)).”

To control for the variations resulting from different ribbon proportions/angles, we focused on assessing the fractions of tethered SVs or docked SVs, rather than their total numbers. Normalization to the AZ area, as it is often done for the quantification of SVs at conventional synapses, is not perfectly suitable for ribbon synapses. The AZ area does not necessarily correlate with the ribbon size and often exceeds the tomogram, as the reviewer also mentioned above. As said, for these reasons we chose to present the fractions as also done in previous studies (Chakrabarti et al., 2018 or Kroll et al., 2020, JCS).

We further added that this is one limitation of the study in line 398:

“As the full inclusion of the ribbon is relatively rare in 250-nm sections (Figure 6A-D), representing a limitation of the study, we…”